# Determinants of chemoselectivity in ubiquitination by the J2 family of ubiquitin-conjugating enzymes

Anuruti Swarnkar[1], Florian Leidner [ID][2], Ashok K Rout[3,5], Sofia Ainatzi [ID][4], Claudia C Schmidt [ID][1,6], Stefan Becker [ID][3], Henning Urlaub[4], Christian Griesinger[3], Helmut Grubmüller [ID][2] & Alexander Stein [ID][1][✉]

## Abstract

Ubiquitin-conjugating enzymes (E2) play a crucial role in the attachment of ubiquitin to proteins. Together with ubiquitin ligases (E3), they catalyze the transfer of ubiquitin (Ub) onto lysines with high chemoselectivity. A subfamily of E2s, including yeast Ubc6 and human Ube2J2, also mediates noncanonical modification of serines, but the structural determinants for this chemical versatility remain unknown. Using a combination of X-ray crystallography, molecular dynamics (MD) simulations, and reconstitution approaches, we have uncovered a two-layered mechanism that underlies this unique reactivity. A rearrangement of the Ubc6/Ube2J2 active site enhances the reactivity of the E2-Ub thioester, facilitating attack by weaker nucleophiles. Moreover, a conserved histidine in Ubc6/Ube2J2 activates a substrate serine by general base catalysis. Binding of RING-type E3 ligases further increases the serine selectivity inherent to Ubc6/Ube2J2, via an allosteric mechanism that requires specific positioning of the ubiquitin tail at the E2 active site. Our results elucidate how subtle structural modifications to the highly conserved E2 fold yield distinct enzymatic activity.

**Keywords** Ubiquitin-conjugating Enzyme; ER-associated Protein Degradation; Noncanonical Ubiquitination; RING E3 Ligase; Posttranslational Modification
**Subject Categories** Post-translational Modifications & Proteolysis; Structural Biology

## Introduction

Posttranslational modification of proteins with ubiquitin regulates many aspects of eukaryotic biology. An enzymatic cascade of transacylation reactions mediates ubiquitination. First, the ubiquitin-activating enzyme (E1) converts the chemically relatively inert carboxyl terminus of ubiquitin (Ub) into a Ub-adenylate, followed by the formation of a thioester with its active site cysteine. Ubiquitin is then conjugated to the active site cysteine of the ubiquitin-conjugating enzyme (E2). Lastly, ubiquitin is transferred to a nucleophilic group in a substrate, usually the ε-amino group of a lysine residue, forming an isopeptide bond. Ubiquitin ligases (E3) catalyze either a direct nucleophilic attack of a substrate on the E2-Ub thioester, as is the case for RING E3s, or they transiently form covalent E3-Ub thioester, which the substrate then attacks nucleophilically. E6-AP carboxyl terminus (HECT), RING-in-between-RING (RBR) and recently identified RING-Cys-Relay (RCR) E3s follow the latter mechanism (Horn-Ghetko and Schulman, 2022; Wang et al, 2020; Zheng and Shabek, 2017). Repetitions of this cascade lead to the formation of ubiquitin chains, in which ubiquitin can connect to any of the seven lysines and the N-terminus of another ubiquitin, thus giving rise to a large variety of possible ubiquitin chains (Komander and Rape, 2012; Kwon and Ciechanover, 2017; Yau and Rape, 2016). Interaction of receptor proteins with specific ubiquitin chains or single ubiquitin moieties determine downstream reactions such as the proteasomal degradation of proteins with Lys48-linked ubiquitin chains (Husnjak and Dikic, 2012; Oh et al, 2018).

Besides the transthioesterification onto cysteines, lysine modification and the chemically similar modification of the amino terminus of polypeptides, ubiquitination can also target the side chains of serines and threonines, as well as hydroxy groups of non-protein molecules (Dikic and Schulman, 2023; Kelsall, 2022; McClellan et al, 2019; McDowell and Philpott, 2013; Sakamaki and Mizushima, 2023). These alcohol groups are weaker nucleophiles than either thiols or amines, raising the question of how a machinery tailored for amine modification has adapted to modify the less reactive hydroxy groups. Previous findings suggest that this adaptation falls into two broad mechanistic categories. In the first, exemplified by the RCR E3 MYCBP2 (Pao et al, 2018), the RBR HOIL-1 (Kelsall et al, 2022), and the atypical RING E3 RNF213 (Ahel et al, 2020; Otten et al, 2021), ubiquitination occurs via transiently formed E3-Ub thioesters. Subsequently, ubiquitin is esterified to a hydroxyl group in a substrate molecule. In these cases, specific structural elements of E3 ligases beget reactivity towards hydroxy groups. In reactions mediated by classical RING E3s, the E2 appears to encode reactivity toward hydroxy groups.

[1]Research Group Membrane Protein Biochemistry, Max Planck Institute for Multidisciplinary Sciences, Am Fassberg 11, 37077 Göttingen, Germany. [2]Department of Theoretical and Computational Biophysics, Max Planck Institute for Multidisciplinary Sciences, Am Fassberg 11, 37077 Göttingen, Germany. [3]Department of NMR-based Structural Biology, Max Planck Institute for Multidisciplinary Sciences, Am Fassberg 11, 37077 Göttingen, Germany. [4]Research Group Bioanalytical Mass Spectrometry, Max Planck Institute for Multidisciplinary Sciences, Am Fassberg 11, 37077 Göttingen, Germany. [5]Present address: Institut für Chemie und Metabolomics, Universität zu Lübeck, 23562 Lübeck, Germany. [6]Present address: ETH Zürich, Otto-Stern-Weg 3, 8093 Zürich, Switzerland. ✉E-mail: alexander.stein@mpinat.mpg.de

These types of reactions are particularly prevalent in endoplasmic reticulum-associated protein degradation (ERAD), a quality control pathway that degrades membrane and soluble proteins of the ER (Christianson and Carvalho, 2022; Christianson et al, 2023). In ERAD, hydroxy group modifications occur in conjunction with members of the J subfamily of E2s, which in humans includes Ube2J1 and Ube2J2, and *S. cerevisiae* Ubc6.

In yeast, degradation of lysine-free versions of ERAD substrates requires both Ubc6 and the E2 Ubc7, along with its cofactor Cue1. In contrast, the lysine-containing counterparts can be degraded with Ubc7/Cue1 alone (Habeck et al, 2015; Lips et al, 2020; Mehrtash and Hochstrasser, 2022; Weber et al, 2016). Notable examples for this phenomenon include Sbh2, a tail-anchored membrane protein targeted by the E3 Doa10, and CPY*, a misfolded point mutant of carboxypeptidase Y and a Hrd1 substrate. In addition, Ubc6 undergoes autoubiquitination on a serine residue in collaboration with Doa10, leading to its degradation (Walter et al, 2001; Weber et al, 2016). In mammals, Ube2J2 is required for the degradation of lysine-free MHC I heavy chain during viral infections, a process mediated by the viral E3 mK3 or TMEM129 (van den Boomen et al, 2014; Wang et al, 2007; Wang et al, 2009). Ube2J2, together with the E3 MarchF6, also ubiquitinates squalene epoxidase (SQLE) on a serine residue (Chua et al, 2019). In the case of Ube2J1, serine ubiquitination has been reported to occur in lysine-free MHC I heavy chain in conjunction with Hrd1 (Burr et al, 2011; Burr et al, 2013). Mass spectrometry has confirmed serine ubiquitination in both Ubc6 autoubiquitination and SQLE modification. In other instances, mutagenesis of lysine residues and the susceptibility of ubiquitin modifications to alkaline hydrolysis have suggested the presence of ubiquitin oxyesters (Ferri-Blazquez et al, 2023). Finally, purified Ube2J2 has been shown to transfer ubiquitin onto serine residues of small peptides in the absence of an E3 ligase (Abdul Rehman et al, 2024).

Canonical E2s use two key catalytic strategies to mediate lysine modification. First, they stabilize the high-energy anionic tetrahedral intermediate formed after nucleophilic attack on the E2-Ub thioester. This stabilization occurs through hydrogen bonding, involving an asparagine side chain within the highly conserved HPN motif located near the active site cysteine (Wu et al, 2003; Yunus and Lima, 2006). Second, the CES/D site (conserved E2 serine/aspartate) increases the nucleophilicity of the incoming amino group by stabilizing its non-protonated form (Valimberti et al, 2015). The intrinsic reactivity of E2s toward amines is drastically increased upon RING E3 binding, which restricts the flexibility of ubiquitin attached to the E2 and stabilizes so-called "closed" conformations (Branigan et al, 2020; Branigan et al, 2015; Dou et al, 2012; Plechanovova et al, 2012; Pruneda et al, 2011). The specific features of J family E2s that enable them to ubiquitinate serine or threonine residues remain unclear. These E2s consist of an N-terminal UBC domain, a presumably unstructured linker region, and a C-terminal transmembrane segment. Unlike canonical E2s, J family E2s lack important functional elements such as the HPN motif and the CES/D site (Burroughs et al, 2008) (Appendix Fig. S1), but show a greater tendency to form closed conformations in the absence of a RING domain compared to other E2s (Lips et al, 2020).

Here, we investigated how the J2 family mediates the ubiquitination of hydroxy groups. Based on experimental and simulated structures of free and ubiquitin-attached Ubc6 we identify variations of the conserved E2 architecture. Reconstitution experiments reveal two mechanisms by which these variations function: First, a remodeled active site exhibits a generally higher but promiscuous reactivity towards the side chains of Lys, Ser, Tyr, and even His. Second, a conserved histidine activates serine residues in the substrate by general base catalysis. RING E3 binding to ubiquitin-loaded Ubc6 not only enhances Ubc6's overall reactivity but alters its reactivity profile, causing Ubc6 to strongly favor serine over other amino acid side chains. These characteristics are largely conserved in Ubc6's human homolog Ube2J2.

# Results

## Structure of Ubc6

To uncover structural determinants of noncanonical E2 activity, we determined two structures of the UBC domain of yeast Ubc6 by X-ray crystallography to resolutions of 1.21 and 1.33 Å (Fig. 1A), which were virtually identical (RMSD 0.394 Å, Appendix Table S1). As expected from the overall conservation and the structure of human Ube2J2 (PDB 2F4W) (Sheng et al, 2012), Ubc6 adopts the characteristic α/β-fold of the UBC family, but compared to canonical E2s, one can discern crucial differences (Fig. 1B). First, the loop connecting the four-stranded β-sheet and the active site, which harbors the HPN motif in canonical E2s, is shorter, and contains a conserved GRF motif (called GRF loop). Second, a 12 amino acids long segment, which we will call the Thr-flap, directly connects the crossover helix α2 with helix α4, while the short helix α3 of canonical E2s is lacking. A hydrogen bonding network centered on Arg79 of the GRF motif connects these two segments with each other, the crossover helix α2 and the active site cysteine. In contrast to the published Ube2J2 structure (Sheng et al, 2012), the active site is well resolved (Figs. 1C and EV1A). It forms a distorted helical structure stabilized by hydrogen bonding between the side chain of Ser91 and Leu88. On top of the active site Cys87 lays an aromatic ring (Tyr93), engaged in a π–cation interaction with Arg85, which itself forms a salt bridge with Asp92. A conserved histidine (His94) points towards the active site Cys87, consistent with a role in the enzymatic mechanism of the ubiquitin transfer reaction.

Next, to assess conservation of structural element identified in Ubc6, we generated a multiple sequence alignment (MSA) of Ube2J1 and Ube2J2 homologs from a diverse set of eukaryotes and constructed a phylogenetic tree (Appendix Fig. S2). This confirmed that Ubc6 is a Ube2J2 homolog (Burroughs et al, 2008; Lester et al, 2000), and showed that the split into the J1 and J2 subfamilies seems to be old. Most organisms contain both variants, but some lineages have lost a Ube2J1 homolog, e.g., *S. cerevisiae*, *S. pombe*, and *D. melanogaster*. Both subfamilies contain the characteristic GRF motif and a highly conserved insertion immediately C-terminal to the catalytic cysteine. This insertion is invariable in length, with three additional amino acids as compared to canonical E2s such as Ube2D2 or Rad6, and features a conserved histidine (His94 in Ubc6). The Thr-flap is only conserved in Ube2J2 homologs (see Fig. 1D; Appendix Fig. S3A for conserved elements of J2 and J1 families, respectively, and Appendix Fig. S3B for an Alphafold3 structure prediction of Ube2J1).

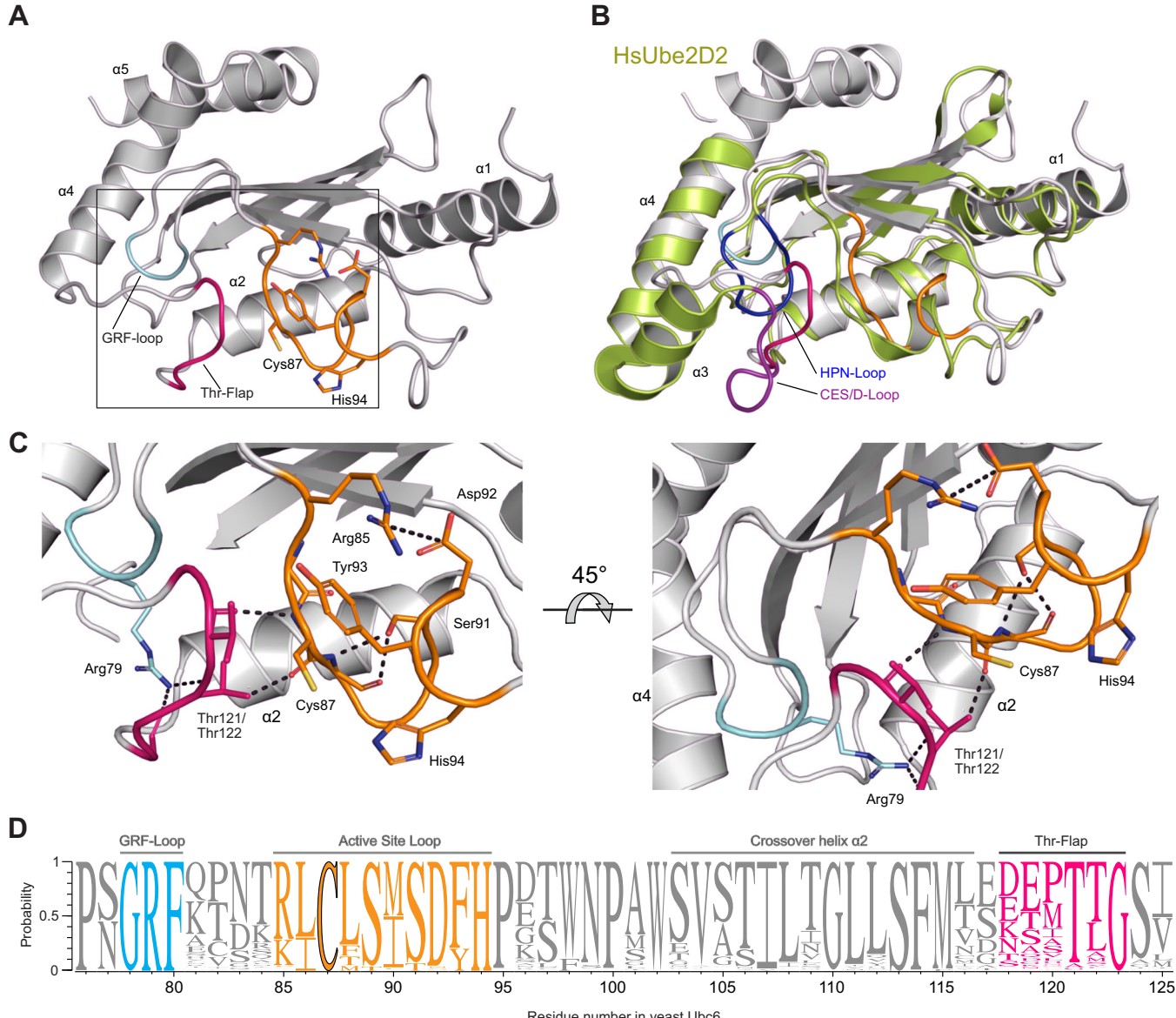

**Figure 1. Active site architecture of the J2 family E2 Ubc6.**

(A) Cartoon representation of the 1.33 Å resolution X-ray structure of the yeast Ubc6 UBC domain in gray (PDB: 9EN5). The GRF loop, the active site loop and the Thr-flap are colored in cyan, orange, and pink, respectively. Side chains for Arg85, Cys87, and Asp92-His94 of the active site loop are shown as sticks. Numbering of helices according to the convention for canonicals E2s. (B) Structural overlay the Ubc6 UBC domain with the canonical E2 Ube2D2 (PDB: 6SQO). Coloring for Ubc6 as in (A), Ube2D2 in green. To highlight structural rearrangement in Ubc6 compared to canonical E2s, the HPN motif-containing loop (HPN-loop) and the corresponding GRF loop in Ubc6 are shown in blue and cyan, respectively. Likewise, the CES/D site-containing loop and the corresponding Thr-flap of Ubc6 are shown in purple and pink, respectively. (C) Zoomed-in views of the active site arrangement from (A). Black dashed lines indicate electrostatic interactions between conserved residues. Coloring and labeling as in (A). (D) Weblogo for the active site proximal region of Ube2J2/Ubc6 homologs, generated from an alignment of 199 sequences from diverse set of eukaryotes. Conserved features are colored as in (A). Symbol heights within a stack represent relative frequency of each residue. The residue numbering corresponds to *S. cerevisiae* Ubc6. The alignment used for generating the Weblogo can be found in Dataset EV1.

## Structures of ubiquitin-loaded Ubc6

We also determined the structure of a ubiquitin-conjugated form of Ubc6 to 2.6 Å resolution (Fig. 2A; Appendix Table S2). To stabilize the linkage between ubiquitin and the UBC, we mutated the active site cysteine (Cys87) to Lys, resulting in an isopeptide bond in place of of the native thioester (Plechanovova et al, 2012). The

asymmetric unit contains two copies of Ubc6-Ub, both adopting so-called "closed" conformations. These conformations are characterized by contacts between a hydrophobic patch centered on Ile44 in ubiquitin and the crossover helix α2 in the UBC domain (Fig. 2B) (Branigan et al, 2015; Dou et al, 2012; Plechanovova et al, 2012). The Ubc6 active site is virtually unchanged in one copy of the heterodimer when compared to the ubiquitin-free form.

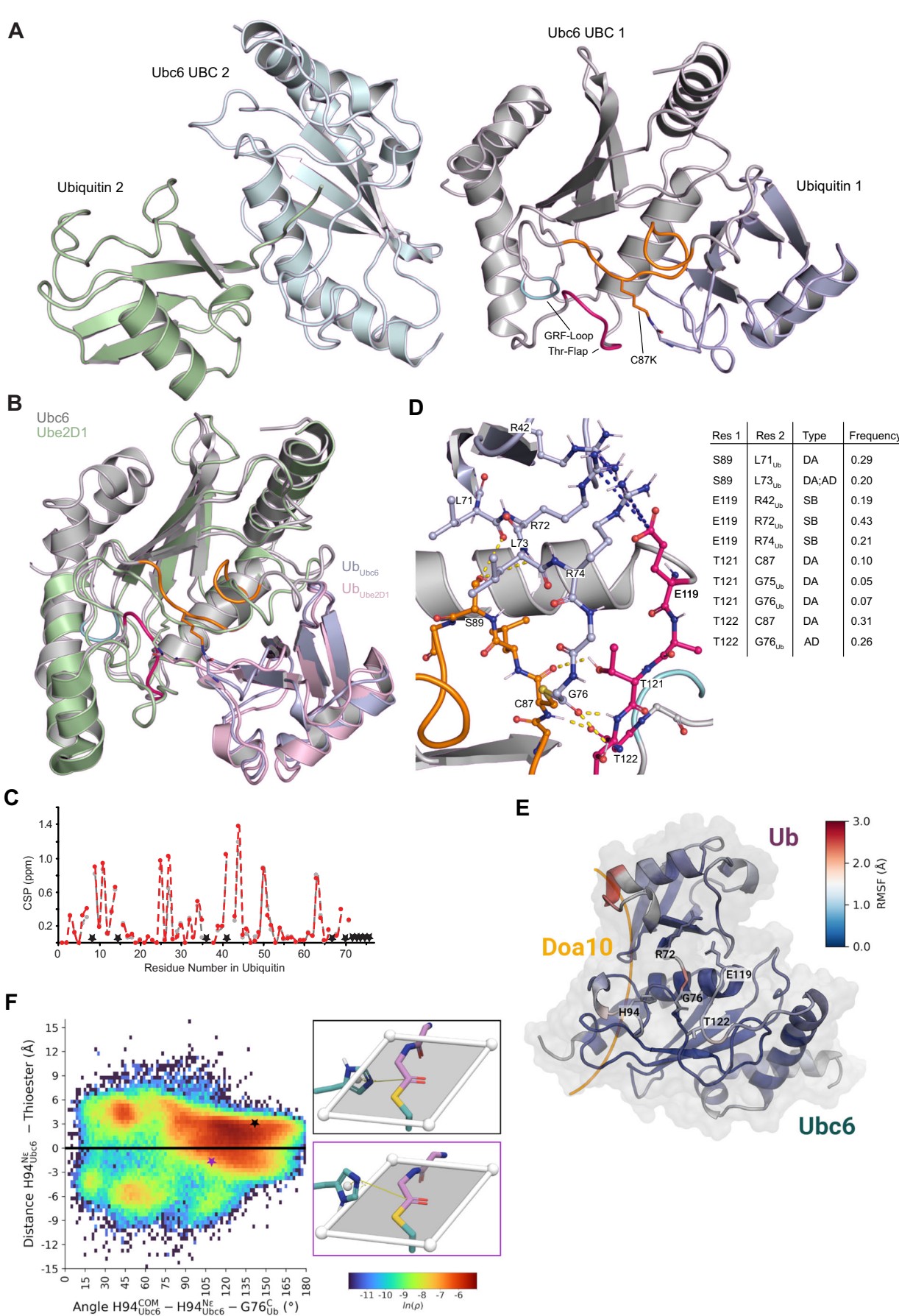

**Figure 2. Structure of Ubc6 UBC domain linked to ubiquitin.**

(A) Cartoon representation of the Ub-loaded UBC domain of Ubc6. The native thioester linkage was replaced with an isopeptide bond by mutating Cys87 to Lys. The asymmetric unit contains two copies of this assembly. In the first copy, Ubc6 is labeled as in Fig. 1B and Ub in light blue, in the second copy, Ubc6 and Ub are labeled in pale cyan and pale green, respectively. (B) Structural overlay of copy 1 of Ubc6-Ub as in (A) and the crystal structure of the heterotrimeric Rnf4-Ube2D1-Ub complex (PDB: 4AP4, Rnf4 not shown) with Ube2D1 and Ub in pale green and light pink, respectively. (C) Chemical shift perturbation (CSP) in $^{15}$N-ubiquitin, disulfide linked to WT Ubc6 and Ubc6 S89A, in gray and red, respectively. The dashed lines along with a star on the X-axis correspond to the residues of ubiquitin that display NMR signal broadening from interaction with Ubc6 as a result of slowed exchange rate. (D) Zoomed-in view of the ubiquitin attachment site in the native Ubc6-Ub complex derived from MD simulations. Parts of the active site loop, the Thr-Flap and the ubiquitin tail are shown as sticks with coloring as in Fig. 1 and in (A). Hydrogen bonds and salt bridges are indicated as dashed yellow and blue lines, respectively. The table on the right provides the frequencies and types of interactions between key residues (Res 1 and 2) over all simulation replicates. DA donor-acceptor; AD acceptor-donor; SB salt bridge. (E) Cartoon representation of Ubc6-Ub as derived from molecular dynamics (MD) simulations of the native, thioester-linked complex. Key residues are shown as sticks. The putative Doa10 binding site is highlighted in yellow. Coloring by per-residue root mean square fluctuation (RMSF) over a total of 100 µs of MD simulations. (F) Configurations of His94 relative to the thioester are depicted as a function of orientation angle and distance from the thioester plane. The orientation is determined as the angle between the center of mass (COM) of the imidazole ring, the unprotonated ε-nitrogen ($N_ε$) and the carbonyl carbon of Gly76$_{Ub}$. The offset is determined by the signed distance between $N_ε$ and the thioester plane. Positive distances represent orientations where the ε-nitrogen faces the solvent cavity, negative distances denote configurations where the ε-nitrogen is situated on the bulk solvent side of the thioester plane. Two example configurations are shown on the panels to the right. The colored stars indicate the orientation and offset values of these configurations.

However, in the second copy, there is a drastic conformational change for residues Lys87 to Trp98 (Fig. EV1B). While this alternative conformation could be attributed to crystal packing, it suggests some conformational flexibility in this region, which coincides with an unresolved part in the Ube2J2 structure (PDB 2F4W) (Sheng et al, 2012). We concentrated our analysis on the first copy. The backbone of the C-terminal ubiquitin tail is traceable in the electron density map, but most side chains in this region are not well resolved (Fig. EV1C). The conserved Ser89 forms hydrogen bonds with the backbone amide of Leu73$_{Ub}$ and the carbonyl oxygen of Leu71$_{Ub}$. The hydrophobic side chains of both leucines shield these hydrogen bonds from solvent.

To test if hydrogen bonding by Ser89 significantly contributes to the stability of closed Ubc6-Ub conformations we used NMR spectroscopy. Analysis of [$^{15}$N,$^1$H]-HSQC spectra of Ubc6 coupled via a disulfide bond to $^{15}$N-labeled ubiquitin G76C showed chemical shift perturbations (CSPs) typical of closed ubiquitin conformations, confirming that such conformations of Ubc6 are also prominently present in solution (Lips et al, 2020). We observed very similar CSPs for Ubc6 S89A coupled to $^{15}$N-ubiquitin (Figs. 2C and EV1D). Furthermore, rotational correlation times $τ_C$ determined by [$^{15}$N,$^1$H]-TRACT experiments, which report on the interdomain flexibility of the attached ubiquitin moiety relative to the UBC domain, showed that Ub-adducts to WT and mutant Ubc6 do not differ significantly and thus adopt closed conformations to a similar extent (Fig. EV1E). Therefore, the strong preference for Ser at this position cannot be explained by an impact on the stability of closed conformations.

To characterize potential distortions caused by the artificial isopeptide linkage, we complemented the structural studies with molecular dynamics (MD) simulations. For this purpose, we substituted Lys87 for the native cysteine. Simulations required re-parameterization of the thioester force field parameters to obtain good agreement with ab initio calculations (Fig. EV2A–C). To reduce model bias, we used a simulated annealing protocol to generate a diverse set of configurations. A total of 1024 conformations of the native complex were generated (Fig. EV2D). Inspection of these structures revealed that, compared to the crystal structure, the shorter side chain of Cys87 shifts the C-terminal ubiquitin tail towards the Thr-flap. Together with adjacent regions in Ubc6, the C-terminal tail of ubiquitin walls off a solvent-filled cavity occupied by a median of seven water

molecules (Fig. EV2E,F). A network of hydrogen bonds stabilizes the C-terminal tail of ubiquitin in the Ubc6 active site (Fig. 2D). Further, the carbonyl oxygen of the thioester forms hydrogen bonds with the backbone amide and side chain of Thr122, as well as the side chain of Thr121. Finally, a salt bridge formed between Glu119 and one of three arginine residues in ubiquitin (Arg42$_{Ub}$, Arg72$_{Ub}$, and Arg74$_{Ub}$) restricts access to the cavity from the surrounding solvent.

To gain further insight into the dynamics of the native complex, we performed 100 independent one-µs all-atom MD simulations. In accordance with the NMR measurements, the Ubc6-Ub complex remained in the closed conformation in all replicates. The loops forming the Ubc6 active site and the ubiquitin tail retained their overall geometry, but exhibited considerable flexibility during simulations, with an average RMSF from the initial model between 1 and 2 Å (Fig. 2E). In line with the fact that the active site region (Cys87–Trp98) was unresolved in the Ube2J2 structure and adopted two different conformations in our Ubc6-Ub structure, this region was particularly dynamic (Fig. EV2G). Furthermore, the interaction between the thioester oxygen and Thr122 and Thr121 was maintained in all simulations. In this configuration, the carbonyl oxygen faces away from His94, thus excluding the possibility that His94 promotes ubiquitin transfer by stabilizing the tetrahedral intermediate. Alternatively, His94 could facilitate nucleophilic attack by accepting a proton from an incoming nucleophile. For this to occur, an unprotonated nitrogen of His94 must be oriented ~3 Å away from the thioester plane toward the carbonyl. During simulations, the ε-nitrogen protonated tautomer remained in a conformation incompatible with base catalysis. For the δ-nitrogen protonated tautomer, however, we identified several configurations that satisfy the geometric criteria for general base catalysis (Figs. 2F and EV2H–J). Due to the solvent-filled cavity, nucleophilic attack could occur from either side of the thioester plane. However, an attack from the interior of the cavity seems unlikely because its average volume is too small to accommodate looping in and out of a polypeptide substrate, and the Glu119-Arg$_{Ub}$ salt bridges isolate it from bulk solvent for the most part. In contrast, nucleophilic attack from the bulk solvent side of the thioester plane is supported by structures of E2-RING-substrate complexes that capture or mimic the state immediately after ubiquitin transfer to the substrate (Liwocha et al, 2024; Reverter and Lima, 2005). Access from this side of the thioester plane is

restricted by Tyr93, but the thioester carbon is occasionally exposed allowing nucleophilic attack.

Together, MSAs, the X-ray structures and MD models identify potential signature residues of Ubc6/Ube2J2 and suggest how variations specific to the J2 family result in reactivity towards hydroxy groups of amino acids: First, the Thr-flap takes over the role of the HPN motif in canonical E2s, activating the thioester by partial protonation or stabilizing the tetrahedral intermediate. Second, His94 activates a nucleophilic amino acid of the substrate by general base catalysis. Third, additional interactions between the ubiquitin tail and Ser89 and Glu119 in Ubc6 confine ubiquitin motion and stabilize specific conformations of the ubiquitin tail relative to Ubc6. In the following, we will test these hypotheses.

## Mutations in Ubc6 signature motifs stabilize Ubc6 in yeast

In yeast, Ubc6 is unstable and degraded by the proteasome with a half-life of ~1 h. Degradation depends on its own enzymatic activity that leads to an intramolecular autoubiquitination and is enhanced by its cognate E3 Doa10 (Schmidt et al, 2020; Walter et al, 2001; Weber et al, 2016). We reasoned that mutations in the proposed crucial regions would impair autoubiquitination activity and thus lead to increased stability. To test this, we introduced different mutants of a previously described internally HA-tagged version of Ubc6 into ubc6 knockout cells and measured Ubc6-HA stability in cycloheximide chase experiments (Walter et al, 2001). This confirmed that Ubc6-HA is unstable and that mutation of its active site Cys87 to Ala (C87A) results in stabilization. Mutations S89A, H94A, the more conservative H94Q, and to a lesser degree T121A resulted in stabilization of Ubc6-HA (Fig. 3A–C). These results confirm that the identified motifs are important for Ubc6 function.

## Active site proximal mutants impair ubiquitin transfer

We used reconstituted systems to understand how specific structural elements in Ubc6 contribute to its reactivity. Mutations of His94 to alanine or glutamine, or of Ser89 to alanine did not discernably affect the E1-mediated loading reaction (Fig. EV3A). When we incubated Ubc6-liposomes with E1, ubiquitin and ATP, we observed Ubc6 autoubiquitination (Fig. 4A), as previously reported (Schmidt et al, 2020). The Ubc6 mutants S89A and H94A exhibited drastically reduced autoubiquitination, whereas the H94Q mutant retained an intermediate activity. When we co-reconstituted full-length Doa10, autoubiquitination of WT Ubc6 was drastically faster, whereas autoubiquitination of mutant Ubc6 versions was only slightly enhanced compared to reactions without Doa10 (Fig. 4B,C). In the same experiments, we also observed ubiquitin transfer to Doa10, visible as a smearing out of the Doa10 band towards higher molecular weight. Similar to Ubc6 auto-ubiquitination, replacing WT Ubc6 with the S89A or H94A mutants almost entirely abolished Doa10 ubiquitination, whereas the H94Q mutant maintained some Doa10 ubiquitination.

## Ubiquitination of hydroxy amino acids and lysine residues in Sbh2

Next, we tested how mutations in Ubc6 affect ubiquitin transfer to the Doa10 substrate Sbh2. We chose Sbh2, because a mutant

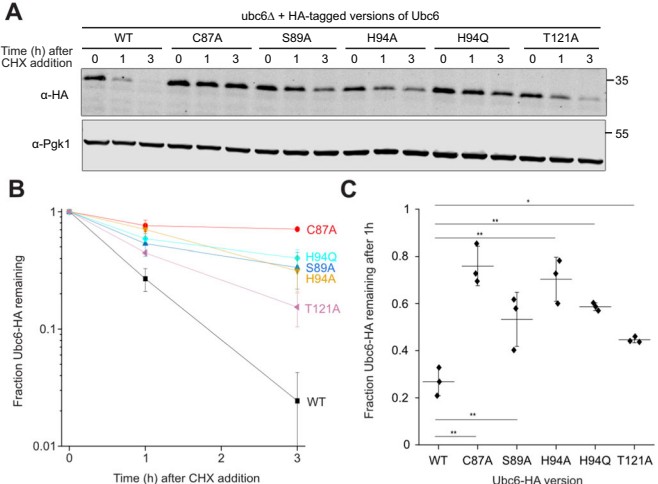

**Figure 3. Mutations of residues in the active site loop and Thr-flap stabilize Ubc6.**

(A) Cycloheximide (CHX) chase analysis of HA-tagged wild-type (WT) Ubc6 and the indicated mutants expressed from a plasmid under the control of the endogenous *UBC6* promoter in a *S. cerevisiae* ubc6Δ background. The degradation of Ubc6-HA was followed after inhibition of protein synthesis by CHX. Whole-cell extracts of cells collected at the indicated time points were analyzed by SDS-PAGE and western blotting. Ubc6-HA was detected with an anti-HA antibody. Phosphoglycerate kinase (Pgk1) was used as a loading control. (B) Quantification of three experiments as in (A). Error bars represent mean ± one standard deviation. (C) Box chart quantification of the fraction of Ubc6-HA remaining 1 h after CHX addition representing mean ± one standard deviation from three independent experiments as in (A). * and ** denote *P* values < 0.1 and <0.01, respectively, derived from one-way ANOVA significance tests with Tukey's post hoc analysis. Individual *P* values are: 0.0000374 (C87A, WT), 0.00898 (S89A, WT), 0.000123 (H94A, WT), 0.00212 (H94Q, WT), and 0.099 (T121A, WT). Source data are available online for this figure.

version of Sbh2, in which all but one lysines in its cytoplasmic part were mutated to arginine (Sbh2 4KR), is degraded in a strictly Ubc6-dependent manner, whereas WT Sbh2 is degraded also in the absence of Ubc6, dependent on Ubc7/Cue1 (Lips et al, 2020; Weber et al, 2016). We reconstituted this system with purified full-length proteins, using Sbh2 N-terminally modified with an N-acetylated fluorescent peptide, thus easing detection in SDS-PAGE and excluding ubiquitination of the amino terminus (Appendix Fig. S4). When we incubated proteoliposomes containing Ubc6, Sbh2 and Doa10 with E1, ubiquitin and ATP, Sbh2 was robustly ubiquitinated (Fig. 4D,E). When we replaced WT Sbh2 with Sbh2 4KR, ubiquitination was less efficient, consistent with the idea that some residues that can be ubiquitinated in the WT protein were now absent (Figs. 4E and EV3B,C). An Sbh2 mutant, in which the same lysine residues were replaced by serine (Sbh2 4KS), was ubiquitinated with efficiency and kinetics identical to WT Sbh2 (Fig. 4D,E). When we omitted Ubc6 from liposomes and instead added Cue1/Ubc7, WT Sbh2 was polyubiquitinated, whereas the Sbh2 4KS and 4KR mutants remained largely unmodified (Fig. 4D for WT and 4KS, Fig. EV3B for 4KR, quantification in Fig. 4F). Presence of both, Ubc6 and Cue1/Ubc7 restored polyubiquitination of the 4KR and 4KS variants (Fig. 4D (4KS), Fig. EV3B (4KR)). These results show that Ubc6 mediates ubiquitination of Sbh2 variants largely lacking lysines, whereas Ubc7 requires the presence of a lysine residue, either in Sbh2 or in an already attached

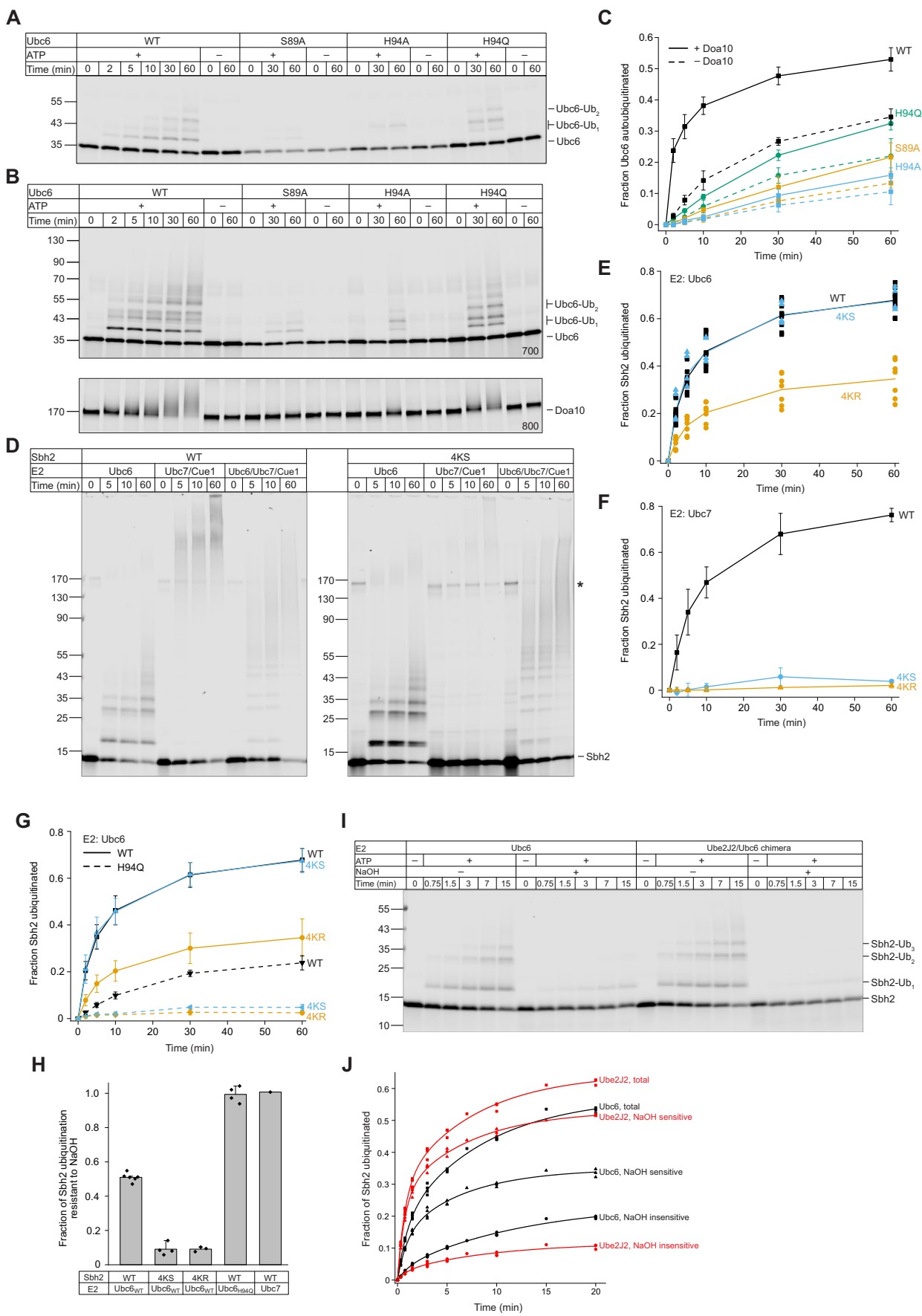

**Figure 4. His94 in Ubc6 is required for reactivity toward the hydroxy group.**

(A) Time course of Ubc6 autoubiquitination in the absence of Doa10. Fluorescently labeled wild-type (WT) Ubc6 or its indicated variants were reconstituted into liposomes and incubated with E1, ubiquitin, and ATP. Where indicated, ATP was omitted. Samples were analyzed by reducing SDS-PAGE and fluorescence scanning. The concentration of Ubc6 was identical for variants; the varying signal strength is due to different labeling efficiencies. (B) Time course of Ubc6 autoubiquitination as in (A), but with co-reconstituted Doa10 (Top). In the same samples, autoubiquitination of fluorescently labeled Doa10 was analyzed (Bottom). Ubc6 and Doa10 were detected in separate fluorescence channels at 700 and 800 nm, respectively. (C) Quantification of the autoubiquitinated fraction of the indicated Ubc6 variants from experiments as in (A, B). Data points and error bars indicate mean ± one standard deviation from the mean (SDM) from three experiments. Solid and dashed lines connect means for experiments with and without Doa10, respectively. $N = 3$ for reactions without Doa10, $N = 4$ for reactions with Doa10. (D) Time course of ubiquitination of WT Sbh2 and its mutant Sbh2 4KS. Fluorescently labeled Sbh2 variants were co-reconstituted with Doa10. In addition, either Ubc6, Ubc7/Cue1, or all three proteins were co-reconstituted. Samples were analyzed by SDS-PAGE and fluorescence scanning. The asterisk marks a band that appears upon co-reconstitution of Doa10 with fluorescently labeled Sbh2. It probably reflects a partially SDS-resistant complex of the two proteins. (E) Quantification of Sbh2, Sbh2 4KS and 4KR ubiquitination in the presence of Doa10 and Ubc6 from at least four experiments as in (D) and Fig. EV3B. Solid lines connect means. $N = 10$ for WT Sbh2, $N = 7$ for Sbh2 4KR, $N = 4$ for Sbh2 4KS. (F) Quantification of Sbh2, Sbh2 4KS, and 4KR ubiquitination in the presence of Doa10 and Ubc7/Cue1 from at least four different experiments as in (D) and Fig. EV3B. Data points and error bars indicate mean ± SDM. $N = 7$ for WT Sbh2, $N = 4$ for Sbh2 4KR, $N = 3$ for Sbh2 4KS. (G) Quantification of Sbh2, Sbh2 4KS, and 4KR ubiquitination in the presence of Doa10 and Ubc6 H94Q. Data points, connected by dashed lines, and error bars indicate mean ± SDM. $N = 7$ for WT Sbh2, $N = 3$ for Sbh2 4KR, $N = 2$ for Sbh2 4KS. For comparison, data from (E) for WT Ubc6 is reproduced here, with data points connected with solid lines. (H) Bar plots showing the fraction of Sbh2 ubiquitination resistant to sodium hydroxide (NaOH) treatment, as an indication for lysine ubiquitination. Individual experiments are shown as dots. Error bars indicate SDM. Samples were collected after 1 h from reactions as in Fig. EV3D,E. For samples containing WT Ubc6, $N = 6$ for WT Sbh2, $N = 4$ for Sbh2 4KS, $N = 3$ for Sbh2 4KR. For WT Sbh2 in combination with Ubc6 H94, $N = 4$. $N = 1$ for WT Sbh2 with Ubc7. (I) Time course of the emergence of NaOH-resistant and -sensitive Sbh2 ubiquitinations in reactions with co-reconstituted Doa10 and either WT Ubc6 or a Ube2J2/Ubc6 chimera. Where indicated, samples were treated with NaOH to preserve only lysine modifications. Reactions lacking ATP serve as controls. Samples were analyzed by SDS-PAGE followed by fluorescence scanning. (J) Quantification of experiments as in (I). The NaOH-sensitive fraction represents the difference between the total and NaOH-resistant fraction. For visualization, double-exponential fits to the data are shown as solid lines. $N = 4$ for Ubc6, $N = 3$ for Ube2J2/Ubc6 chimera. Source data are available online for this figure.

ubiquitin. Thus, our reconstituted system recapitulates the main characteristics of the ubiquitination pathway observed in vivo.

Replacing WT Ubc6 with S89A or H94A mutants abolished ubiquitination of both WT and 4KR Sbh2 (Fig. EV3C), whereas the H94Q mutant retained some ubiquitination of WT, but not of Sbh2 4KR or 4KS (Fig. 4G), suggesting that His94 is important for Ser/Thr modification. To test this notion directly, we incubated samples of the ubiquitination reaction with sodium hydroxide (NaOH) prior to SDS-PAGE analysis. This treatment hydrolyzes ubiquitin oxyesters, but leaves isopeptide linkages intact (Ferri-Blazquez et al, 2023). As expected, polyubiquitination mediated by Ubc7 was NaOH-resistant indicating that only lysines were modified (Figs. 4H and EV3D). In reactions with Ubc6 as the sole E2, $51 \pm 3\%$ of ubiquitin linkages to WT Sbh2 after 1 h were resistant to NaOH treatment, compared to only $9.1 \pm 1.3\%$ and $9.1 \pm 3.5\%$ for Sbh2 4KR and 4KS, respectively (Figs. 4H and EV3E). Ubiquitination of WT Sbh2 mediated by Ubc6 H94Q mutant was insensitive to NaOH treatment, showing that this mutant only creates isopeptide linkages (Figs. 4H and EV3E).

Next, to assess conservation of the preference for hydroxy group over lysine modification in the J2 family, we conducted the Sbh2 ubiquitination assay with a chimeric version of Ube2J2 and Ubc6, in which we exchanged only the UBC domain of Ubc6 with that of Ube2J2. This J2-Ubc6 chimera mediated efficient Sbh2 ubiquitination (Fig. 4I). Notably, based on the resistance of ubiquitinated species to alkaline treatment, the UBC domain of Ube2J2 was more selective for serine/threonine than that of Ubc6 (Fig. 4I). For both, Ubc6 and the chimeric Ube2J2, preference for Ser/Thr is also reflected in the considerably faster build-up of oxyester linkages (Fig. 4J). Furthermore, mutation of His101 to Gln in Ube2J2, equivalent to H94Q in Ubc6, led to a drastic reduction in Sbh2 ubiquitination (Fig. EV3F). Together, these experiments show that J2 family E2s preferentially mediate ubiquitination of hydroxy group containing amino acids (Ser or Thr), but can also modify lysines. Consistent with the idea that the conserved His acts as a base on hydroxy group containing nucleophiles, the H94Q/H101Q

mutants only maintain some reactivity towards lysine whereas the capacity for Ser/Thr ubiquitination is largely lost.

## Intrinsic reactivity of Ubc6-Ub conjugates for nucleophilic attack

In experiments with Sbh2, we cannot distinguish if preferential ubiquitination of Ser/Thr is due to the higher frequency of these amino acids in the substrate, a more suitable presentation of these residues by Doa10, or an intrinsic chemoselectivity of Ubc6 for Ser/ Thr. To probe for chemoselectivity more specifically, we next employed ubiquitin discharge assays, first in the absence of an E3. In this assay, the E1 first loads a soluble fragment of Ubc6 comprising only the UBC domain with ubiquitin, followed by dilution of the reaction into buffer containing EDTA to quench E1 activity (Fig. 5A). The loaded state hydrolyzed with a half-life of $46 \pm 5$ min, corresponding to a pseudo-first-order rate constant of $0.015 \pm 0.002$ min$^{-1}$, with minimal autoubiquitination for this construct (Fig. 5B,C). To probe for reactivity toward hydroxyl groups, we performed the same reactions in the presence of increasing amounts of glycerol. This led to faster ubiquitin discharge, consistent with a second-order rate constant for discharge onto glycerol of $k_{2,glycerol} = 3.8 \pm 1.1 \cdot 10^{-4}$ mmol$^{-1}$ min$^{-1}$. We obtained similar rate constants when we used a fluorescence anisotropy assay with labeled ubiquitin (Appendix Fig. S5A), and confirmed transfer of ubiquitin to glycerol by mass spectrometry (Fig. EV4A,B).

We then compared ubiquitin transfer to different free amino acids (Fig. 5D,E). L-serine and L-lysine accelerated discharge to similar degrees, whereas Ubc6-Ub was essentially inert to L-alanine or L-threonine. This shows that, similar to canonical E2s (Pickart and Rose, 1985), the secondary α-amino group cannot nucleophilically attack the Ubc6-Ub thioester, and suggests a selectivity of Ubc6 for primary alcohols. Experiments with an array of primary, secondary and tertiary alcohols confirmed this notion (Appendix Fig. S5B). Ubc6-Ub readily reacted with the strong nucleophile

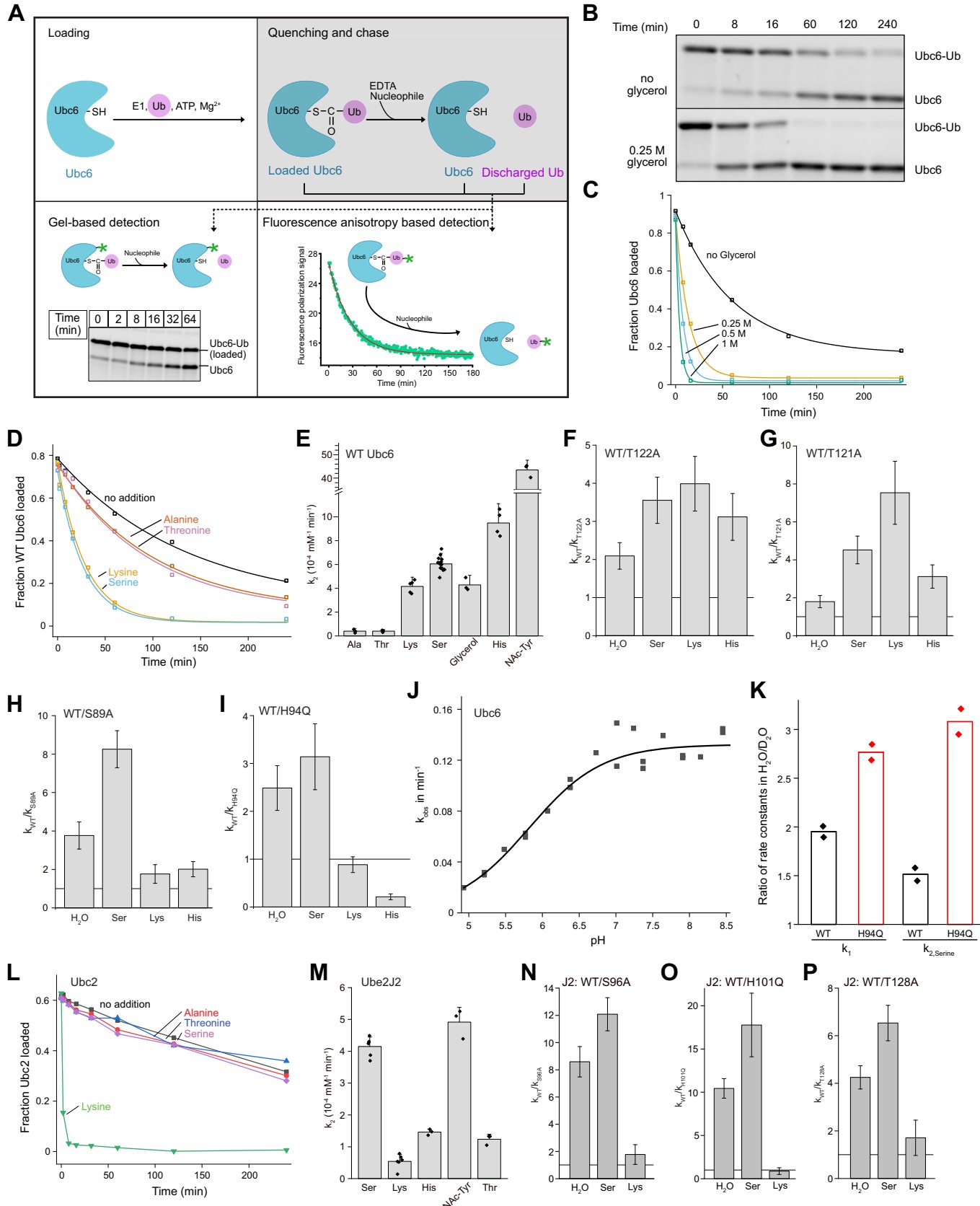

Figure 5. Intrinsic reactivity profiling of Ubc6 by discharge assays.

(A) Scheme for nucleophile discharge assay. Loading: E1, the UBC domain of Ubc6, and ubiquitin (Ub) are incubated with ATP in the presence of magnesium ions to generate the Ubc6-Ub thioester-linked conjugate. Quenching and chase: EDTA inactivates E1 by chelating magnesium ions, thereby preventing E1-mediated reloading of E2 discharged by hydrolysis or another nucleophile. Reaction rates are determined either by densitometry analysis of SDS-PAGE or by fluorescence anisotropy using fluorescently labeled ubiquitin. (B) Ubc6 discharge assay as described in (A) in the absence or presence of 250 mM glycerol during the chase step. Samples taken at the indicated time points of the chase reaction were analyzed by non-reducing SDS-PAGE and fluorescence scanning. (C) Quantification of discharge rates from experiments as in (B). The fraction of the loaded and discharged state were quantified by densitometry. The data were then globally fitted to determine rate constants for hydrolysis ($k_1$) and discharge by glycerol ($k_{2,glycerol}$), accounting for a minor fraction of autoubiquitination. Solid lines show the result of the fitting procedure. (D) As in (C), but without and with 100 mM of the indicated free amino acids. (E) Bar plots showing second-order rate constants ($k_2$) for discharge by the indicated free amino acids or glycerol. Individual data points represent fitting results, each derived from an experiment as in (D). Means and 1.5 standard deviations from the mean were determined from at least three experiments (see Source Data for exact N numbers). (F–I) Bar plots comparing reactivity of the indicated Ubc6 mutants towards the indicated nucleophiles with that of WT Ubc6. Discharge by hydrolysis is indicated as "$H_2O$". Rate constants of mutants ($k_{Mut}$) for different nucleophiles were determined as in (D) and averages determined as described for (E). Plotted are fold differences of discharge rates $\frac{k_{WT}}{k_{Mut}}$; (F) T122A, (G) T121A, (H) S89A, and (I) H94Q. Average rate constants for each mutant and nucleophile were determined from at least three experiments (see Source Data for exact N numbers). Errors ($\partial$) were calculated from standard deviations of average rate constants ($\partial k$) for each mutant and nucleophile according to $\partial\left(\frac{k_{WT}}{k_{Mut}}\right) = \frac{k_{WT}}{k_{Mut}}\sqrt{\left(\frac{\partial k_{WT}}{k_{WT}}\right)^2 + \left(\frac{\partial k_{Mut}}{k_{Mut}}\right)^2}$. (J) pH profile of the discharge rate form Ubc6 in the presence of 100 mM glycerol determined by fluorescence anisotropy. WT UBC domain of Ubc6 was loaded with fluorescently labeled ubiquitin at pH 7.4, followed by dilution into buffers of the indicated pH values with 100 mM glycerol. Discharge rates were determined by fitting fluorescence anisotropy traces with a mono-exponential function, resulting in the plotted $k_{obs}$ values for each anisotropy trace. The solid line is derived from a dose-response fit. (K) Bar plots showing a kinetic solvent isotope effect on hydrolysis ($k_1$) and serine-mediated ubiquitin discharge ($k_{2,serine}$) from either WT Ubc6 or its H94Q mutant. Discharge rates were determined as in (D, E) in either deuterated or non-deuterated solvent. Individual data points represent rates determined from experiments with $H_2O$ or $D_2O$ performed in parallel, followed by calculation of the ratio of the two rate constants. (L) As in (D), but with yeast Ubc2, comparing discharge without and with 100 mM of the indicated free amino acids. Values for each time point are shown as colored symbols connected by solid lines. (M) As in (E), but with the UBC domain of human Ube2J2. A representative example of the data included here and the result of the fitting procedure is shown in Fig. EV4L. $N = 6$ for lysine and serine, $N = 3$ for histidine, NAc-Tyr, and threonine. (N–P) Bar plots comparing reactivity of the indicated Ube2J2 mutants towards the indicated nucleophiles with that of WT Ube2J2. Discharge by hydrolysis is indicated as "$H_2O$". Average rate constants for each mutant and nucleophile were determined from at least three experiments as in (E) (see Source Data for exact N numbers). Plotted are fold differences of WT/mutant discharge rates; (N) S96A, (O) H101Q, (P) T128A. These ratios with associated errors were determined as described for (F–I). Source data are available online for this figure.

L-cysteine (Fig. EV4C). Somewhat unexpectedly, also L-histidine, N-acetyl-L-histidine, and N-acetyl-L-tyrosine (NAc-Tyr) accelerated ubiquitin discharge (Figs. 5E and EV4D–F). Two experiments confirmed that L-histidine is indeed a ubiquitin acceptor and does not merely facilitate hydrolysis by acting as a base. First, mass spectrometry showed the formation of ubiquitin–histidine adducts (Fig. EV4G). Second, in continuous turnover experiments, in which the E1 can reload ubiquitin discharged by hydrolysis onto Ubc6 but ubiquitin discharged by a different nucleophile becomes inert, we observed loss of Ubc6 reloading over time in the presence of histidine or glycerol. The deubiquitinating enzyme Usp2 reversed this inactivation, indicating that this enzyme can also hydrolyze ubiquitin–histidine and ubiquitin-glycerol adducts (Appendix Fig. S5C).

## Mutations in Ubc6 signature motifs have divergent effects on different nucleophiles

Next, we investigated how mutation of Ubc6 signature residues affect its intrinsic reactivity. First, we tested residues in the Thr-flap (T121, T122, and E119). In line with the notion that Thr122 facilitates nucleophilic attack by increasing electrophilicity of the thioester carbon, the T122A mutant exhibited reduced reactivity towards L-serine, L-lysine, and L-histidine to similar extent (Fig. 5F). Likewise, mutation of Thr121, which in MD simulations contacts either Gly75$_{Ub}$ or the thioester carbonyl, reduced reactivity to all nucleophiles (Fig. 5G). We then examined the role of Glu119, which forms a salt bridge with the arginine patch in ubiquitin. The E119A affected discharge mildly, but interestingly showed reduced reactivity to L-serine but increased reactivity to L-lysine and L-histidine (Fig. EV4H). The E119D mutation exhibited similarly reduced reactivity to all of these nucleophiles (Fig. EV4I). To

understand these effects better, we performed MD simulations with these mutants: The E119A mutation not only led to exposure of the cavity to bulk solvent but also destabilized the link between the ubiquitin C-terminal tail and the Thr-flap; mutation E119D resulted in less conformational flexibility of the ubiquitin tail (Fig. EV4J,K). These observations underline the importance of the positioning of the Thr-flap relative to the ubiquitin tail for reactivity and chemoselectivity towards serine.

In line with the strong defect observed for the S89A mutation in liposome assays (Fig. 4), this mutant showed a reactivity defect towards all amino acids, that was most pronounced for serine (Fig. 5H). In agreement with the results on Sbh2 ubiquitination, the H94Q mutation impaired hydrolysis and reactivity to L-serine and glycerol, but not to L-lysine (Fig. 5I; Appendix Fig. S5D). The H94N mutant was much less susceptible to hydrolysis, inert to L-serine, and maintained reactivity for L-lysine and L-histidine (Appendix Fig. S5E). Strikingly, the H94Q mutation increased reactivity to L-histidine, NAc-Tyr, and L-cysteine drastically, and even conferred reactivity towards L-glutamate (Fig. 5I; Appendix Fig. S5F–H).

Together, these data emphasize the central role His94 plays for reactivity towards serine. For His94 to catalyze nucleophilic attack by base catalysis, it has to be present in one of its two neutral tautomeric states, so that acidic pH should inhibit hydroxy group ubiquitination. We tested this by performing discharge assays in the presence of glycerol at different pH values. Discharge rates onto glycerol increased with increasing pH, but leveled off around pH 7.0 (Fig. 5J). While this pH profile reflects changes in the protonation state of multiple groups in Ubc6 and ubiquitin, it is consistent with a model in which His94 acts as a base during catalysis. Furthermore, when we measured ubiquitin transfer onto L-serine in either $H_2O$ or $D_2O$, we observed a significantly slower reaction in $D_2O$. This solvent isotope effect was stronger when we

used the Ubc6 H94Q mutant instead of WT Ubc6 (Fig. 5K), indicating that at least one proton transfer reaction contributes to the overall rate of the discharge reaction and that His94 catalyzes at least one of these proton transfer reactions.

## Comparison to Ube2J2 and a canonical E2

For comparison, we measured intrinsic reactivity of a canonical E2, yeast Rad6/Ubc2 towards free amino acids. Compared to Ubc6-Ub, Ubc2-Ub was more resistant to hydrolysis, essentially inert to L-serine, L-threonine and L-alanine, and by more than an order of magnitude more reactive to L-lysine (Fig. 5L). Given the importance of His94 in Ubc6 for Ser/Thr ubiquitination of Sbh2, we also tested if introducing histidine at a structurally roughly equivalent position in Ubc2 (Q93H) conferred reactivity towards L-serine. However, this was not the case (Appendix Fig. S5I), suggesting that additional structural elements are necessary for such a reactivity.

To assess how conserved the contribution of signature residues to Ubc6's intrinsic reactivity is, we also measured discharge from Ub-loaded human Ube2J2 (J2-Ub). While the active site loop is strongly conserved between the two species, the Thr-flap shows more variation, with only Thr121 being conserved in Ube2J2, but Glu119 and Thr122 not (Figs. EV4M and 1D). Like Ubc6-Ub, J2-Ub was susceptible to hydrolysis, and more reactive to L-serine than to L-threonine, in agreement with a previous report (Abdul Rehman et al, 2024). Importantly, compared to Ubc6, J2-Ub showed reduced reactivity to NAc-Tyr and L-histidine, and was essentially inert to L-lysine (Figs. 5M and EV4L). The Ube2J2 mutants S96A, H101Q, and T128A (corresponding to Ubc6 S89A, H94Q, and T121A, respectively) showed strongly reduced reactivity to L-serine and were less susceptible to hydrolysis, whereas the low reactivity of J2-Ub toward lysine was hardly affected by these mutations (Fig. 5N–P). Together, these experiments show that that intrinsic reactivity towards free serine is a conserved property of the J2 family. The low intrinsic reactivity of J2-Ub towards weak nucleophiles like histidine or tyrosine correlates with variations in the Thr-flap, which probably results in less basal activation of its E2-Ub thioester. Furthermore, the divergent effects of the H94Q and E119A mutations in Ubc6 toward different free amino acids—L-serine on the one hand, L-lysine, L-histidine, and NAc-Tyr on the other—suggest that nucleophilic attack by these compounds might occur by different pathways.

## RING E3 binding sharpens substrate profile towards OH-groups

While Ubc6-Ub discharge experiments in the absence of an E3 showed no preference for L-serine over L-lysine and even reactivity towards L-histidine and L-tyrosine, experiments on Sbh2 ubiquitination with co-reconstituted Doa10 showed selective ubiquitination of Ser/Thr over Lys residues. This suggests that the Doa10 interaction enhances chemoselectivity towards serine or that Doa10 presents serines of this particular substrate more efficiently for ubiquitination. To distinguish between these two possibilities, we performed discharge assays in the presence of a soluble Doa10 RING construct. Recent cryo-EM structures of Doa10 (Botsch et al, 2024; Wu et al, 2024), functional data on Doa10 and its human homolog MarchF6 (Mehrtash and Hochstrasser, 2022; Nguyen et al, 2022; Zattas et al, 2016), and

Alphafold2 predictions that we performed of other RING-CH containing E3 ligases (Appendix Fig. S6A,B) suggested that a conserved C-terminal element (CTE) of Doa10/March6 is important for full functionality of its RING-CH domain. Comparison of Sbh2 ubiquitination by wt Doa10 and a CTE mutant version (G1308L, N1314A, called 2CTM (Mehrtash and Hochstrasser, 2022; Zattas et al, 2016)) in liposomes supported this notion (Fig. EV5A,B). We therefore generated a soluble Doa10 RING construct, in which we fused the RING-CH domain to the CTE via a short linker (called RING-CTE) and compared its activity to that of the RING domain alone. To exclude that quenching of E1 activity by EDTA inhibits the E3, we used a purified Ub-loaded Ubc6 for this experiment. Indeed, the RING-CTE construct accelerated hydrolysis of Ubc6-Ub more strongly than the RING domain alone (Fig. 6A–C). EDTA addition only marginally affected Doa10 RING activity. As slow autoubiquitination of Ub-loaded Ubc6 during the purification procedure complicated kinetic analysis, we performed further experiments using the protocol described in Fig. 5A, i.e., with acute loading and EDTA-quenching of E1.

In experiments with free amino acids, RING-CTE increased reactivity to L-serine more strongly than to L-lysine, whereas it only mildly accelerated discharge by NAc-Tyr, and had no effect on discharge by L-histidine (Figs. 6D and EV5C; Appendix Fig. S6C). Also in the presence of RING-CTE, Ubc6 exhibited selectivity for L-serine over L-threonine. However, compared to the virtual lack of reactivity towards L-threonine in the absence of an E3, RING-CTE stimulated discharge to L-threonine by a similar factor as to L-serine (Appendix Fig. S6D,E). RING-mediated enhancement of Ubc6 chemoselectivity was not specific for Doa10, as also the Hrd1 RING domain stimulated discharge by L-serine more than discharge by L-lysine (Appendix Fig. S6F,G).

Stimulation of reactivity towards L-serine strongly depended on the previously identified signature residues in Ubc6, as demonstrated by the total lack of stimulation for the S89A, and severe impairment for H94Q, T121A, T122A, E119A, and E119D mutants of Ubc6 (Fig. 6E). We then tested how the S89A and H94Q mutations affect reactivity towards L-lysine in the presence of RING. This was prompted by the observation that in the absence of RING, reactivity towards L-lysine was unchanged by the H94Q (Fig. 5K). In contrast, the H94Q mutation severely impaired stimulation of discharge onto L-lysine by RING-CTE (Fig. 6F), indicating that in the RING-bound state, the reaction with lysine also involves His94. As the E119A mutation had shown an increased reactivity towards L-lysine compared to WT Ubc6, we also tested how this mutation affects lysine modification in reactions with Sbh2. In this context, the E119A mutation similarly impaired both Lys and hydroxy modifications on Sbh2 (Appendix Fig. S6H), indicating that the reaction with free L-lysine might occur via a different pathway than ubiquitin attachment to lysine in a polypeptide.

Next, we tested how the presence of an E3 affects Ube2J2 reactivity. A RING-CTE version of MarchF6 accelerated hydrolysis and discharge by L-serine (Fig. 6G,H). As with the Ubc6/Doa10 pair, this was strictly dependent on residues Ser96 and His101, and greatly diminished when T128 in the Thr-flap was mutated (Fig. 6H). In the presence of L-lysine, NAc-Tyr, and L-histidine, discharge rates were consistently lower than in the absence of amino acids. This indicated that these compounds interfered with

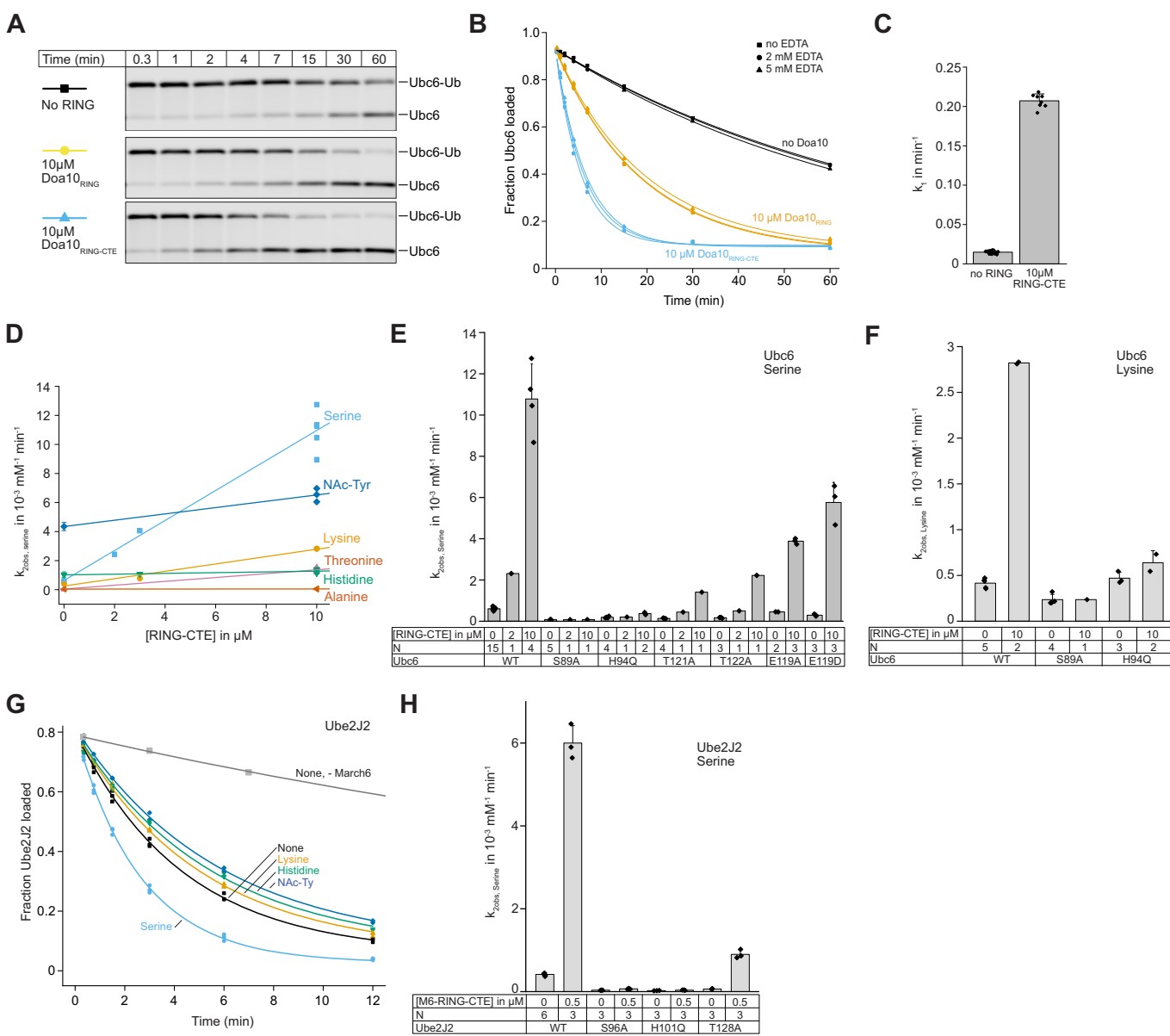

**Figure 6. RING binding enhances intrinsic Ubc6 reactivity toward hydroxy groups.**

(A) Ubc6 ubiquitin discharge assay performed by directly diluting purified fluorescently labeled Ubc6-Ub in chase buffer without or with 10 μM of the indicated Doa10 RING domain construct, in the absence of EDTA. Samples taken at the indicated time points were analyzed by non-reducing SDS-PAGE and fluorescence scanning. (B) Quantification of reactions as in (A). In addition, reactions are shown that contained 2 or 5 mM EDTA. Solid lines represent fits of the data. (C) Bar plots showing observed rates of hydrolysis ($k_1$) of Ubc6-Ub in the presence or absence of 10 μM Doa10 RING-CTE, derived from exponential fits of plots as in (B) from 8 and 25 experiments, respectively. Bars show the mean, error bars one standard deviation. (D) Scatter plot showing observed rates of discharge by the indicated free amino acids plotted against the RING-CTE concentration. For reactions containing RING-CTE, each data point represents the apparent rate constant determined by global fitting of a pair of reactions in the absence or presence of the indicated nucleophile. For reaction in the absence of RING-CTE, values determined in experiments for Fig. 5 are reproduced, with error bars indicting the standard deviation from the mean. Solid lines represent linear fits for each nucleophile. Examples of data used here are shown in Fig. EV5C and Appendix Fig. S6D. (E) Bar plots showing reactivity of the indicated Ubc6 variants to free serine at the indicated RING-CTE concentrations. Observed second-order rate constants for discharge by serine ($k_{2obs,serine}$) were determined as described for (D). Error bars indicate one standard deviation from the mean. The number of replicates is indicated in the table ($N$). (F) As in (E), but with free lysine as a nucleophile ($k_{2obs,lysine}$). (G) Ubiquitin discharge assays performed with 2 μM ubiquitin-loaded Ube2J2 in the presence of 50 mM of the indicated nucleophiles and 0.5 μM of a MarchF6 RING-CTE. $N = 3$ for reactions in the absence of additional nucleophile, lysine, and serine, $N = 2$ for reactions with histidine and NAc-Tyr. Solid lines represent global exponential fits to all data derived from each nucleophile. For comparison, a hydrolysis reaction is shown in the absence of MarchF6. (H) As in (E), but showing reactivity of the indicated Ube2J2 variants to free serine at the indicated concentrations of MarchF6 RING-CTE (M6-RING-CTE). Source data are available online for this figure.

Ube2J2 binding to M6-RING-CTE and thus precluded further conclusions on an impact of RING binding on the reactivity profile of Ube2J2 (Appendix Fig. S6I).

Overall, these results show that RING binding enhances the chemoselectivity of Ubc6 for Ser over Lys, explaining the observed preference for hydroxy group ubiquitination over Lys ubiquitination in Sbh2. In a final set of experiments, we sought further insight into whether the intrinsic reactivity of Ubc6 toward free L-histidine and NAc-Tyr also leads to the ubiquitination of these residues in a polypeptide context. By mass spectrometry, we did not detect such modifications, possibly due to their low chemical stability during LC-MS analysis. Therefore, we used Ubc6 autoubiquitination as a model. In this case, ubiquitination primarily occurs on Ser196 (Weber et al, 2016), which we confirmed by mass spectrometry (Appendix Fig. S6J). Mutation of Ser196 to alanine strongly reduced autoubiquitination (Fig. EV5D). A mutagenesis screen, substituting Ser196 with either Thr, Lys, His (Fig. EV5D,E), or Tyr (Fig. EV5F,G) showed that only the threonine substitution restored some autoubiquitination. Substitution with Lys, His, or Tyr yielded no ubiquitination beyond the S196A background. However, this result does not exclude that Ubc6 mediates His or Tyr ubiquitination. Rather, the lack of lysine modification here, despite clear evidence that Ubc6 can mediate such a modification in Sbh2, highlights that the polypeptide context plays a major role in whether a residue be ubiquitinated. The same may apply to His and Tyr ubiquitination.

# Discussion

The ubiquitination system is more versatile than previously understood. Landmark studies on noncanonical ubiquitination of proteins and of non-protein molecules—such as lipids, sugars, nucleic acids and potentially other metabolites—revealed an expanded substrate range and physiological potential of the ubiquitination system (Abdul Rehman et al, 2024; preprint: Ahel et al, 2021; Ahel et al, 2020; preprint: Dearlove et al, 2024; Kelsall et al, 2022; Otten et al, 2021; Sakamaki and Mizushima, 2023; preprint: Zhu et al, 2024; Zhu et al, 2022). However, our mechanistic understanding of these phenomena is limited. Here, we have shown how specific molecular features of the J2 family of ubiquitin-conjugating enzymes expand the chemical spectrum of ubiquitination beyond lysine residues.

Ubiquitination reactions exhibit high chemoselectivity, typically targeting the primary ε-amino group of lysine. Canonical E2s employ two main catalytic strategies to achieve reactivity towards amino groups: hydrogen bonding involving the HPN motif stabilizes the high-energy anionic tetrahedral intermediate (Wu et al, 2003; Yunus and Lima, 2006), and the CES/D site suppresses the $pK_a$ of the incoming amino group, increasing its nucleophilicity (Valimberti et al, 2015). We found that the J2 family employs different strategies. Instead of the HPN motif, residues in the Thr-flap contact the thioester and probably also the tetrahedral intermediate. This rewiring of the active site increases the electrophilicity of the thioester carbonyl carbon, enhancing susceptibility to nucleophilic attack from weaker nucleophiles such as aliphatic and phenolic hydroxy groups, or the imidazole nitrogen of histidine. In addition, Ubc6 and Ube2J2 use a conserved histidine to catalyze hydroxy group nucleophilic attack by general

base catalysis. Ubiquitination reactions that recapitulate a physiological reaction show that His94/His101 is crucial for the formation of oxyester ubiquitin linkages. The pH profile of ubiquitin discharge reactions, a solvent isotope effect, and MD simulations showing that His94 can adopt compatible conformations, support a general base catalysis mechanism. Conservative mutations of His94 to Gln or Asn, which partially retain the hydrogen bonding capacity of histidine, but result in a specific loss of reactivity toward serine, argue against a model in which His94 would merely stabilize specific reactive conformations. However, there is no evidence for a catalytic triad, as nearby carboxylates (Asp92 and Asp96) are too distant to hydrogen bond with His94 without major conformational changes upon RING binding.

We show that Ubc6-Ub, and to a lesser degree J2-Ub, are reactive towards the imidazole ring of free histidine and the phenolic hydroxy group of free tyrosine. However, while assays with small nucleophiles were useful to assess the intrinsic reactivity of Ubc6-Ub and its determinants, several observations suggest that these residues are unlikely to be modified in polypeptides (Lechtenberg and Komander, 2024). RING binding only slightly enhances reactivity toward Tyr and not at all toward His. Furthermore, mutations like H94Q or E119A, which reduce reactivity toward both free L-serine and polypeptide serine side chains, increase reactivity towards free histidine and tyrosine, suggesting different mechanisms of attack for these nucleophiles. It is conceivable that in the RING-free state, small nucleophiles can access the water-filled cavity to attack the thioester from this direction, but RING binding may close off this cavity. Ultimately, evidence by mass spectrometry is needed to settle this question. We provide here a protocol for the quantitative synthesis of ubiquitin–histidine and -tyrosine adducts, which can aid in developing methods to preserve phenolic esters and imidazolides of ubiquitin during sample processing.

RING binding sharpens the intrinsic chemoselectivity of Ubc6 towards serine, whereas Ube2J2 already exhibits high intrinsic chemoselectivity for serine. Based on known structures of homologous E2-Ub-RING complexes, the RING-mediated enhancement of Ubc6/Ube2J2 activity likely occurs via an allosteric effect, though the transmission mechanism remains speculative. Our data indicate that RING binding not only stabilizes the closed conformations within the global Ubc6-Ub conformational ensemble, but also reactive conformations of the active site loop and the ubiquitin tail. Our X-ray structure of Ubc6-Ub, the structure of Ube2J2, and our MD simulations suggest conformational flexibility in the active site loop, leading to the displacement of the catalytically important histidine residue away from the active site cysteine. Stabilization of specific conformations of the ubiquitin tail is supported by the observation that the S89A mutation abolishes any RING-mediated reactivity gain, although it does not globally interfere with closed conformations and is unlikely to disrupt RING binding. RING binding may also favor productive conformations of His94 and exposure of the thioester group for nucleophilic attack. Together with recent single-molecule experiments on a canonical E2-RING pair (Branigan et al, 2020), these observations support the idea that for efficient ubiquitin transfer from E2-Ub, factors beyond the closed conformation must be in place. In the future, high-resolution structures of J2-Ub in complex with a RING domain, ideally combined with MD simulations, should provide further insight into this question.

Our phylogenetic analysis indicates that the split into the two J families occurred early in their evolution. While the GRF motif and the active site loop are conserved in both families, the region corresponding to the Thr-flap in J2 members shows significant divergence in J1 members. This implies structural differences and suggests that the mechanism of action in J1 E2s may be distinct, potentially showing a lower intrinsic reactivity toward weaker nucleophiles, such as hydroxy groups. Supporting this idea, the only report of noncanonical activity in Ube2J1, to our knowledge, found only a minor fraction of non-lysine modifications in a Ube2J1 substrate (Burr et al, 2013). Therefore, the ubiquitination mechanism of Ube2J1, particularly its role in noncanonical ubiquitination, warrants further investigation.

The presence of a critical histidine residue is a recurring theme found in ubiquitinating enzymes that modify hydroxy groups. The RCR E3 ligase MYCBP2 (Pao et al, 2018), the RBR E3 ligase HOIL-1 (Cohen et al, 2020; Kelsall et al, 2022), and the RING finger and AAA ATPase domain-containing RNF213 (preprint: Ahel et al, 2021; Ahel et al, 2020; Otten et al, 2021) mediate substrate ubiquitination through covalent E3-ubiquitin thioester intermediates. In each of these cases, mutation of a conserved histidine residue in the E3 ligase active site rendered the proteins inactive towards hydroxy group containing substrates. For RING-type DELTEX E3 ligases, the DTC domain contributes a conserved histidine to the active site of the thioester-bound E2-Ub (Zhu et al, 2022). More recently, a critical histidine residue was also identified in the noncanonical E2 Ube2Q (Abdul Rehman et al, 2024). In both of these cases, E3s catalyze the direct transfer of ubiquitin from the E2 onto a substrate hydroxy group; the proposed catalytic histidine is either part of an additional domain provided by the E3 (DELTEX) or resides in a C-terminal extension of the E2 UBC domain (Ube2Q). In the majority of these examples, a general base catalyst function has been attributed to the conserved histidine residue (Zhu et al, 2022). However, only in some cases have experimental structures of intermediates or reaction mimics corroborated this model (Mabbitt et al, 2020). Other plausible functions of histidine such as the stabilization of the high-energy tetrahedral intermediate should also be considered. Such a role for histidine has been proposed for the HPH motif that replaces the conventional HPN motif in some E2s, e.g., Ube2W (Burroughs et al, 2008; Michelle et al, 2009; Qi et al, 2015). Moving forward, additional experimental structures and computational models of reaction intermediates, combined with the reconstitution of relevant substrate ubiquitination reactions, are needed to further substantiate the catalytic role of conserved histidine residues.

We have identified the mechanism by which J2 E2s modify hydroxy groups; however, the evolutionary advantage of the expanded reactivity of the ubiquitination pathway in ERAD remains speculative. This study and others (Wang et al, 2009) highlight the ability of J2 family E2s to preferentially ubiquitinate serine over lysine residues. Expanding reactivity beyond lysine may increase the versatility of the protein quality control system in targeting a broader substrate spectrum. The ability to prime ubiquitin onto multiple nucleophilic site on a target may offer a kinetic advantage by increasing the likelihood of polyubiquitination upon substrate encounter. Additionally, employing a second E2 of the J1 or J2 family with distinct chemoselectivity provides a potential mechanism for the regulated degradation of substrates that are not typically degraded due to a lack of aptly positioned

lysine residue. This hypothesis suggest that activity of J family E2s would need to be regulated, as suggested for Ube2J1 (Elangovan et al, 2017; Oh et al, 2006), though, to our knowledge, such regulation has not been investigated for Ube2J2. At present, however, the prevalence of J1 and J2 family-mediated hydroxy group ubiquitination and its physiological relevance are largely unknown. Moreover, it needs to be investigated whether J family E2s also mediate the ubiquitination of non-protein molecules.

# Methods

**Reagents and tools table**

| Reagent/resource | Reference or source | Identifier or catalog number |
|---|---|---|
| **Experimental models** | | |
| BY4741 MATa his3Δ1 leu2Δ0 met15Δ0 ura3Δ0 | EUROSCARF | Y00000 |
| IncSc1 MATa his3D1 leu2 trp1-289 ura3-52 MAT his3D1 leu2 trp1-289 ura3-52 | Invitrogen | C81000 |
| DOA10 deletion MATa his3Δ1 leu2Δ0 met15Δ0 ura3Δ0 ssm4::KanR | Schmidt et al, 2020 | N/A |
| UBC6 deletion MATa his3Δ1 leu2Δ0 met15Δ0 ura3Δ0 ubc6::natMX6 | This study | N/A |
| UBC7 deletion MATα his3Δ1 leu2Δ0 lys2Δ0 ura3Δ0 ubc7::KanR | This study | N/A |
| *E. coli* NEB 10-beta | New England Biolabs | C3019I |
| *E. coli* BL21 -CodonPlus (DE3)-RIPL | Agilent | 230280 |
| *E. coli* BL21-Gold | Agilent | 230130 |
| *E. coli* BL21 (DE3) | New England Biolabs | C2527I |
| **Recombinant DNA** | | |
| Yeast MoClo System | Addgene Kit | #1000000061 |
| pET39b(+) | Merck | 70090 |
| pRS316 | ATCC | 77145 |
| pBAD His/C | Invitrogen | V43001 |
| K27 (used for all constructs with N-terminal His14-SUMO tag) | Stein et al, 2014 | Stein lab #6 |
| H14_SUMO_Ubc6_LPETGG in K27 | This study | Stein lab #343 |
| H14_SUMO_Ubc6_S89A_LPETGG in K27 | This study | Stein lab #938 |
| H14_SUMO_Ubc6_H94A_LPETGG in K27 | This study | Stein lab #949 |
| H14_SUMO_Ubc6_H94Q_LPETGG in K27 | This study | Stein lab #1293 |
| H14_SUMO_Ubc6_E119A_LPETGG in K27 | This study | Stein lab #1762 |
| H14_SUMO_Ubc6_1-172_LPETGG in pBAD His/C | This study | Stein lab #1173 |
| H14_SUMO_Ubc6_1-172_C87K_LPETGG in pBAD His/C | This study | Stein lab #1246 |
| H14_SUMO_Ubc6_1-172_S89A_LPETGG in pBAD His/C | This study | Stein lab #1310 |
| H14_SUMO_Ubc6_1-172_H94A_LPETGG in pBAD His/C | This study | Stein lab #1181 |
| H14_SUMO_Ubc6_1-172_H94Q_LPETGG in pBAD His/C | This study | Stein lab #1334 |
| H14_SUMO_Ubc6_1-172_H94N_LPETGG in pBAD His/C | This study | Stein lab #1328 |
| H14_SUMO_Ubc6_1-172_T121A_LPETGG in pBAD His/C | This study | Stein lab #1336 |

| Reagent/resource | Reference or source | Identifier or catalog number |
|---|---|---|
| H14_SUMO_Ubc6_1-172_T122A_LPETGG in pBAD His/C | This study | Stein lab #1335 |
| H14_SUMO_Ubc6_1-172_E119A_LPETGG in pBAD His/C | This study | Stein lab #1757 |
| H14_SUMO_Ubc6_1-172_E119D_LPETGG in pBAD His/C | This study | Stein lab #1758 |
| H14_SUMO_Ubc6_S196A_LPETGG in K27 | This study | Stein lab #1578 |
| H14_SUMO_Ubc6_S196T_LPETGG in K27 | This study | Stein lab #1591 |
| H14_SUMO_Ubc6_S196K_LPETGG in K27 | This study | Stein lab #1593 |
| H14_SUMO_Ubc6_S196H_LPETGG in K27 | This study | Stein lab #1592 |
| H14_SUMO_Ubc6_S196Y_LPETGG in K27 | This study | Stein lab #1801 |
| H14_SUMO_Rad6/Ubc2 in K27 | This study | Stein lab #1137 |
| H14_SUMO_Rad6/Ubc2_Q93H in K27 | This study | Stein lab #1577 |
| H14_SUMO_Ube2J2_1-179_LPETGG in K27 | This study | Stein lab #1803 |
| H14_SUMO_Ube2J2_1-179_S96A_LPETGG in K27 | This study | Stein lab #1850 |
| H14_SUMO_Ube2J2_1-179_H101Q_LPETGG in K27 | This study | Stein lab #1851 |
| H14_SUMO_Ube2J2_1-179_T128A_LPETGG in K27 | This study | Stein lab #1853 |
| H14_SUMO_Ube2J2_1-180_Ubc6_179-250_LPETGG in K27 | This study | Stein lab #1854 |
| H14_SUMO_Ube2J2_1-180 (H101Q)_Ubc6_179-250_LPETGG in K27 | This study | Stein lab #1856 |
| H14_SUMO_Doa10_1-129 (RING) in K27 | This study | Stein lab #847 |
| H14_SUMO_Doa10_RING-CTE In K27 | This study | Stein lab #1754 |
| H14_SUMO_MARCHF6_RING-CTE in K27 | This study | Stein lab #1755 |
| yeast Ubiquitin untagged in K27 | Vasic et al, 2020 | Stein lab #835 |
| H14_SUMO_Gly-Ubiquitin yeast in K27 | Stein et al, 2014 | Stein lab #187 |
| Ubiquitin G76C untagged in K27 | This study | Stein lab #1433 |
| His14-SUMO-Cue1-TEV-SBP38-lpetgg in pET39b(+) | Vasic et al, 2020 | Stein lab #672 |
| pET28a-LIC_human Usp2 | Addgene | #36894 |
| H14_SUMO_Hrd1_325-551_TEV_SBP in K27 | Stein et al, 2014 | Stein lab #65 |
| Doa10 (aa 1-468, Doa10-N) in pRS26-Pgal1 | Schmidt et al, 2020 | Stein lab #376 |
| Doa10 (aa 225-1319) TEV_SBP in pRS26-Pgal1 | Schmidt et al, 2020 | Stein lab #375 |
| SBP_SUMOEuB_CGG_Doa10 (aa 1-308, Doa10-N) | This study, generated by yeast MoClo system | Stein lab #1118 |
| Doa10 (aa 308-1319, Doa10-C) | This study, generated by yeast MoClo system | Stein lab #1119 |
| SBP_SUMOEuB_GGG_Doa10 2CTM (G1309L_N1314A) | This study, generated by yeast MoClo system | Stein lab #1177 |
| SBP_SUMOEuB_GGG_Sbh2 | This study, generated by yeast MoClo system | Stein lab #1159 |
| SBP_SUMOEuB_GGG_Sbh2_4KR | This study, generated by yeast MoClo system | Stein lab #1384 |
| SBP_SUMOEuB_CGG_Sbh2_4KS | This study, generated by yeast MoClo system | Stein lab #1744 |
| Ubc6_internal HA tag, Ubc6 promoter, pRS316 | This study, generated according to Sommer and Jentsch, 1993 | Stein lab #675 |

| Reagent/resource | Reference or source | Identifier or catalog number |
|---|---|---|
| Ubc6_internal HA tag, Ubc6 promoter, C87A pRS316 | This study | Stein lab #1528 |
| Ubc6_internal HA tag, Ubc6 promoter, S89A pRS316 | This study | Stein lab #942 |
| Ubc6_internal HA tag, Ubc6 promoter, H94A pRS316 | This study | Stein lab #946 |
| Ubc6_internal HA tag, Ubc6 promoter, H94Q pRS316 | This study | Stein lab #1576 |
| Ubc6_internal HA tag, Ubc6 promoter, T121A pRS316 | This study | Stein lab #1555 |
| **Antibodies** | | |
| Anti-SBP (clone 20), mouse monoclonal | Merck | RRID:AB_10631872 |
| Anti-HA (clone 3F10), rat monoclonal | Roche | RRID:AB_390919 |
| Anti-Pgk1 (clone 22C5D8), mouse monoclonal | Invitrogen | RRID:AB_2532235 |
| Anti-rat IgG secondary antibody (IRDye 800 CW), goat polyclonal | Li-Cor | RRID:AB_1850025 |
| Anti-mouse IgG secondary antibody (IRDye 680 RD), goat polyclonal | Li-Cor | RRID:AB_10956588 |
| **Chemicals, enzymes, and other reagents** | | |
| Gly-Gly-Gly-Cys peptide | Thermo Fisher Scientific | Custom synthesis |
| GDN detergent | Anatrace | GDN101 |
| n-Octyl β-D-glucopyranoside (OG) | Glycon Biochemicals | D97001 |
| n-Decyl β-D-Maltopyranoside (DM) | Glycon Biochemicals | D99003 |
| Dodecyl-β-D-maltoside (DDM) | Carl Roth | CN26.5 |
| 1-palmitoyl-2-oleoyl-glycero-3-phospho-choline (POPC) | Avanti Polar Lipids | 850457P |
| 1,2-dioleoyl-sn-glycero-3-phosphoethanolamine (DOPE) | Avanti Polar Lipids | 850725P |
| 1,2,-dioleoyl-sn-glycero-3-phospho-L-serine (DOPS) | Avanti Polar Lipids | 840035P |
| Ergosterol (>95%, HPLC) | Sigma-Aldrich | 45480 |
| ATP | PanReac AppliChem | A1348 |
| DyLight 680 maleimide | Thermo Fisher Scientific | 46618 |
| DyLight 800 maleimide | Thermo Fisher Scientific | 46621 |
| AlexaFluor488 maleimide | Thermo Fisher Scientific | A10254 |
| Pierce Detergent removal spin columns | Thermo Fisher Scientific | 87779 |
| HisPur Ni-NTA resin | Thermo Fisher Scientific | 88223 |
| Pierce High Capacity Streptavidin Agarose | Thermo Fisher Scientific | 20361 |
| Pierce Streptavidin Magnetic Beads | Thermo Fisher Scientific | 88817 |
| **Software** | | |
| ImageStudio Lite | Li-Cor, Free download | |
| Image Lab Software Version 6.0.1 | BioRad | Gel Doc EZ System |
| Pymol | http://www.pymol.org/pymol | |
| Origin 2018B PRO | | |
| AlphaFold (v2.0) | DeepMind | https://www.deepmind.com/open-source/alphafold |
| ColabFold (v1.5.5) | Ovchinnikov and Steinegger laboratories | https://github.com/sokrypton/ColabFold |
| Phenix Package | https://phenix-online.org/ | |
| GROMACS version 22.3 | https://manual.gromacs.org/2022.3/download.html | |

| Reagent/resource | Reference or source | Identifier or catalog number |
|---|---|---|
| MDAnalysis | https://www.mdanalysis.org/ | |
| **Other** | | |
| 4–20% Criterion™ TGX Stain-Free™ Protein Gel, 26 well, 15 µl | BioRad | 5678095 |

## S. cerevisiae and E. coli strains

For expression of full-length Doa10 in *S. cerevisiae*, a doa10 deletion strain derived from BY4741 (EUROSCARF) was generated (MATa his3Δ1 leu2Δ0 met15Δ0 ura3Δ0 ssm4::kanMX6). For expression of Sbh2 and its mutants, a ubc7 deletion strain derived from BY4742 was used (MATα his3Δ1 leu2Δ0 lys2Δ0 ura3Δ0 ubc7::kanMX4). For cycloheximide chase experiments, a ubc6 deletion strain derived from BY4741 was generated (MATa his3Δ1 leu2Δ0 met15Δ0 ura3Δ0 ubc6::natMX6). Uba1 was expressed in strain InvSc1 (Invitrogen). Full-length Ubc6, Ubc7, Cue1, Usp2, and soluble version of Doa10, March6 and Ube2J2 were expressed in the *E. coli* strain BL21-CodonPlus (DE3)-RIPL (Agilent). *S. cerevisiae* Hrd1 RING domain (aa 325–551) was expressed in *E. coli* BL21 DE3 cells (NEB). Soluble versions of Ubc6 were expressed in *E. coli* strains DH10B (NEB) or BL21-Gold (Agilent).

## Plasmids

The following constructs have been previously described: Uba1, Hrd1 RING, Ulp1, and Cys-ubiquitin (Stein et al, 2014), yeast ubiquitin (Vasic et al, 2020), and full-length Cue1 and Doa10 (Schmidt et al, 2020). The construct for the catalytic domain of human Usp2 in the pET28a vector was a gift from Cheryl Arrowsmith (Addgene plasmid # 36894).

For Ubc6, the *S. cerevisiae* S288C sequence was used. For purification purposes, full-length Ubc6 and all its mutant derivatives were expressed as N-terminal His14-SUMO and C-terminal sortase tag (LPETGG) fusions (Schmidt et al, 2020). For the soluble version of Ubc6, the UBC domain (aa 1-172) was inserted into pBAD/HisC vector (Invitrogen) together with an N-terminal His14- SUMO tag and a C-terminal sortase tag (LPETGG). For cycloheximide chase experiments, a construct with an internal HA tag (Sommer and Jentsch, 1993) was introduced into pRS316 (Sikorski and Hieter, 1989) together with 550 nucleotides 5' to the UBC6 start codon as a promoter. All mutations (S89A, H94A, H94N, H94Q, E119A, E119D, T121A, T122A, S196A, S196H, S196K, S196T, and S196Y) were introduced by standard mutagenesis methods.

A codon-optimized version was inserted into the K27 vector (Stein et al, 2014) for expression as an N-terminal His14-SUMO and C-terminal sortase tag (LPETGG) fusion.

For the RING domain of Doa10, the coding sequence from *S. cerevisiae* DOA10 for residues 1–129 was cloned into the K27 vector (Stein et al, 2014), which contains an N-terminal His14-SUMO tag. For the RING-CTE construct, the sequence for the N-terminal 103 residues was fused to the most C-terminal thirteen residues, separated by a five-residue linker, resulting in the C-terminal sequence TIY-GSGGQ-KGRALENLPDES. Similarly, for human MarchF6-CTE in the K27 vector, the same linker was introduced between residues 65

and 883, resulting in the C-terminal sequence PIY-GSGGQ-LVGQRLVNYERKSGKQGSSPPPPQSSQE. For the 2CTM mutant version of full-length Doa10 (Zattas et al, 2016), G1309L and N1314A mutations were introduced by site-directed mutagenesis.

For expression of Rad6/Ubc2, *S. cerevisiae* RAD6 (aa 1-172) was cloned into K27 (Stein et al, 2014).

To generate a construct for expression of Sbh2 and its 4KR/4KS variants (K15, K23, K25 and K28) a GoldenGate cloning system was used (Lee et al, 2015). Coding sequences from *S. cerevisiae* Sbh2 variants containing a TAA STOP codon were ordered as gene fragments (Integrated DNA Technologies, IDT) with appropriate restriction sites to generate type 3b plasmids from pYTK001. As an N-terminal affinity tag, coding sequences for streptavidin binding peptide (SBP) (Keefe et al, 2001) and SUMO[Eu] (Vera Rodriguez et al, 2019) were used to generate a type 3a plasmid from pYTK001. For BsaI/T4 ligase (NEB) mediated GoldenGate assembly, these plasmids were combined with pYTK008, pYTK030, pYTK073, pYTK074, pYTK082, and pYTK084, resulting in 2 µ plasmids with ura3 marker, encoding for SBP-SUMO[Eu]-Sbh2 under control of a Gal1 promoter. The MoClo-YTK plasmid kit was a gift from John Dueber (Addgene kit # 1000000061).

## Protein purification

All steps were performed at 4 °C or on ice unless indicated otherwise. Expression and purification of full-length Ubc6 and its mutants were performed as described (Schmidt et al, 2020). For expression of the Ubc6 UBC domain and its mutants an overnight culture was diluted 1:50 into LB-medium with ampicillin (100 µg/L) and grown at 37 °C. Expression was induced at an $OD_{600}$ of 0.7 with the addition of 0.5% (w/v) ʟ-arabinose (Sigma-Aldrich A3256). Cells were grown at 18 °C for 16 h and harvested by centrifugation at 4000 rpm, 15 min. Cell pellets were resuspended in about four times the cell volume with a buffer containing 50 mM Tris-HCl pH 8.0 (4 °C), 100 mM sodium chloride, 30 mM imidazole, supplemented with 1 mM PMSF, 1 mg/ml lysozyme, and 0.1 mg/mL DNaseI. After incubation for 30 min at ambient temperature, cells were subjected to sonication. Subsequently, the NaCl concentration was increased to 500 mM. This lysate was clarified by two centrifugation steps (1: 2000×g, 20 min, 2: Ti70 rotor, 42,000 rpm, 30 min). The supernatant was incubated with 1 mL Ni-NTA resin (Thermo Scientific) for 3 h. The beads were then filtered off and washed with 4 ×25 ml of buffer containing 50 mM Tris-HCl pH 8.0, 250 mM NaCl, 30 mM Imidazole. To elute the protein from the bead material, 1 µM Ulp1 was added to beads resuspended in a minimal amount of buffer and incubated for at least 30 min. After separation from bead material by filtration, 1 mM DTT was added to the eluted protein. Ubc6 UBC was further purified by size exclusion chromatography using Superdex 75 16/600 column (GE Healthcare), equilibrated with 10 mM HEPES-NaOH, 150 mM NaCl, 0.1 mM TCEP at pH 7.4.

For the preparation of isopeptide-linked Ubc6-ubiquitin, Ubc6 UBC C87K was incubated at 30 °C for 16 h with ubiquitination machinery (f.c. 100 µM Ubc6 UBC (C87K), 240 µM ubiquitin, 1.5 µM E1 Uba1, 3 mM ATP, 5 mM magnesium acetate) in 10 mM HEPES-KOH (pH 7.4), 150 mM KCl. Isopeptide-linked Ubc6 UBC (C87K)-Ub was further purified by size exclusion chromatography using Superdex 75 16/600 column (GE Healthcare) equilibrated with 10 mM HEPES-KOH (pH 7.4), 150 mM KCl. For the

preparation of ubiquitin-loaded UBC domain used in Fig. 6, Ubc6 UBC DL680 was incubated for 10 min at 25 °C with ubiquitination machinery (f.c. 130 μM Ubc6 UBC, 250 μM ubiquitin, 2 μM Uba1, 2 mM ATP, 5 mM magnesium acetate) in 10 mM HEPES-KOH (pH 7.4), 150 mM KCl. After adjusting the pH to 5.5 with citrate buffer (f.c. 100 mM), the ubiquitin-loaded species was purified by size exclusion chromatography using a Superdex 75 increase 10/300 column, equilibrated with 10 mM sodium citrate/citric acid, 100 mM KCl, pH 5.5.

Expression of Doa10 RING domain (aa 1–129), Doa10-CTE fusion, MarchF6 RING-CTE, Hrd1 RING domain (325–551), and Rad6 was induced at an OD$_{600}$ of 0.7 with the addition of 0.5 mM IPTG (Formedium). Cells were grown at 18 °C overnight and harvested (4000 rpm, 15 min, 4 °C). The cell pellet was resuspended in about five times the cell volume in buffer (50 mM Tris-HCl pH 8.0 (4 °C), 500 mM NaCl, 30 mM Imidazole). Cells were lysed by using a microfluidizer (80 psi, two passages) and incubated immediately after with 1 mM PMSF. Subsequent steps were performed as described for Ubc6 UBC.

Expression and purification of Uba1, Ubc7 (Stein et al, 2014), SNAREs, Cue1, ubiquitin (Schmidt et al, 2020), and Usp2 (Vasic et al, 2020) were purified as described.

Expression and purification in yeast were performed as previously described (Schmidt et al, 2020). Cells were lysed by cryo-milling (Planetary ball mill Retsch PM100, 400 rpm for 2.5 min cycles, 8 rounds). Roughly 150 g of cryo-milled yeast cell powder was resuspended in 600 mL buffer (20 mM HEPES-NaOH, 400 mM NaCl, 0.2 mM TCEP, pH 7.4), supplemented with 1 mM PMSF and 2 μM Pepstatin A. The suspension was ultracentrifuged (Ti45, 40,000 rpm, 30 min), and the pellet was resuspended in 300 mL of the same buffer supplemented with 1 mM PMSF and 2 μM Pepstatin A. After douncer-homogenization, this suspension was passed once through a microfluidizer at low pressure (25 psi). Debris was pelleted by centrifugation (2,500 x g, 10 min). The supernatant was ultracentrifuged (Ti45, 40,000 rpm, 45 min). The resulting crude membrane pellet was resuspended in a minimal volume of buffer G (20 mM HEPES-NaOH, 400 mM NaCl, 0.2 mM TCEP and 200 mM Sucrose at pH 7.4) and stored at −80 °C.

For Doa10 purification, the membrane fraction was solubilized in detergent GDN 1.3% (w/v) (Anatrace) at a total protein concentration of 3 mg/mL in 20 mM HEPES-KOH (pH 7.4), 300 mM NaCl, 0.5 mM TCEP at pH 7.4, supplemented with 1 mM PMSF and one EDTA-free protease inhibitor cocktail tablet (Roche) per 100 mL lysate. After incubation for 1 h, insoluble material was pelleted by ultracentrifugation (Ti45, 40,000 rpm, 30 min). The supernatant was incubated with 4 mL of Pierce High Capacity Streptavidin Agarose bead slurry (Thermo Scientific) for 3 h. The beads were then filtered off and washed with 4–6 ×25 mL of wash buffer (20 mM HEPES-NaOH, 400 mM NaCl, 0.2 mM TCEP at pH 7.4). In this process, the GDN concentration was successively lowered to 0.1%, 0.05%, and 0.025% (w/v) GDN. Subsequently, Doa10-SBP was eluted within the same buffer, supplemented with 2 mM biotin. Doa10-SBP was further purified by size exclusion chromatography using Superose 6 increase 10/300 column (Cytiva), equilibrated with 20 mM HEPES-NaOH, 150 mM NaCl, 0.2 mM TCEP, 0.004% GDN, pH 7.4.

To purify Sbh2 and its variants, the membrane fraction was solubilized to 3–4 mg/mL total protein in buffer G, supplemented with 1% n-decyl-maltoside (DM, Glycon), 1 mM PMSF, and 2 μM

Pepstatin A. After solubilization for 1 h, insoluble material was pelleted (Ti45, 40,000 rpm, 30 min). The supernatant was incubated with Pierce High Capacity Streptavidin Agarose bead slurry (Thermo Scientific) for 3 h at 4 °C. The beads were then filtered off and washed with 4 ×25 mL buffer G supplemented with 6 mM DM. Sbh2 was eluted from beads by the addition of 1 μM Sumo$^{EuB}$ protease (Vera Rodriguez et al, 2019) and incubated for 30 min with rotation. The flow through was collected and further purified by size exclusion chromatography using Superdex 200 increase 10/300 column (Cytiva) equilibrated with 20 mM HEPES-NaOH, 250 mM NaCl, 200 mM Sucrose, 6 mM DM at pH 7.4.

## Labeling with fluorescent dyes

Sortase mediated labeling of proteins at the C-terminus via LPETGG tag and at the N-terminus via GGGC tag was carried out as previously described (Schmidt et al, 2020). N-terminal labeling of ubiquitin with Alexa Fluor 488 C$_5$ maleimide dye for anisotropy experiments was done as described (Stein et al, 2014).

## Multiple sequence alignments (MSAs)

For the MSA in Appendix Fig. S1, the indicated sequences were retrieved from the UniProt website, aligned using MAFFT with L-INS-i settings (Katoh and Standley, 2013), and visualized in Jalview (Waterhouse et al, 2009). For the identification of Ubc6/Ube2J2 homologs, first a seed alignment was generated using MAFFT with human Ube2J2 and Ube2J1, S. cerevisae and S. pombe Ubc6, A. thaliana UBC33 and UBC32. The seed was then used as a search model in HMMER hmmsearch (Potter et al, 2018) using the rp15 representative proteome database (Chen et al, 2011). Sequences were retrieved with cut-offs that excluded human E2s other than Ube2J1 and Ube2J2. These sequences were aligned using MAFFT with L-INS-i settings, including yeast Ubc4 and human Ube2D1 as an outgroup to facilitate root placement. The alignments were manually inspected, and long (>500 residues) and short (<150 residues) sequences, as well as sequences with long gaps or insertions in the UBC domain removed. A phylogenetic tree was constructed using IQ-TREE (Nguyen et al, 2015). This showed a clear separation into two major branches, which included either human Ube2J2, yeast and S. pombe Ubc6, and A. thaliana UBC33, or human Ube2J1 and A. thaliana UBC32 (shown in Appendix Fig. S2 for a smaller set of sequences). All sequences from the Ube2J1 branch were removed from the alignment, and the sequences truncated at position 144 of yeast Ubc6. From this alignment of 199 sequences (Source Data to Fig. 1), the WebLogo in Fig. 1A was generated (Crooks et al, 2004). Similarly, to generate the weblogo in Appendix Fig. S3A, sequences from the Ube2J2 branch were removed.

## Crystallization

### Ubc6 UBC domain in cadmium containing buffer

Purified Ubc6 UBC (aa 1-172) was concentrated to 22 mg/mL in 10 mM HEPES-NaOH (pH 7.4, 4 °C), 150 mM NaCl, 1 mM TCEP at pH 7.4. Crystals were grown at 20 °C in hanging drops by mixing with a precipitant solution containing 15% (v/v) PEG Smear (Molecular Dimensions, MD2-100-259), 10% (v/v) ethylene glycol, 0.2 M ammonium sulfate, 0.01 M cadmium chloride hydrate, and

0.1 M PIPES buffer at pH 7. Crystals appeared within 5 days. Crystals were cryoprotected using mother liquor solution concentrated to 80% the original volume using a SpeedVac (Eppendorf, Germany), supplemented with glycerol to a final concentration of 20% (v/v), and flash-frozen in liquid nitrogen.

### Ubc6 UBC domain in citrate-containing buffer

Crystals were grown at 20 °C in sitting drops with a precipitant solution containing 23.2% (v/v) PEG 3350 (Molecular Dimensions, MD2-100-259), 0.34 M di-ammonium hydrogen citrate and a seed stock prepared from smaller crystals obtained with the same precipitant solution in hanging drops. Crystals appeared within 3 days. Crystals were cryoprotected using mother liquor solution concentrated to 80% its original volume, supplemented with glycerol to a final concentration of 20% (v/v) and flash-frozen in liquid nitrogen.

### Isopeptide-linked Ubc6 UBC (C87K)-Ub

Purified isopeptide-linked Ubc6 UBC (C87K)-Ub was concentrated to 20 mg/mL in 10 mM HEPES-KOH pH 7.4, 150 mM KCl. Crystals were grown at 20 °C in sitting drops by mixing with a precipitant solution containing 12% (w/v) PEG smear medium (Molecular Dimensions, MD2-100-259), 100 mM TRIS/HCl pH 8, 75 mM sodium acetate, 100 mM sodium chloride. Crystals appeared within 10 days. Crystals were cryoprotected using mother liquor solution concentrated to 80% its original volume, supplemented with glycerol to a final concentration of 20% and flash-frozen in liquid nitrogen.

Diffraction data were collected at PXII-X10SA beamline at the Swiss Light Source (SLS, Paul Scherrer-Institute, Villigen, Switzerland) at 100 K. All datasets were processed using automated data processing software provided by the SLS facility, using XDS (Kabsch, 2010) and autoPROC (Vonrhein et al, 2011). For structure determination, we used the Phenix package (Liebschner et al, 2019). For the first Ubc6 UBC domain structure containing cadmium (II) ions, the structure was solved by single-wavelength anomalous dispersion (SAD) using AutoSol (Terwilliger et al, 2009). For the Ubc6 UBC structure in citrate-containing buffer, structure solving was done by molecular replacement (MR) in PHASER (McCoy, 2007) using the previous structure as a search model. For the ubiquitin-bound structure, the structure was solved by MR in PHASER using the first Ubc6 UBC domain structure and ubiquitin from PDB ID 4AP4 (Plechanovova et al, 2012) as search models. Iterative cycles of manual model building and refinement were performed in Coot (Emsley et al, 2010) and in Phenix.refine (Afonine et al, 2012). Figures were prepared in Pymol (Available at: http://www.pymol.org/pymol).

## Force field parameters

The force field parameters for the thioester connecting $Cys87_{Ubc6}$ with the C-terminus of $G76_{Ub}$ were derived from the Charmm General Force Field (CGenFF). We utilized the ParamChem web server for the assignment of atom types and preliminary parameters, based on glycine, capped by a N-terminal Acetate and a C-terminal S-Methyl (Fig. EV1A). Any parameters not available in CGenFF were assigned by analogy. The penalty score provided by ParamChem suggested a reasonable analogy for the partial charges. Although most bonded parameters were available, the potential function describing the rotation around the carbonyl-Cα bond of was inferred from an unreliable analogy, as indicated by a penalty score of 58. To determine, how accurately these parameters describe the potential, we performed potential energy scans at 15° increments using the

wavefront propagation implemented in torsiondrive software (Qiu et al, 2020). Quantum Mechanical calculations were carried out in Psi4 at MP2/6-311 G(d) model chemistry (Smith et al, 2020). Comparison of molecular mechanics potential with ab initio calculations, confirmed that the initial parameters inadequately represented the potential energy surface (Fig. EV2B,C). We therefore calibrated these parameters to better reproduce the quantum mechanical potential.

Initial optimization focused on the rotation around the C–Cα bond, which corresponds to the protein backbone Ψ dihedral. However, we were unable to find parameter that reproduced the QM potential accurately. We identified two issues: First, we identified an aberrant potential function on the SG-C-Cα-$H_{Cα}$ atoms. To rectify this without having to make modifications to CGenFF, we introduced a new atom type for the aliphatic $H_{Cα}$ (HGX2). Except for the SG-C-Cα-$H_{Cα}$ dihedral term, this atom type inherited all bonded and nonbonded parameters from the HGA2 atom type that had previously been used for $H_{Cα}$. the SG-C-Cα-$H_{Cα}$ dihedral term was set to zero. A second issue was identified by visual inspection of the QM and Molecular mechanics-based dihedral scans. These revealed, that the CGenFF parameters underestimated the barrier heights for the adjacent dihedral centered on the Cα–N bond, which corresponds to the φ dihedral. This was not unexpected; CGenFF is intended as a broad starting point for the parameterization of small molecules. The dihedral parameters describing the dynamics of the protein backbone are complex and described with much higher accuracy in the CHARMM Protein force fields (MacKerell et al, 2004). To determine the interdependence between the φ and Ψ dihedral angles, we performed 2D potential energy scans of the two dihedral angles at 15° increments. Because this required significantly longer calculations, we used MP2/6-31 G(d) model chemistry. Afterward, we refitted the φ and Ψ angles to the 2D QM scans using the forcebalance software (Wang et al, 2013; Wang et al, 2014). The final reparametrized potential functions reproduced the QM data (Fig. EV2B,C). The final parameters for the thioester were converted in GROMACS format and are made available in the supporting data. The thioester bond is defined in the specbond.dat file and requires a distance of 1.4 ± 0.14 Å between the cysteine sulphur and the C-terminal carbonyl. The system topology can be created automatically using the GROMACS *pdb2gmx* utility, however the improper dihedral term for the glycine carbonyl needs to be added after the topology has been created, because it cannot be defined in the pdb2gmx input files.

## Molecular dynamics simulation of Ubc6-thioester loaded ubiquitin model

The initial model of the native Ubc6-Ub complex was created using the Pymol mutagenesis tool (available at: http://www.pymol.org/pymol). Lysine 87 was changed to cysteine and the system was energy minimized and geometry optimized with Schrödinger Protein Preparation Wizard using default parameters (Madhavi Sastry et al, 2013). The pKA and the protonation state of the titratable groups were predicted with Propka3 (Olsson et al, 2011). Since the tautomeric state of the monoprotonated His94 affects its putative catalytic function, subsequent steps were performed for both tautomers unless otherwise noted. Parameterization, energy minimization and simulations were carried out using GROMACS version 22.3 (Abraham et al, 2015). The system was parameterized

using the CHARMM36M forcefield (Huang et al, 2017). Custom parameters were used for the C87$_{Ubc6}$-G76$_{Ub}$ thioester. Thereafter, the molecules were solvated in a rhombic dodecahedral box, leaving at least 1.4 nm between the box boundaries and the nearest protein-heavy atom. Sodium and chloride ions were added to neutralize the charge of the system and achieve a total ion concentration of 150 mM. Subsequently, the system was minimized using the steepest descent algorithm, while applying position retrains (1000 kJ/mol nm$^2$) to the protein-heavy atoms. This was followed by a second minimization without position restrains. Both minimizations were carried out to machine precision.

Energy minimization was followed by a series of short simulations to equilibrate the system. During those simulations, the temperature was kept constant at 300 K using the modified Berendsen thermostat (Bussi et al, 2007). In the first simulation, initial velocities were drawn from a Maxwell distribution, in subsequent simulations initial velocities were assigned based on previous simulation. The P-LINCS algorithm was used to constrain the length of all bonds with hydrogens (Hess, 2008). Nonbonded forces were truncated at 1.2 nm and van der Waals forces were smoothly switched to 0 between 1.0 nm and 1.2 nm. Long-range electrostatics were calculated using the particle mesh ewald methods with a 0.12 nm grid spacing (Essmann et al, 1995). All simulations were carried out using a 2 fs time step. For equilibration the system was first simulated for 1 ns in the NVT ensemble. Position retrains of 1000 kJ/mol nm$^2$ were applied to the protein. This was followed by a 1 ns simulation in the NPT ensemble. Pressure was kept at 1 bar using the Berendsen thermostat (Berendsen et al, 1984), position retrains were applied to the protein backbone atoms. This was followed by a second set of simulations in the NVT and NPT ensemble without position restrains. In the NPT simulation, exponential relaxation pressure coupling (C-rescale) was used to keep a constant pressure of 1 bar (Bernetti and Bussi, 2020).

Simulated annealing was performed in GROMACS using the equilibrated system. The pressure was kept constant at 1 bar using the C-Rescale barostat. Protein and solvent were coupled to separate thermostats, and target temperature was set using the V-rescale thermostat. To this end, protein complex was simulated at 400 K for 16 ns, water and ions were coupled to a separate thermostat and kept at 300 K over the same period. To prevent unfolding, position retrains of 1000 kJ/mol nm$^2$ were applied to the protein backbone (N, C, CA, O). No restrains were applied to the residues in the active site (79$_{Ubc6}$–98$_{Ubc6}$; 115$_{Ubc6}$–128$_{Ubc6}$; 71$_{Ub}$–76$_{Ub}$). After 16 ns, the temperature of both solvent and solute was decreased linearly to 50 K over a period of 32 ns. Finally, the system was minimized to machine precision using steepest descent. No position restraints were applied during minimization.

After simulated annealing, the systems were simulated for 1 μs using the same parameters used in the last (npt) equilibration. A total of 100 simulations with His94 in the δ-nitrogen protonated state and 16 simulations with His94 in the ε-nitrogen protonated state were performed. The initial velocities were assigned from a Maxwell distribution. The initial coordinates for the production MD simulation were chosen from the configurations generated by simulated annealing. Principal component analysis of the conformation generated by simulated annealing indicated, a unimodal distribution of the configurations around a common mean, accordingly we chose the structure that represented that structure the closest. The δ-nitrogen protonated His94 tautomer was identified as the preferred tautomer due to its ability to form an

additional hydrogen bond with the backbone of Leu88. For the MD simulations of the E119A and E119D mutant variants, we used the coordinates that were also used for the WT simulations. The mutations were introduced using the Pymol mutagenesis tool. Eight replicates were simulated for each variant. Only the δ-N protonated tautomer of His94 was simulated for the mutant variants.

## Simulation analysis

Hydrogen bond analysis was carried out using the MDAnalysis toolkit (Naughton et al, 2022). The default geometric criteria were used to describe the hydrogen bonds: At most 0.12 nm between the acceptor atom and the donor hydrogen. At most 0.3 nm between the donor and the acceptor and a Donor–Hydrogen–Acceptor angle of at most 150°. For salt bridge analysis, we decided to define a salt bridge based on the distance between the centers of mass of the ion pairs, which should not exceed 5 Å. This cutoff is more lenient than criteria proposed in the literature (Kumar and Nussinov, 2002), however, the structures analyzed here are snapshots from a trajectory at 300 K rather than structure averages. Furthermore, it was shown that most ion pairs have stabilizing contributions if there are centroids within 5 Å (Kumar and Nussinov, 2002). We used POVME 2.0 to measure shape and volume of the active site pocket (Durrant et al, 2014). First, the trajectories were aligned to a common reference, minimizing the root mean square deviation between the C$_\alpha$ atoms in the active site. We defined a spherical volume with a radius of 12 Å, centered on the axis between residues C87$_{Ubc6}$Q49$_{Ub}$ and F114$_{Ubc6}$–R72$_{Ub}$ as the putative pocket volume. For each frame, the inclusion region was flooded with equidistantly spread points. The spacing between points was 0.5 Å. All grid points that were within 1.09 Å of any heavy atom were deleted. Likewise, grid points outside of the convex hull were removed as well. Finally, the pocket volume was calculated from the remaining grid points. The His94 dihedral angles were analyzed using the MDAnalysis toolkit. The sidechain dihedral angles were defined by the following atoms: $\chi_1 = N$-C$_\alpha$-C$_\beta$-C$_\gamma$, and $\chi_2 = C_\alpha$-C$_\beta$-C$_\gamma$-N$_\delta$. The signed distance of the unprotonated His94 nitrogen from the thioester plane, i.e., the plane through Cys87$_{Ubc6}^{S\gamma}$, Gly76$_{Ub}^C$ and Gly76$_{Ub}^O$, was calculated by first centering the coordinates on Cys87$_{Ubc6}^{S\gamma}$. Subsequently we calculated the signed distance as the dot product between the plane normal and the centered His94$_{Ubc6}^{N\epsilon}$ coordinate vector.

## Cycloheximide chase assays

Cycloheximide chase assays were performed essentially as described (Gardner et al, 1998). Pre-cultures of *S. cerevisiae* strains were diluted into selective drop-out medium (SDC-uracil) and grown at 30 °C. The log-phase cultures were pelleted, resuspended to an OD$_{600}$ of 5 in SDC-uracil medium and cycloheximide was added to a final concentration of 0.25 mg/ml (Hampton and Rine, 1994). At indicated time points, 1 mL samples were collected, centrifuged using a benchtop centrifuge at 6000 rpm for 3 min, washed with cold water and immediately snap-frozen in liquid nitrogen. For analysis, total protein extract was prepared by resuspending cells in 100 μL cracking buffer pre-heated to 70 °C. The cracking buffer was composed of 2 mL of cracking buffer stock (8 M urea, 50 g/L SDS, 40 mM Tris-HCl pH 6.8 and 0.1 mM EDTA), 20 μL 2-mercaptoethanol, 100 μL 100 mM PMSF and 100 μL protease inhibitor (PI) cocktail solution. The PI cocktail

was prepared by resuspending one cOmplete, EDTA-free tablet (Roche) in 2 mL MilliQ water. Cell suspensions were transferred into 2 mL screw-cap tubes containing 0.5 mm zirconia beads (Roth). After incubation for 10 min at 70 °C, cells were lysed using a Bead Ruptor 12 (Omni International), for a single round at maximum speed for 30 s. The samples were then centrifuged using a benchtop centrifuge at 10,000 rpm for 5 min and analyzed by SDS-PAGE, followed by blotting onto the nitrocellulose membrane. Membranes were blocked in 5% (w/v) skimmed-milk in Tris-buffered saline (TBS, 20 mM Tris-HCl pH 7.4, 150 mM NaCl), supplemented with 0.1% (v/v) Tween20 (TBS-T) for 1 h at 25 °C. Antibody decoration was done by incubating with an anti-HA-tag antibody diluted 1:1000 in 5% (w/v) milk in TBS-T (rat anti-HA, clone 3F10, Roche) for 1 h at RT, followed by incubation with goat anti-rat 800 IRDye® CW (Li-Cor Biosciences) secondary antibody (diluted 1:15000 in 5% (w/v) milk in TBS-T) for 1 h at RT. For loading control, membranes were incubated with an antibody against Pgk1 (diluted 1:2000 in 5% (w/v) milk in TBS-T, mouse/IgG1 κ chain anti-Pgk1 from Invitrogen cat# 22C5D8) for 1 h, followed by incubation with goat anti-mouse 680 IRDye® RD (Li-Cor Biosciences) secondary antibody (diluted 1:15000 in 5% (w/v) milk in TBS-T) for 1 h at RT. Blots were imaged on an Odyssey CLx scanner (Li-COR) and band intensities were quantified using the ImageStudio Lite software (Li-COR).

## Reconstitution of proteins into liposomes and characterization of proteoliposomes

Preparation of protein-free liposomes and reconstitution of full-length Ubc6 and its variants as well as Doa10 and Cue1 into liposomes was done as described (Schmidt et al, 2020). For co-reconstituting Sbh2 with Ubc6, Cue1, and Doa10, a direct 1-step co-reconstitution protocol was used. Protein-free liposomes (4 mM final lipid concentration) were mixed with DM (final concentration 6 mM) and proteins in 20 mM HEPES-KOH, 150 mM potassium chloride, 5 mM magnesium chloride, pH 7.4 supplemented with 1 mM DTT. Sbh2, Ubc6 and Doa10 were each co-reconstituted at a molar lipid: protein ratio of 2000. After 1 h RT incubation, detergent was removed by incubation with buffer-washed Pierce detergent removal spin column resin (Thermo Scientific) in three steps, with 45 mg of resin added to 130 µl reconstitution reaction in each step and incubated for 20 min each at RT. Beads were removed using standard spin columns and a benchtop centrifuge at 3500 rpm for 2 min.

To test for efficiency of reconstitution, we used Nycodenz co-floatation assay. An 80% (w/v) Nycodenz stock solution was prepared in 20 mM HEPES-KOH, 150 mM potassium chloride, 5 mM magnesium chloride, 0.1 mM TCEP, pH 7.4. At the bottom-most layer, 50 µL of proteoliposomes were mixed with 50 µL of 80% (w/v) Nycodenz, overlaid with 40 µL of 30% (w/v) and 15% (w/v) Nycodenz and a final layer of 40 µL buffer. The resulting step gradient was ultracentrifuged (S55-S rotor, 50,000 rpm, 1 h at 4 °C). The gradient was fractionated from the top into six fractions and analyzed by SDS-PAGE followed by fluorescence scanning on an Odyssey CLx (Li-COR) and Coomassie staining.To determine the orientation of Sbh2 in liposomes, a trypsin protection assay was used. In this assay, proteoliposomes were diluted 1:10 in buffer and incubated with 6.6 µg/mL Trypsin (Roche) protease at RT. As a positive control, 1% (v/v) Triton X-100 (Anapoe-X-100, Anatrace) was added. The reaction was incubated for 1 h and samples were

collected after 20 min and 1 h and the reaction was stopped with the addition of 4 mM PMSF for 10 min. Subsequently, LDS sample buffer (f.c. 50 mM Tris/HCl, pH 7, 4% (w/v) lithium dodecyl sulfate (LDS), 10% (w/v) glycerol, 0.017% (w/v) Coomassie Blue G250) (Schagger, 2006) was added. Samples were analyzed by SDS-PAGE and fluorescence scanning. Densitometry was performed using ImageStudio Lite Software (Li-COR). To determine co-reconstitution of Sbh2 (and its variants) with Doa10 and Ubc6, pull-down assays using the C-terminal SBP tag of Doa10 were used. In this reaction, 20 µL of liposomes were incubated with 20 µL of Pierce Streptavidin Magnetic Beads (Thermo Scientific) prewashed with buffer (20 mM HEPES-KOH, 150 mM potassium chloride, 5 mM magnesium chloride), supplemented with 0.25 mg/mL BSA. After binding for 1 h at RT with rotation, the supernatant was removed, and the beads washed thrice with 100 µL of buffer. Bound material was eluted with 20 µL of buffer supplemented with 2 mM biotin. Input, inbound and elution samples were analyzed by LDS-PAGE followed by fluorescence scanning. Densitometry analysis was performed using ImageStudio Lite Software (Li-COR).

## In vitro ubiquitination using proteoliposomes

Unless indicated otherwise, all ubiquitination reactions were performed at 30 °C. For measuring Ubc6 autoubiquitination using full-length Ubc6 and its mutants reconstituted with Doa10, ubiquitination assays were performed as described (Schmidt et al, 2020). For Sbh2 experiments, liposomes were diluted 1:2 (f.c. 0.4 µM Sbh2/Sbh2 4KS, 0.4 µM Ubc6 (or mutants), 0.4 µM Doa10 and 0.4 µM Cue1 (where indicated)). Following, components of the ubiquitination machinery were added 0.1 µM Uba1 (E1), 1 µM Ubc7, 120 µM ubiquitin and 2.5 mM ATP, 0.1 mg/mL BSA. For experiments shown in Figs. 4I,J and EV3F and Appendix Fig. S6H, 1 µM Uba1 was used, instead of 0.1 µM. Samples were collected at indicated time points and the reaction was stopped by adding reducing or non-reducing LDS sample buffer. The reducing sample buffer contained 2% (v/v) 2-mercaptoethanol. For monitoring serine/threonine ubiquitination, samples were first collected in LDS sample buffer supplemented with 6 mM DTT and subsequently diluted 1:2 into either sample buffer or sample buffer containing 200 mM sodium hydroxide, followed by incubation at 70 °C for 5 min. Samples were then analyzed by SDS-PAGE and fluorescence scanning.

For analysis of full-length Ubc6 autoubiquitination, the non-ubiquitinated band was quantified using ImageStudio Lite software (Li-Cor) and normalized to the first time point at $t = 0$. For analysis of Sbh2 ubiquitination, intensities of modified and non-modified species were determined for each lane either in Fiji (Schindelin et al, 2012) or the ImageStudio Lite Software (Li-COR). The fraction of non-modified Sbh2 was then calculated by dividing the intensity of the non-modified species by the sum of the intensities of modified and non-modified species. A background band in the upper part of the gel (marked with an asterisk in Fig. 4E), which probably arises from an SDS-resistant association of labeled Sbh2 with Doa10, but is absent in the individual purified proteins, was subtracted from each lane. Similarly, for quantification of the fraction of the ubiquitination sensitive to sodium hydroxide, intensities of bands corresponding to the non-modified and all modified Sbh2-species were determined separately, and the fraction modified was calculated as a fraction of the total signal for each

lane. In this case, background signal deriving from impurities as determined in the -ATP sample were subtracted. The NaOH-insensitive fraction was determined by dividing the background-corrected modified fraction from samples with NaOH treatment by the background-corrected modified fraction from samples without NaOH treatment, the NaOH-sensitive fraction as the difference between these two values.

## Ubiquitin loading and discharge assays

For analysis of Ubc6 loading with ubiquitin in Fig. EV3A, 2 µM DyLight 680-labeled full-length Ubc6 or its mutant variants were incubated in the presence of 50 nM Uba1, 10 µM ubiquitin, 1 mM ATP, and 5 mM magnesium acetate for 5 min at 25 °C in CHC buffer (Newman, 2004) (final concentrations 11.1 mM citric acid, 16.7 mM HEPES, 22.2 mM CHES, 100 mM potassium chloride pH 7.4 adjusted by mixing appropriate volumes of pH 4 and pH 10 CHC stock solutions), supplemented with 0.2 mg/mL BSA and 0.03% (w/v) DDM. Samples were collected at indicated time points in non-reducing LDS sample buffer and analyzed by SDS-PAGE and fluorescence scanning using an Odyssey CLx scanner (Li-COR).

In vitro ubiquitin discharge assays were performed in two steps, E1-mediated E2 loading followed by E2 discharge. For step 1, 20 µM unlabeled UBC domain was incubated in CHC buffer in the presence of 50 nM Uba1, 30 µM ubiquitin, 1 mM ATP, 5 mM magnesium acetate, 0.2 mg/ml BSA, 0.03% (w/v) DDM for 7 min at 25 °C. Where indicated, 2 µM DyLight 680-labeled proteins and 3 µM ubiquitin were used instead. For step 2, the reaction from step 1 was diluted 1:10 in CHC buffer supplemented with 5 mM EDTA. These reactions were supplemented as indicated with nucleophiles such as free amino acids or glycerol. Samples were collected at indicated time points in non-reducing LDS sample buffer. The samples were then analyzed by SDS-PAGE, followed by either stain-free detection using a BioRad Gel Doc EZ Imager or by fluorescence scanning.

To perform discharge assays in the presence of free amino acids as nucleophiles, stock solutions of free amino acids were prepared in CHC buffer and if necessary adjusted to pH 7.4 with sodium hydroxide (0.5 M for L-alanine, L-threonine, L-lysine, L-serine, and L-glutamic acid; 0.25 M for L-histidine; 0.2 M for N-acetyl-L-histidine (Thermo Scientific, J65657); 0.1 M for N-acetyl-L-tyrosine (Sigma-Aldrich T4446)). A 0.012 M L-cysteine stock solution was always freshly prepared to minimize oxidation. To measure the solvent isotope effect in Fig. 5K, both the E2 loading and discharge were performed using deuterated water. In this case, all solutions were made with 99.9% deuterium oxide ($D_2O$, Eurisotop), and the pD was adjusted using deuterated sodium hydroxide (NaOD, Sigma-Aldrich) or deuterium chloride (DCl, Sigma-Aldrich) by adding 0.4 to the pH electrode reading (Gadda and Fitzpatrick, 2013; Schowen and Schowen, 1982). For discharge assays in the presence of E3, the chase reaction was supplemented with the indicated concentrations of RING domain proteins, usually 10 µM. For measuring the effect of EDTA on E3 activity, discharge assays were performed by directly diluting a pre-loaded and purified Ubc6 UBC domain.

For ubiquitin discharge from wt Rad6 and its Q93H mutant the assay was performed with slight modifications. In total, 10 µM unlabeled E2 enzyme was incubated with 50 nM Uba1, 10 µM ubiquitin, 1 mM ATP, and 5 mM magnesium acetate for 2 min at 25 °C in CHC buffer, supplemented with 0.2 mg/ mL BSA and 0.03% (w/v) DDM. For E2 discharge (Step 2), thioester Ub-loaded E2 was diluted (1:10) in CHC buffer, supplemented with 10 mM EDTA, and in the absence or presence of free amino acids as indicated.

To monitor autoubiquitination of full-length Ubc6 and its mutants in Fig. EV5, 4 µM of WT Ubc6 or its mutants versions were incubated with 50 nM Uba1, 30 µM ubiquitin, and 1 mM ATP in CHC buffer supplemented with 5 mM magnesium acetate, 0.2 mg/ mL BSA, and 0.03% (w/v) DDM for 2 min at 25 °C, followed by addition of EDTA to 5 mM, followed by a 1 min incubation at 30 °C. Then, the clock was started and samples collected in LDS sample buffer supplemented without or with 10 mM DTT. Samples were analyzed SDS-PAGE and stain-free imaging or fluorescence scanning.

### *Fluorescence anisotropy-based in vitro ubiquitin discharge assay*

These ubiquitin discharge assays were performed essentially as described above with some modifications. The loading reaction contained 2 µM Alexa Fluor 488 labeled ubiquitin, 5 µM Ubc6 UBC, 30 to 50 nM Uba1, 1 mM magnesium acetate, 0.2 mg/ml BSA, 0.03% (w/v) DDM, in CHC buffer. The reaction was started by adding ATP to a final concentration of 1 mM. After ten minutes incubation at 25 °C, this reaction was diluted 1:10 into CHC buffers, supplemented with 0.2 mg/ml BSA, 5 mM EDTA, 0.03% (w/v) DDM. To perform a pH titration in Fig. 5J, reactions from step 1 performed in 0.5× CHC buffer were diluted 1:10 into 2× CHC buffers. CHC buffers of different pH were prepared by mixing the appropriate amounts of stock solutions of either pH 4 or pH 10 (Newman, 2004). The reaction was carried out in a 96-well plate (Corning 3686) and the fluorescence polarization was measured using a BioTek Synergy Neo2 Multi-mode reader (Software version 3.08.01) using dual PMT with filters 485/20 for excitation and 528/20 for emission, a Xenon flash lamp with lamp energy set to low, standard dynamic range, read speed set to normal, ten measurements for data point, and 6.5 nm read height. The gain was set between 54 and 56 with G-factor 1.01216. The temperature was 25 °C. Data points were collected every 10–14 s.

## Analysis of ubiquitin discharge assay

We performed analysis of fluorescence and stain-free images by densitometry in either Fiji (Schindelin et al, 2012) or the ImageStudio Lite Software (Li-COR). The linearity of the fluorescence signal and the signal obtained using stain-free technology was established in preliminary experiments. Importantly, as the stain-free system depends on tryptophan content and because ubiquitin does not contain tryptophan residues, the signal obtained with this method is proportional to the amount of E2 in the protein band. Further data analysis was performed in Origin Pro 2018b (Origin Lab Corporation). The fraction of loaded E2 at each time point was determined by dividing the intensity of the band for the loaded species by the sum of the band intensities of loaded and apo forms. These data were plotted against the time and rate constants were determined by least squares curve fitting using the Levenberg–Marquardt algorithm. Loaded E2s ($E2_L$) predominantly undergo two competing reactions: discharge by hydrolysis (1), a pseudo-first-order reaction, and discharge by a different nucleophile (Nu)(2), a second-order reaction. In addition, we observed low levels of autoubiquitination ($E2_{Ub}$), which were

slightly increased in some mutants, e.g., H94Q and E119A. The autoubiquitinated state runs indistinguishably from the Cys-loaded state in non-reducing SDS-PAGE. Hence, the mono-ubiquitinated E2 state is a mixture of loaded and autoubiquitinated E2. To simplify the fitting procedure, we assumed autoubiquitination occurs intra-molecularly in a first-order reaction (3). Together this results in the following three reactions:

$$E2_L + H_2O \rightarrow E2 + Ub \tag{1}$$

$$E2_L + Nu \rightarrow E2 + Ub - Nu \tag{2}$$

$$E2_L \rightarrow E2_{Ub} \tag{3}$$

These reactions result in the following rate laws:

$$\frac{d[E2]}{dt} = k_1[E2_L] + k_2[E2_L][Nu] \tag{4}$$

$$-\frac{d[E2_L]}{dt} = k_1[E2_L] + k_2[E2_L][Nu] + k_3[E2_L] \tag{5}$$

$$\frac{d[E2_{Ub}]}{dt} = k_3[E2_L] \tag{6}$$

These differential equations have the following solutions:

$$[E2_L] = [E2_L]_0 e^{-(k_1 + k_2[Nu] + k_3)t} \tag{7}$$

$$[E2_{Ub}] = \left(\frac{k_3[E2_L]_0}{k_1 + k_2[Nu] + k_3}\right)\left(1 - e^{-(k_1 + k_2[Nu] + k_3)t}\right) \tag{8}$$

The sum of Eqs. (7) and (8) was then used in the fitting procedure to determine rates of hydrolysis ($k_1$) and discharge by a nucleophile ($k_2$):

$$[E2_L] + [E2_{Ub}] = [E2_L]_0 e^{-(k_1 + k_2[Nu] + k_3)t}$$
$$+ \left(\frac{k_3[E2_L]_0}{k_1 + k_2[Nu] + k_3}\right)\left(1 - e^{-(k_1 + k_2[Nu] + k_3)t}\right) \tag{9}$$

To this end, dataset pairs from reactions lacking or containing nucleophiles like free amino acids at one or several concentrations were globally fitted, with $[E2_L]$, $k_1$, $k_2$, and $k_3$ set to "shared" and $[Nu]$ fixed to the corresponding concentrations. Reactions including RING domains were fitted using the same equation. In this case, reported rate constants rather represent apparent rate constants, only meaningful at this particular RING concentration, as discharge in these reactions occurs from mixtures of RING-bound and free $E2_L$.

## Mass spectrometry

In-gel digestion was performed as described previously (Shevchenko et al, 1996) with some modifications. Briefly, following overnight tryptic digestion at 37 °C, an additional step of chymotryptic digestion was performed by incubating the gel pieces at 25 °C for 4 h with digestion buffer containing 50 mM $NH_4HCO_3$, 5 mM $CaCl_2$ and 0.5 µg of chymotrypsin. The peptides were further extracted by one change of 20 mM $NH_4HCO_3$, 50% acetonitrile and one change of 50 acetonitrile (ACN) for 15 min and dried in a SpeedVac concentrator (Eppendorf, Hamburg, Germany).

LC-MS/MS acquisition was performed using a Q-Exactive HF Hybrid Quadrupole-Orbitrap Mass Spectrometer with a front-end Dionex UltiMate 3000 RSLCnano system (both Thermo Fisher Scientific, Waltham, USA). Dried peptides were resuspended in 5% ACN, 0.05% TFA (v/v) and injected onto a C18 PepMap100-trapping column (0.3 × 5 mm, 5 µM, Thermo Fisher Scientific, Waltham, USA) coupled to a C18 analytical column packed in-house (75 µM × 340 mm, Reprosil-Pur 120 C18-AQ, 3 µM, Dr. Maisch, GmbH, Ammerbuch, Germany). A linear ACN gradient from 9% to 42% buffer B (80% (v/v) ACN, 0.08% (v/v) formic acid) was applied for 73 min at a flow rate of 0.300 µl/min. The eluted peptides were injected into a Q-Exactive HF Mass Spectrometer (Thermo Fisher Scientific, Bremen, Germany) operated in a data-dependent acquisition mode. Full MS1 scans were acquired in the range of 350–1600 *m/z* at a resolution of 60,000 at *m/z* 200, with an automatic gain control (AGC) of 1E6 and maximum injection time of 50 ms. The 30 most abundant precursor ions with charge state of +2 to +6 were selected using a 1.6 *m/z* isolation window and fragmented with a normalized collision energy (NCE) of 30%. MS2 fragment spectra were acquired at a resolution of 15,000 at *m/z* 200 and with AGC target of 1E5 and maximum IT of 60 ms. Dynamic exclusion was applied for 30 s.

Raw files were analyzed using the MaxQuant (MQ) software (version 2.03.0) (Cox et al, 2011; Tyanova et al, 2016). MS1 and MS2 spectra were searched against a custom protein sequence database, including the proteins of interest (Ubc6, Ubi4, Uba1, Doa10) from *S. cerevisiae*. The default MQ search options were used with the following exceptions; in addition to the default variable (methionine oxidation and acetylation of protein N-termini) and fixed (cysteine carbamidomethylation) modifications, the di-glycine (GG) ubiquitin remnant of lysine, histidine, serine and threonine residues was set as a variable modification. Specific digestion with trypsin and chymotrypsin was selected, allowing up to three missed cleavage sites per peptide.

Acetone precipitated ubiquitin was resuspended in 50 µl of 200 mM ammonium acetate, pH 7. Overall, 1 µl of ubiquitin solution was mixed with 19 µl of 2%ACN, 0.05% formic acid and injected onto a C18 PepMap100-trapping column (0.3 × 5 mm, 5 µM, Thermo Fisher Scientific, Waltham, USA) coupled to a C18 analytical column packed in-house (75 µM × 340 mm, Reprosil-Pur 120 C18-AQ, 3 µM, Dr. Maisch, GmbH, Ammerbuch, Germany). Both columns were equilibrated with a mixture of 90% buffer A (0.1% (v/v) FA in water) and 10% buffer B (80% (v/v) ACN, 0.1% FA in water). A linear ACN gradient ranging from 25% to 60% of buffer B for 23 min was applied at a flow rate of 0.300 µl/min followed by a wash step with 95% buffer B for 6 min. The eluted ubiquitin molecules were further injected into a Q-Exactive HF Mass Spectrometer (Thermo Fisher Scientific, Bremen, Germany) operating in a full MS mode. Full MS scans in the range of 500 -2000 m/z were acquired at a resolution of 60,000 at *m/z* 200, with an automated gain control (AGC) of 1E6 and a maximum injection time of 200 ms. Raw spectra were deconvoluted using the Xtract node of Thermo Xcalibur FreeStyle version 1.8 SP1. The default Xtract options were used. Deconvolution was set to generate neutral masses (M) from charged ions falling in the range of 500–2000 *m/z*.

## NMR spectroscopy

Isotopically labeled $^{15}$N-Ubi was produced in isotope-enriched M9 minimal medium using $^{15}$NH$_4$Cl as the source of nitrogen. Disulfide-linked Ubc6-SS-Ub conjugates were obtained by incubating at least 250 μM Ubc6 UBC WT (aa 1-172) or S89A containing a single cysteine residue with a threefold molar excess of a disulfide-linked adduct of Ub(G76C) with 2-nitro-5-thiobenzoate (TNB) for 1 h at RT followed by ion exchange chromatography (Lorenz et al, 2016). The Ub(G76C)-TNB adduct was prepared prior to Ubc6 conjugation by incubating 1 mM Ub(G76C) with 5 mM DTNB (Ellman's reagent) followed by desalting and buffer exchange over Sephadex G-25 column. Disulfide-linked fractions were analyzed on SDS-PAGE under non-reducing conditions, fractions containing pure Ubc6 UBC–SS–Ub complex were pooled and concentrated for NMR experiments.

NMR experiments were recorded on a 900 MHz Bruker Avance spectrometer with a cryogenic probe at 25 °C in 10 mM MES pH 6.5, 190 mM NaCl, 8% D$_2$O, the final concentration of E2-Ub disulfide linked conjugate or free Ub was 1 mM. The two-dimensional [$^{15}$N,$^1$H]-HSQC and [$^{15}$N,$^1$H]-TRACT (with relaxation delays for free ubiquitin: 0, 28, 56, 84, 112, 140, 168, and 196 ms for alpha and 0, 8, 16, 24, 32, 40, 48, and 56 ms for beta; for Ubc6-Ub complex: 0, 8, 16, 24, 32, and 40 ms for alpha and 0, 1.5, 3.0, 4.5, 6.0, and 7.5 ms for beta) data were acquired. Ubiquitin concentration was 200 μM and the ubiquitin–ubc6 complex was prepared as 1:1 ratio (Lee et al, 2006). NMR datasets were processed using NMRPipe (Delaglio et al, 1995) and analyzed using CCPN (Vranken et al, 2005).

## Data availability

Atomic coordinates have been deposited in the PDB under the accession codes 9EWP and 9EN5 for the UBC domain of Ubc6, and 9EYH for the Ubc6-Ub structure. The model of the Ubc6-Ubiquitin complex with the native thioester bond has been deposited to the ModelArchive (https://www.modelarchive.org) database and assigned the identifier ma-smi96 (https://www.modelarchive.org/doi/10.5452/ma-smi96, access code 6Uw5xA5pWq). Force field files for the Cys to Gly thioester are available via https://doi.org/10.5281/zenodo.13813236.

The source data of this paper are collected in the following database record: biostudies:S-SCDT-10_1038-S44318-024-00301-3.

## Peer review information

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

## Acknowledgements

The authors thank Iris Bickmeyer and Nupur Ranjan for excellent technical assistance, Ulrich Steuerwald and Jürgen Wawrzinek for their support in the crystallization facility of the MPI-NAT, Uwe Pleßmann, Monika Raabe and Sabine König for mass spectrometry assistance, Melanie Wegstroth for technical assistance in NMR sample preparation, and the team of the PXII beamline at the Swiss Light Source (Paul Scherrer Institut, Switzerland) for support during X-ray data collection. The authors would also like to thank Reinhard Jahn, Sonja Lorenz, Kai Tittmann, and the members of the Stein lab for discussions. This work was supported by the European Research Council (ERC) under the Horizon2020 research and innovation program (grant # 677770) and by the Deutsche Forschungsgemeinschaft SFB1190, P15 (both to AS). FL acknowledges funding from the European Union under the Marie Skłodowska-Curie grant agreement No. 101065616-TRISAFA-HORIZON-MSCA-2021-PF-01.

## Author contributions

**Anuruti Swarnkar**: Resources; Data curation; Formal analysis; Validation; Investigation; Visualization; Methodology; Writing—original draft; Writing—review and editing. **Florian Leidner**: Resources; Data curation; Software; Formal analysis; Validation; Investigation; Visualization; Methodology; Writing—review and editing. **Ashok K Rout**: Resources; Data curation; Formal analysis; Validation; Investigation; Visualization; Methodology. **Sofia Ainatzi**: Data curation; Formal analysis; Validation; Investigation; Visualization; Methodology. **Claudia C Schmidt**: Resources; Methodology; Writing—review and editing. **Stefan Becker**: Resources; Writing—review and editing. **Henning Urlaub**: Supervision; Validation; Methodology. **Christian Griesinger**: Supervision; Validation; Methodology; Writing—review and editing. **Helmut Grubmüller**: Supervision; Funding acquisition; Validation; Methodology; Writing—review and editing. **Alexander Stein**: Conceptualization; Data curation; Formal analysis; Supervision; Funding acquisition; Validation; Investigation; Visualization; Methodology; Writing—original draft; Project administration; Writing—review and editing

Source data underlying figure panels in this paper may have individual authorship assigned. Where available, figure panel/source data authorship is listed in the following database record: biostudies:S-SCDT-10_1038-S44318-024-00301-3.

## Funding

## Disclosure and competing interests statement

The authors declare no competing interests.

# Expanded View Figures

**Figure EV1.  Structures and conformational dynamics of Ubc6 and Ubc6-Ub.**

(**A**) Electron density map (2Fo-Fc, contoured at the 1.5 σ level) for the active site loop (orange) and the Thr-flap (pink) of the UBC domain of Ubc6 (PDB: 9EN5). (**B**) Comparison of the two copies of the Ubc6-Ub assembly in the asymmetric unit. For alignment in Pymol, the second copy was aligned to the Ubc6 of the first copy. Thr-flap and GRF loop are colored as in Figs. 1 and 2. The active site loop (residues 85–94) is colored orange and red for copies 1 and 2, respectively. The Ubc6-Ub linkage and residue His94 are shown as sticks. The two structures differ markedly in the conformation of the active site proximal region (Lys87-Trp98). Importantly, in the second copy, His94 points away from the ubiquitin attachment site and is involved in crystal packing contacts (not shown). Furthermore, the ubiquitin tails and the relative orientation of ubiquitin towards the UBC domain differ, so that in the second copy, ubiquitin adopts a more "open" conformation (distance $C_\alpha Ser113_{Ubc6}$ – $C_\alpha Ile44_{Ub}$ 6.8 Å and 9.0 Å for copies 1 and 2, respectively). (**C**) Electron density map (2Fo-Fc, contoured at the 1.0 σ level) for the active site loop (Arg85-His94, in orange) and the ubiquitin tail of the first copy (Leu71-Gly76, in light blue) (PDB: 9EN5). (**D**) The two-dimensional [$^{15}$N,$^{1}$H]-HSQC spectrum of ubiquitin. The assignments are indicated by the corresponding number along the protein primary sequence followed by the one-letter amino acid code. (**E**) Plot of rotational correlation time ($\tau_c$) of Ubc6$_{WT}$-($^{15}$N)Ub and Ubc6$_{C89A}$-($^{15}$N)Ub in blue and red, respectively. For comparison $\tau_c$ for free ($^{15}$N)Ub is shown in gray. Ubc6 and ubiquitin were coupled by a disulfide bond using the ubiquitin mutant G76C. The x-axis denotes residue numbering in ubiquitin.

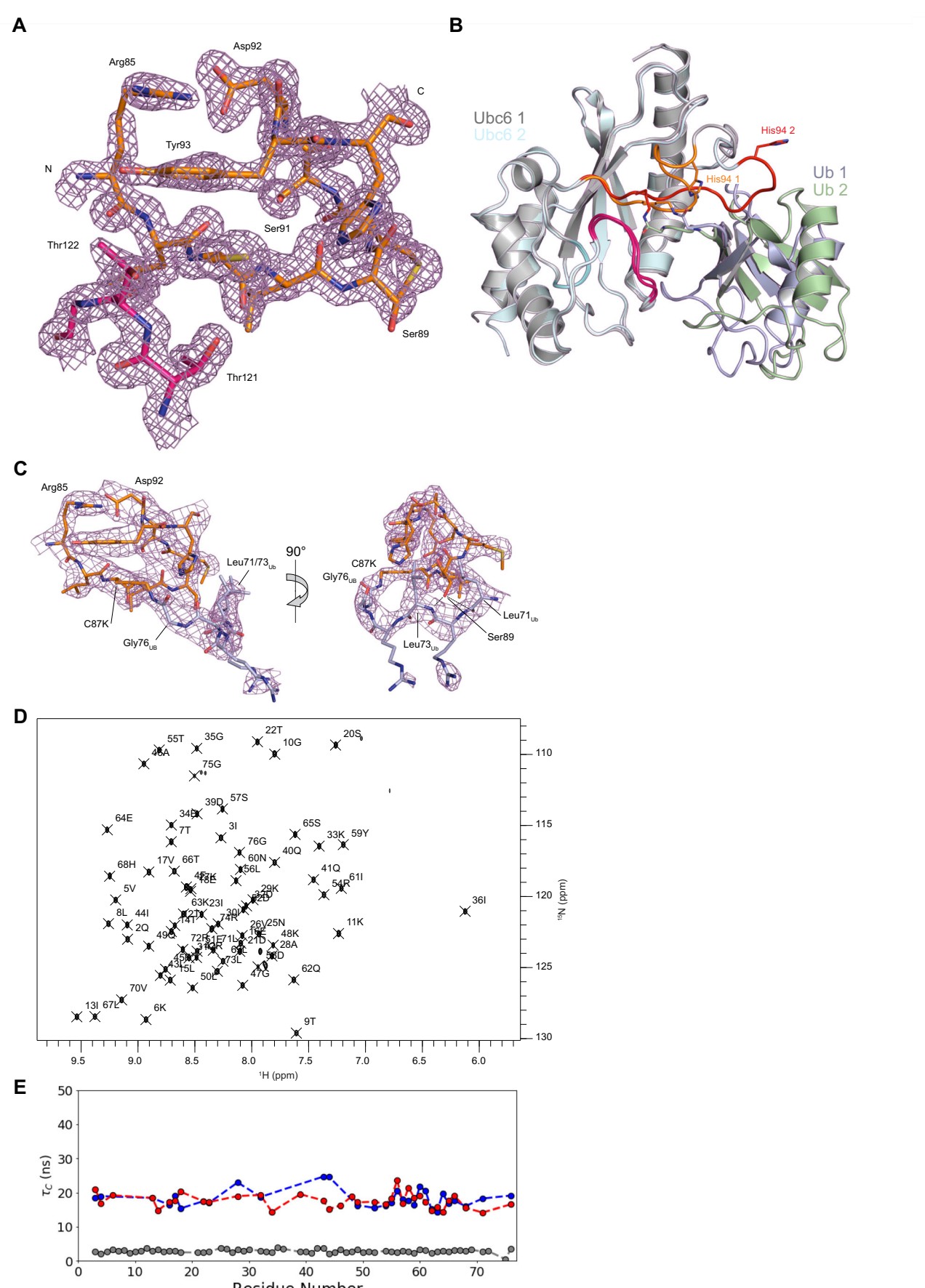

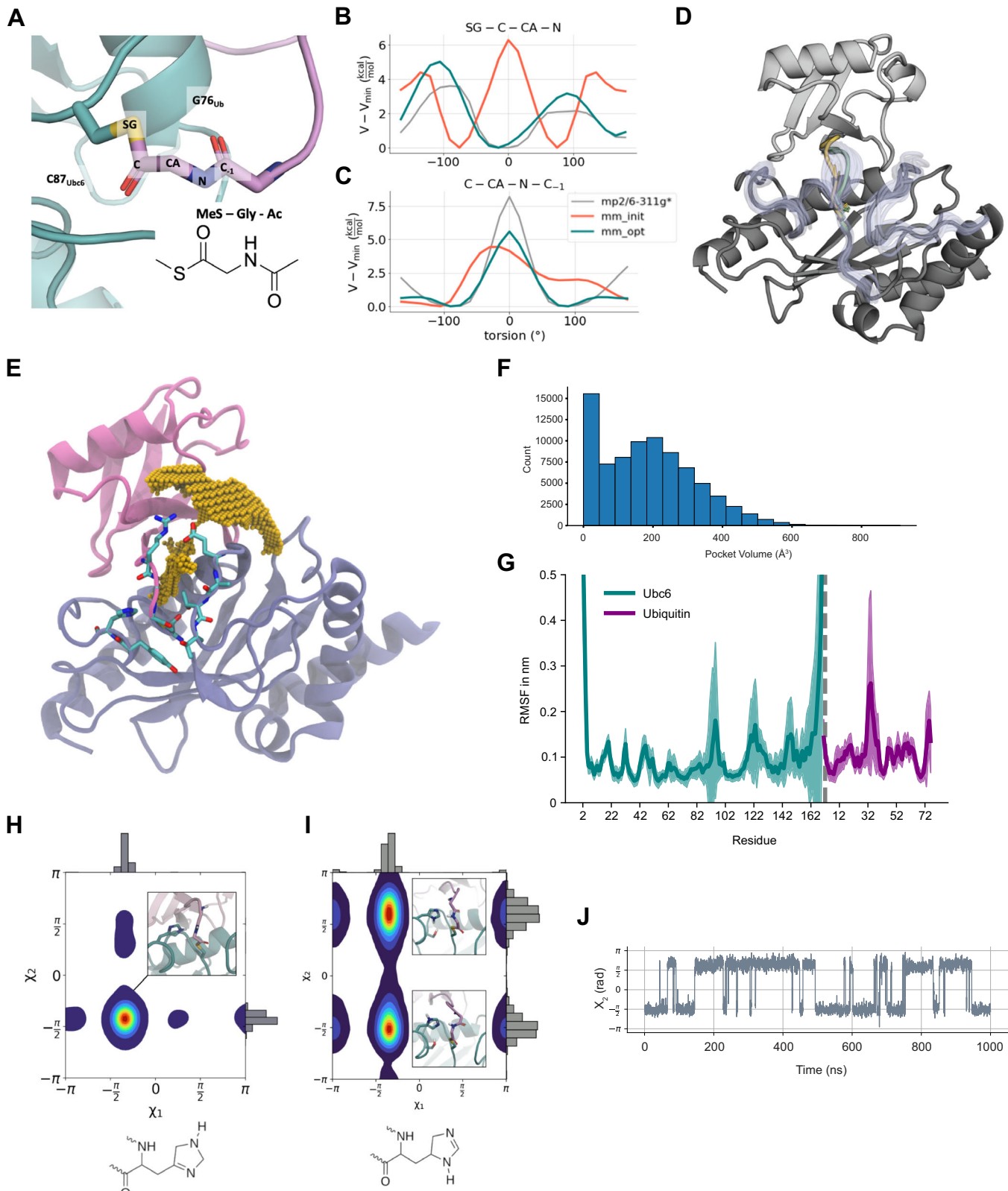

◀

**Figure EV2. Molecular dynamics simulations of Ubc6-Ub.**

(A) Structure of the $C87_{Ubc6}$-$G76_{Ub}$ thioester bond. The residues involved are shown as sticks, the rest of the complex is shown in cartoon representation. The inset shows the capped glycine used for assignment and parameterization of the force field parameters. (B, C) Optimization of force field parameters. (B) Dihedral scans of the SG-C-CA-N ($\Psi$) torsion. QM potential is shown in gray, the initial force field based potential is shown in red and the optimized potential is shown in teal. All potentials were zeroed on the minimum. (C) Dihedral scans of the C-CA-N-$C_{-1}$ ($\phi$) torsion. Plot structure is the same as (B). (D) Cartoon representation of the average structure of the native Ubc6-Ub complex obtained by the simulated annealing procedure. Ubc6 is colored in dark gray, ubiquitin in light gray. The isopeptide-linked Ub-tail from the crystal structure is depicted in yellow, the thioester-linked tail in green. For the parts of the model that were not restrained during simulated annealing (Ubc6 residues 79–98 and 115-128; ubiquitin residues 71–76), every tenth configuration is overlaid in transparent blue. (E) A solvent-filled cavity is present near the active site of the Ubc6-Ub complex. Key residues involved in the formation of this cavity are represented as cyan sticks; the rest of Ubc6 and ubiquitin colored in blue and pink, respectively. Small yellow spheres outline the pocket volume, averaged over all configurations generated during simulated annealing. (F) Volume of the solvent-filled cavity enclosed by the C-terminal tail of ubiquitin. The average volume was 304 s.d. 219 $\text{Å}^3$. (G) Root mean square fluctuation (RMSF) of $C_\alpha$ atoms of the Ubc6 (teal) - ubiquitin (purple) complex. The solid line indicates the average of $100 \times 1\,\mu s$ simulations, the shaded area indicates the mean RMSF $\pm 2\sigma$. (H) Contour plot showing the distribution of the $\chi_1$ (N-$C_\alpha$-$C_\beta$-$C_\gamma$) and $\chi_2$ ($C_\alpha$-$C_\beta$-$C_\gamma$-$N_\delta$) dihedral angles of His94 with the $\epsilon$-nitrogen protonated. Each contour level corresponds to 10% of the probability density. The inset plot shows the major configuration ($\chi_1$: -64° $\chi_2$: -84°). The $\epsilon$-protonated nitrogen appears to be incompatible with His94 acting as a base, because its imidazole ring was largely confined to a conformation in which the free electron pair of the $\delta$-nitrogen points away from the region from which a nucleophilic attack would occur. For visualization purposes, 200,000 random samples of the $\chi_1$ and $\chi_2$ values were calculated from the simulation data. Thereafter each sample was incremented by a value chosen at random from $[-2\pi, 0, 2\pi]$; this abrogated the hard boundaries at $-\pi$ and $\pi$. The density was estimated from the resampled data using a Gaussian kernel density estimator. (I) Contour plot showing the distribution of the $\chi_1$ and $\chi_2$ dihedral angles of His94 with the $\delta$-nitrogen protonated. Each contour level corresponds to 10% of the probability density. The inset plots depict the two major conformations ($\chi_1$: -64° $\chi_2$: -88°) and ($\chi_1$: -63° $\chi_2$: 109°), that appeared with similar frequencies. 52% of the structures adopted an $N_\epsilon$ outward-facing orientation, 40% adopted the inward-facing orientation, compatible with a base function for His94. An additional conformation, accounting for 8% of the observed structures, exhibited a $\chi_1$ dihedral angle rotated by 180°. The histograms at the margins show the distributions of each dihedral angle. The dihedral angles were resampled as described in (H). (J) Exemplary time trace of the $\chi_2$ dihedral angles from simulations of the HisD tautomer. The inward and outward-facing conformations interconverted rapidly during simulations and are only separated by an energy barrier of approximately 1 $k_B T$.

**A**

WT
S89A
H94A
H94Q

Fraction Ubc6 loaded

Time (min)

**B**

| Sbh2 | | | | | | | 4KR | | | | | | | | | | | | | |
|---|---|---|---|---|---|---|---|---|---|---|---|---|---|---|---|---|---|---|---|---|
| E2 | Ubc6 | | | | | | Ubc7 | | | | | | Ubc6 & 7 | | | | | | |
| Time (min) | 0 | 2 | 5 | 10 | 30 | 60 | 0 | 2 | 5 | 10 | 30 | 60 | 0 | 2 | 5 | 10 | 30 | 60 |

170
130
90
55
43
35
25
15

*
Sbh2-Ub_n
Sbh2-Ub_3
Sbh2-Ub_3
Sbh2-Ub_2
Sbh2-Ub_1
Sbh2

**C**

Ubc6

WT
S89A
H94A

Sbh2 WT

Sbh2 4KR

Fraction Sbh2 ubiquitinated

Time (min)

**D**

| ATP | − | + | |
|---|---|---|---|
| NaOH | − | | + |

Sbh2-Ub_n
*
170
130
90
55
43
35
25
15
Sbh2

**E**

| Sbh2 | WT | | | | 4KS | | | WT | | | 4KR | | |
|---|---|---|---|---|---|---|---|---|---|---|---|---|---|
| Ubc6 | WT | | | | WT | | | H94Q | | | WT | | |
| ATP | − | + | | | − | + | | − | + | | − | + | |
| NaOH | − | | + | | − | | + | − | | + | − | | + |

170
130
90
55
43
35
25
15

Sbh2-Ub_n
Sbh2-Ub_3
Sbh2-Ub_2
Sbh2-Ub_1
Sbh2
*

**F**

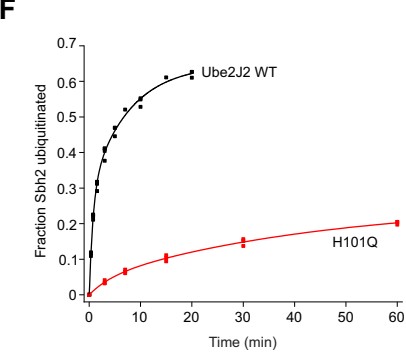

Ube2J2 WT

H101Q

Fraction Sbh2 ubiquitinated

Time (min)

**Figure EV3.   Reconstitution of Sbh2 ubiquitination.**

(A) E1-mediated Ub loading of full-length WT Ubc6 is unaffected in the indicated mutants. Detergent solubilized WT Ubc6 or the indicated mutants were incubated with E1, Ub and ATP. Time course of E1-mediated loading of fluorescently labeled WT Ubc6 and mutants were analyzed by non-reducing SDS-PAGE and detected by fluorescence scanning. Values for each time point are shown as colored symbols connected by solid lines. (B) Time course of ubiquitination of Sbh2 4KR. Fluorescently labeled Sbh2 4KR was co-reconstituted with Doa10. In addition, the indicated E2s/cofactors were co-reconstituted. Samples were analyzed by SDS-PAGE and fluorescence scanning. The asterisk marks a band that appears upon co-reconstitution of Doa10 with fluorescently labeled Sbh2. It probably reflects a partially SDS-resistant complex of the two proteins. (C) Quantification of Sbh2 and Sbh2 4KR ubiquitination in the presence of Doa10 and either WT Ubc6 or Ubc6 mutants S89A and H94A. Data points and error bars indicate mean ± one standard deviation from three experiments. Black for WT Sbh2, yellow for Sbh2 4KR. Solid, dashed and dotted lines for WT Ubc6, Ubc6 S89A, and Ubc6 H94A, respectively. Data for WT Ubc6 is reproduced here from Fig. 4E. (D, E) Representative SDS-PAGE for determining NaOH-resistant and -sensitive Sbh2 ubiquitinations from reactions with either WT Sbh2, Sbh2 4KR, or Sbh2 4KS, co-reconstituted with Doa10 and either Ubc7/Cue1 (D) or the indicated Ubc6 version (E). Samples were collected after 1 h from reactions as in Fig. 4D. Where indicated, samples were treated with NaOH to preserve only lysine modifications. Reactions lacking ATP serve as controls. Samples were analyzed by SDS-PAGE and fluorescence scanning. As in (B), the asterisk marks a band that appears upon co-reconstitution of Doa10 with fluorescently labeled Sbh2. (F) Time course of Sbh2 ubiquitination in liposomes containing Sbh2, Doa10 and the H101Q mutant of the Ube2J2/Ubc6 chimera. Data for the non-mutated chimera is reproduced from Fig. 4J for comparison. For visualization, double-exponential fits to the data are shown as solid lines. $N = 3$.

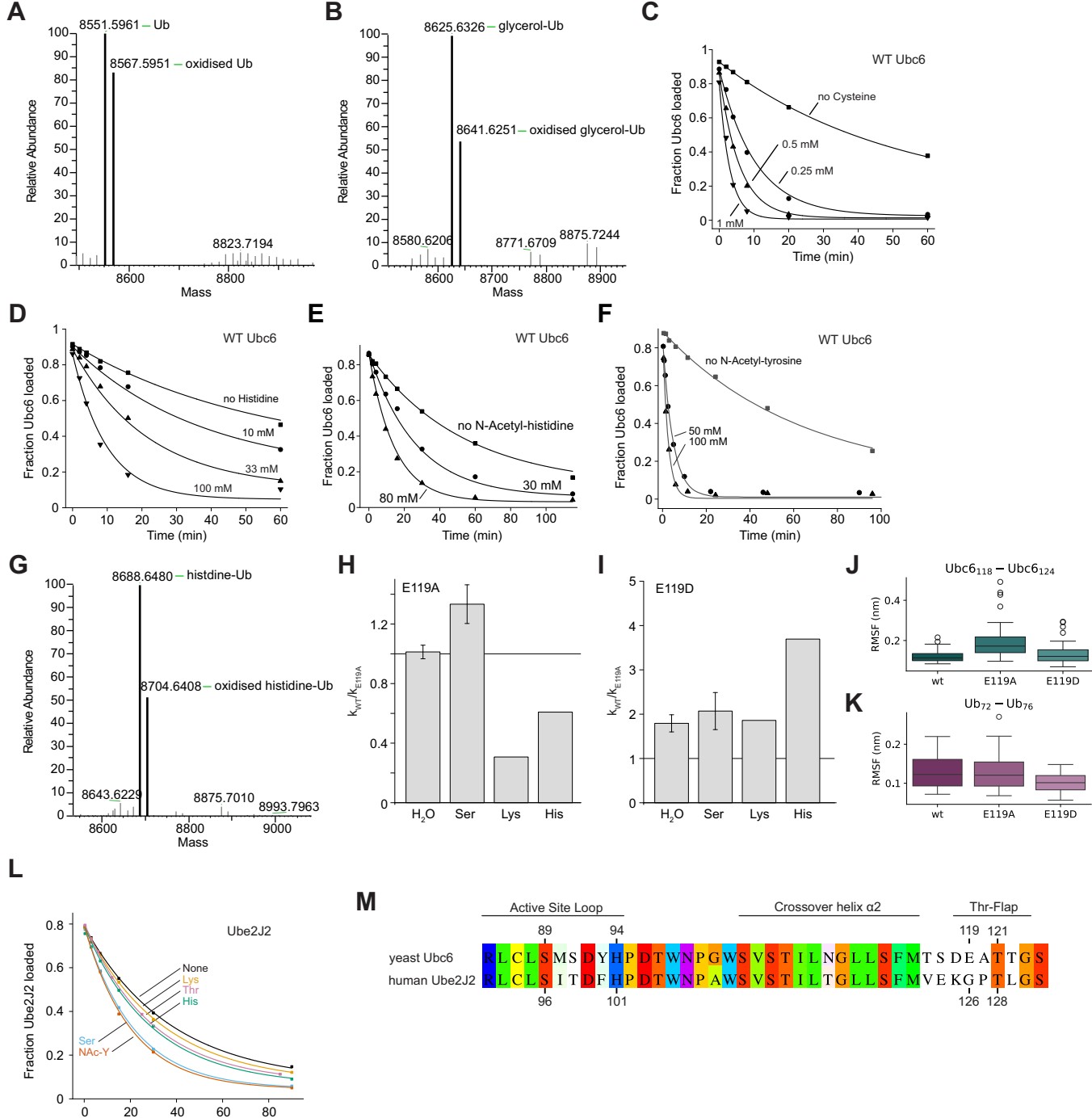

◀ **Figure EV4. Discharge of Ubc6-Ub and J2-Ub by small nucleophiles.**

(A, B) Deconvoluted mass spectra of yeast ubiquitin (A) (mass 8551.5960 Da, reference 8551.5747 Da) and glycerol-ubiquitin (B) (mass 8625.6326 Da, reference 8625.6115 Da). The mass difference of 74 Da corresponds to the mass of glycerol minus one water molecule. (C–F) Ubiquitin discharge assays of WT Ubc6 in the presence of the indicated concentrations of free amino acids. (C) Cysteine, (D) histidine, (E) N-acetyl-histidine, (F) N-acetyl-tyrosine (NAc-Tyr). The fraction of the loaded and discharged state were quantified by densitometry. In each panel, the data were globally fitted to determine rate constants for hydrolysis, and discharge by the indicated free amino acid. Fitting results are shown solid lines; the obtained values for histidine and NAc-Tyr are reported in Fig. 5E. (G) Deconvoluted mass spectrum of histidine-ubiquitin (mass 8688.6482 Da, reference 8688.6336 Da). (H, I) Bar plot comparing reactivity of the Ubc6 mutant E119A (H) or E119D (I) with WT Ubc6 towards the indicated nucleophiles. Discharge by hydrolysis is indicated as "$H_2O$". Rate constants of mutants for different nucleophiles were determined as in Fig. 5D and averages determined as described for Fig. 5E. Plotted are fold differences of WT/mutant discharge rates. N for the E119A mutant towards the nucleophiles "$H_2O$", serine, lysine, and histidine is three, two, one, and one, respectively. For the E119D it is three, three, one, and one, respectively. Error bars represent standards deviations as described for Fig. 5F–I. As only single measurements were performed in the presence of free histidine or lysine, these values have no error bars. (J) Boxplot comparing the root mean square fluctuation (RMSF) of the $Ubc6_{118}$ - $Ubc6_{124}$ loop between WT Ubc6 and the E119A and E119D mutants. The median and interquartile range between the 25th and 75th percentiles is shown by the boxed area. The center line shows the median. The whiskers extend to all data points within 1.5 times the interquartile range, while all additional points beyond this range are shown as dots. For each variant, 8 simulations were used to calculate the RMSF values of the 7 residues ($n = 56$). (K) Boxplot comparing the RMSF of the ubiquitin tail (residues 72-76) between WT Ubc6 and the mutants E119A and E119D. The structure of the boxplots is identical to (J). For each variant, 8 simulations were used to calculate the RMSF values of the 5 residues ($n = 40$). (L) Ubiquitin discharge assay with the UBC domain of human Ube2J2 in the presence of 50 mM of the indicated nucleophiles. Samples taken at the indicated time points were analyzed by non-reducing SDS-PAGE and stain-free imaging. Plots of fractions of Ub-loaded Ube2J2 were globally fitted. Fitting results are shown as solid lines and values are reported in Fig. 5M. (M) Pair-wise alignment of yeast Ubc6 and human Ube2J2 in the region covering the active site loop and the Thr-flap. Source data are available online for this figure.

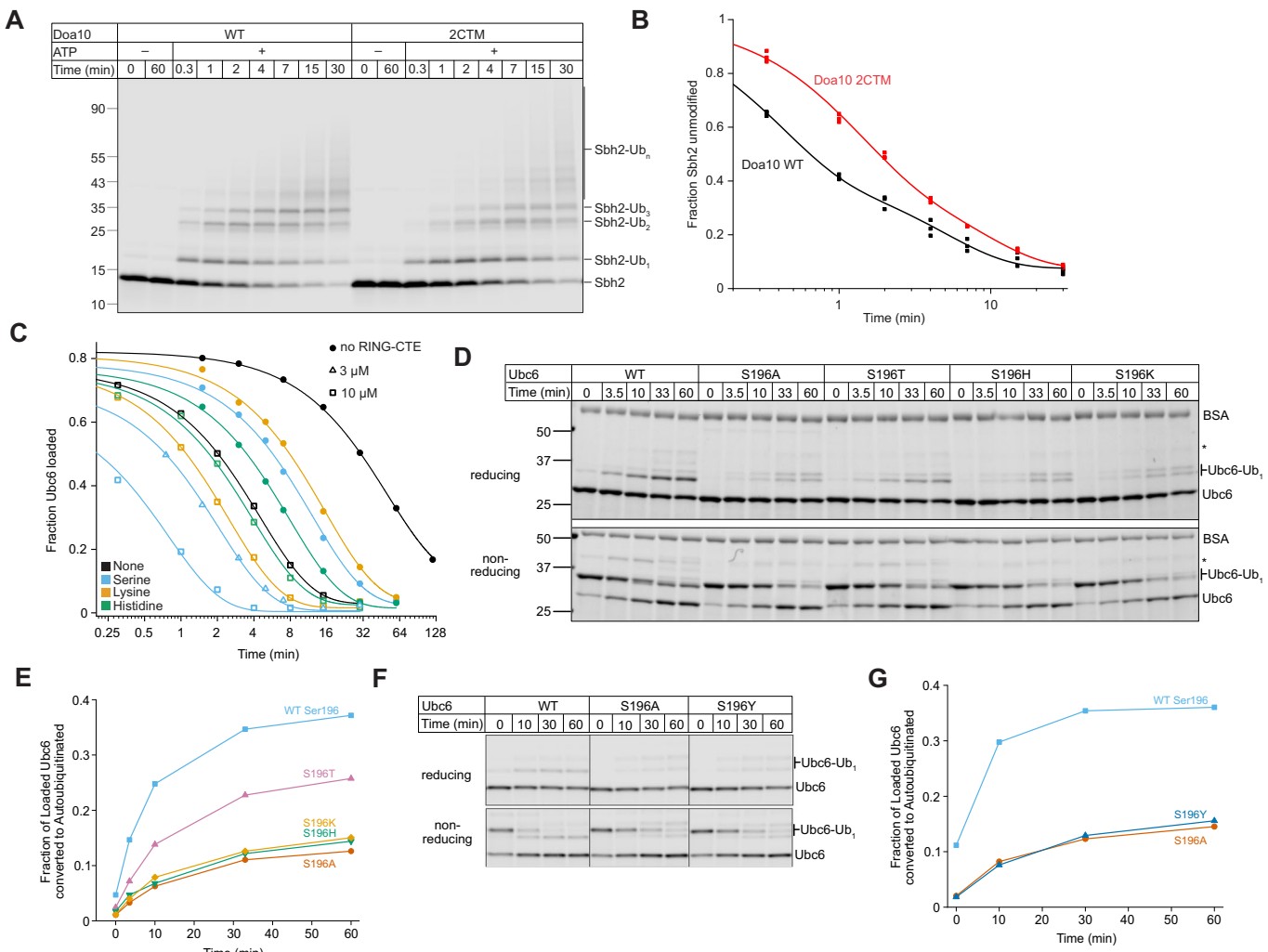

**Figure EV5. RING-effect on Ubc6 and Ube2J2 activity.**

(A) Full-length WT Doa10 is more reactive in Sbh2 ubiquitination than its CTE mutant. Fluorescently labeled Sbh2 was co-reconstituted with Ubc6 and either WT Doa10 or the Doa10 2CTM mutant (G1308L, N1314A). Proteoliposomes were incubated with 1 μM Uba1, 120 μM ubiquitin, and ATP. Samples taken at the indicated time points were analyzed by SDS-PAGE and fluorescence scanning. Indicated samples lacked ATP as a negative control. (B) Quantification of the fraction of unmodified Sbh2 from three experiments as in (A). Solid lines represent global double-exponential fits of the data. (C) Ubiquitin discharge assays performed with WT UBC domain of Ubc6 in the presence of 50 mM of the indicated free amino acids and the indicated Doa10 RING-CTE concentrations. Plots of the fraction of loaded Ubc6 were globally fitted to a mono-exponential function to determine apparent rate constants for hydrolysis and discharge by free amino acids. Solid lines represent fit results. Such data was used to generate the plot in Fig. 6D. (D) Autoubiquitination assay comparing full-length WT Ubc6 and its indicated Ser196 point mutants. A discharge assay with full-length Ubc6 was performed in the presence of the detergent n-dodecyl-β-D-maltoside. After the loading step, EDTA was added to quench E1 activity. Samples collected at indicated time points were analyzed by reducing (top) and non-reducing (bottom) SDS-PAGE and stain-free imaging. The asterisk indicates a band that probably arises from a small fraction of Ubc6-Ub that is autoubiquitinated and Ub-loaded, and that partially converts into a double autoubiquitinated species. (E) Quantification of (D). The fraction of autoubiquitinated Ubc6 was determined by densitometry from the reducing gel and normalized to the loaded fraction at $t = 0$, as determined from the non-reducing gel. (F) As in (D), but using the indicated fluorescently labeled Ubc6 variants. Samples were analyzed by SDS-PAGE and fluorescence scanning. (G) Quantification of (F), as described in (E).

