## [Peer Review File · The EMBO Journal]

Determinants of chemoselectivity in ubiquitination by the J2 family of ubiquitin-conjugating enzymes

Alexander Stein, Anuruti Swarnkar, Florian Leidner, Ashok Rout, Sofia Ainzati, Claudia Schmidt, Stefan Becker, Henning Urlaub, Christian Griesinger, and Helmut Grubmüller

Corresponding author(s): Alexander Stein (alexander.stein@mpinat.mpg.de)

Review Timeline:

Submission Date:	10th May 24
Editorial Decision:	19th Jun 24
Revision Received:	23rd Sep 24
Editorial Decision:	22nd Oct 24
Revision Received:	29th Oct 24
Accepted:	31st Oct 24

Editors: Cornelius Schneider and Hartmut Vodermaier

Transaction Report:

Dear Dr Stein,

Thank you for submitting your manuscript for consideration by the EMBO Journal. We have now received comments from three reviewers, which are included below for your information.

As you will see from the reports, the reviewers appreciate the work, while also indicating a number of constructive points that would need to be addressed before acceptance here. From my side, I find the reviewer comments reasonable. Therefore, based on these positive assessments, I would like to invite you to address the issues raised by the reviewers in a revised manuscript. I would be happy to discuss the revision in more detail via email or phone/videoconferencing.

We generally allow three months as standard revision time. As a matter of policy, competing manuscripts published during this period will not negatively impact on our assessment of the conceptual advance presented by your study. However, please contact me as soon as possible upon publication of any related work to discuss the appropriate course of action. Should you foresee a problem in meeting this three-month deadline, please contact us to arrange an extension.

When preparing your letter of response to the referees' comments, please bear in mind that this will form part of the Review Process File and will therefore be available online to the community. For more details on our Transparent Editorial Process, please visit our website: <https://www.embopress.org/page/journal/14602075/authorguide#transparentprocess>. Please also see the attached instructions for further guidelines on preparation of the revised manuscript.

Please feel free to contact me if you have any further questions regarding the revision. Thank you for the opportunity to consider your work for publication, and I look forward to your revision.

With best regards,

Cornelius

Cornelius Schneider, PhD
Editor
The EMBO Journal
c.schneider@embojournal.org

We realize that it is difficult to revise to a specific deadline. In the interest of protecting the conceptual advance provided by the work, we recommend a revision within 3 months (17th Sep 2024). Please discuss the revision progress ahead of this time with the editor if you require more time to complete the revisions. Use the link below to submit your revision:

Referee #1:

Swarnkar et al. manuscript is focused on the characterization of Ubc6, a yeast ubiquitin conjugating enzyme homologue of the human UBE2J2 ubiquitin conjugating enzyme. The authors produced a crystal structure of Ubc6 loaded with ubiquitin by mutating the catalytic cysteine (C87) to lysine, therefore allowing isopeptide bond formation between the ubiquitin C terminus and the lysine 87 side chain. The structure presents the Ubc6-Ub folded in the so called "closed" conformation in which the E2 interacts with the ubiquitin I44 hydrophobic patch. The authors identify S89 as important residue for the stability of the closed conformation state and also a new feature, the Thr flap, which overall interact and stabilizes the Ub-C terminus. Similarly, the authors identified His94 as an important residues for Ubc6 activity. The authors suggest that the Thr-flap plays a role similar to that of the conserved HPN motif in other E2s. The authors proceed to validate the role of these residues by reconstitute the Ubc6 activity in vitro either by discharge assay, ubiquitination of the Sbh2 substrate or by autoubiquitination assay. Furthermore, the authors study the impact of the E3 cognate Doa1 on Ubc6 activity and demonstrate that Doa1 sharpens the reactivity of Ubc6 toward serine but not lysine.

One of the interesting findings from the paper is that, even in the absence of an E3 ligase the Ubiquitin loaded Ubc6 is in the "closed" conformation which is thought to be activated for ubiquitin transfer. While ubiquitin loaded Ubc6 is indeed active in E3 independent ubiquitin transfer it is very substantially activated by the presence of the Doa 10 RING E3 ligase (Fig. 6A-C). This then begs the question of how the E3 is further activating the thioester bond between ubiquitin and E2. Aside from suggesting that S89 might play a role in stabilising the ubiquitin C terminal tail the discussion of this issue is limited, although it is an important point. This point is also relevant to the section indicating that "RING E3 binding sharpens up the substrate profile towards OH-groups". While the data clearly shows that the RING enhances serine ubiquitination there is no real discussion on how structural alterations induced by the E3 might contribute to this effect. While obtaining a high-resolution structure for the ubiquitin loaded Ubc6 bound to the Doa 10 RING is clearly outwith the scope of this paper, it would be useful to have a more detailed analysis of the RING induced activity, particularly as it relates to substrate ubiquitination. As indicated in Suppl. Fig. 7K the RING enhances modification of Sbh2 serine residues to a greater extent than Sbh2 lysine residues. If these reactions could be done under single turnover conditions it would be possible to determine if this was an effect on Km or on kcat. This might allow conclusions to be drawn on whether the RING was enhancing binding (Km) or altering catalysis (kcat).

Overall, the paper is well-written, it presents robust dataset and introduces incremental novelty with the first Ubc6-Ubiquitin loaded structure, as well as highlighting the role of the Thr flap. The paper discusses other significant aspects, which closely reflect previous literature on UBE2J2 (see Abdul Rehman SA et al., *Sci Adv.* 2024, doi: 10.1126/sciadv.adh0123), such as the central role of His 94 (which should corresponding to H101 in UBE2J2) and the preference for serine over threonine as a non-canonical substrate. The paper is of good quality and should be suitable for publication once the following points are addressed:

Title: The title is slightly misleading as it refers to the human J-family of ubiquitin-conjugating enzymes. While there is homology between Ubc6 and Ube2J1/J2, the authors do not present any data on the J family itself. Thus, there is no proof that all of the interesting aspects highlighted for Ubc6 translate to UBE2J1/J2. Therefore, I recommend changing the title to something more reflective of the content of the paper, such as "Determinants of Chemoselectivity in Ubc6."

2. Line 63: What is the CPY* substrate? Please provide some background information if it is mentioned.
3. Line 66-68: The sentence starting with "Furthermore, yeast Ubc..." is quite convoluted and needs to be rephrased for clarity.
4. Line 77: Similar to CPY*, "so called CES/D site" short background information is needed here as well.
5. Figure 3. It seems there is a mismatch between the Western Blot (Figure 3A) and the relative graph (Figure 3B). For instance, H94Q appears more stable than H94A; however, on the quantification graph, they look quite alike. Also, H94A is less stable than S94A, which is not reflected in the graph. There may be a labelling error and that the blue line in Figure 3A corresponds to H94A, while the green line corresponds to S89A.
6. Line 275-277 and 292-294: It is worth mentioning that the preference of Ubc6 for serine over threonine mirrors that of UBE2J2 (see Abdul Rehman SA et al., *Sci Adv.* 2024, doi: 10.1126/sciadv.adh0123).

Referee #2:

This is an exceptionally thorough and beautifully written study that determines the biochemical mechanism of hydroxy group ubiquitination by the Ubc6 enzyme. There is a lot of interest and excitement in the discovery of non-amine and non-protein ubiquitination, and this study is one of the first to delve into how, at the chemical level, these processes are achieved.

I do not have any concerns about the data as presented (though there is one statement referring to determining the rate constant as data not shown -line 268- is there any practical reason why those data cannot be included in supplementary information?). The results and rationale for each experiment are clearly described, this has been carefully written, with care taken to indicate species, construct, conditions used in each experimental setup.

The only real curiosity I have is what does that second copy of the Ubc6-isopeptide conjugate look like in the relevant region, and is it worth commenting on beyond attributing it to crystal packing?

A few tiny typos - line 114 should include a citation to the published structure
line 152 refers to supporting figures, should be consistent with supplementary as used throughout
line 209 needs a hyphen between ubc6 and containing, otherwise it looks like the protein contains the liposomes.

I really enjoyed reading it, and think it is a great mechanistic study, focusing on the actual molecular details of an important reaction, with careful, well-designed and well-executed experiments.

Referee #3:

The J subfamily of E2 ubiquitin-conjugating enzymes, such as yeast Ubc6, are known to ubiquitinate protein serine side chains, unlike most ubiquitination enzymes, which largely ubiquitinate lysine side chains and sometimes N-terminal primary amines. Swarner et al. perform a detailed analysis of how yeast Ubc6 is capable of this unusual 'chemoselectivity'. They employ an impressive array of biophysical and biochemical methods, including high-resolution X-ray crystallography (ranging from 1.2 Å resolution for the apo-E2 to 2.6 Å for the ubiquitin-charged E2 (linked by an isopeptide bond); NMR, MD simulations, mass spectrometry, complex biochemical reconstitutions, and even some in cellulose degradation assays.

The J E2s were known to lack several key features of canonical E2s such as the HPN motif. The authors here show how the functionality of these motifs is replaced by several others in Ubc6 (the 'Thr flap' and 'GRF loop'). These replacements rewire the active site so that the carbonyl carbon of the ubiquitin thioester with Ubc6 is more electrophilic, allowing attack by weaker nucleophiles such as hydroxyls. Additionally, a conserved histidine in this E2 subfamily, His94 in Ubc6, functions as a general base in promoting attack by hydroxyl nucleophiles.

These are well-executed experiments with very strong data. The paper is well written and careful in its interpretations. As someone very interested in such noncanonical ubiquitination mechanisms, I found this to be an outstanding study. Despite my highly positive view, two issues will need to be considered by the editor regarding the paper's suitability for the journal: 1) is such a detailed in vitro analysis to characterize a subfamily of E2s of broad enough interest to the EMBO Journal readership and 2) does the recent work from Ref. 30 (Abdul Rehman et al., Sci Adv 2024) detract from its novelty. The later paper analyzed both a J subfamily E2 (UBE2J2) and a novel E2 subfamily (UBE2Q1/Q2) that also ubiquitinates hydroxyl groups in proteins and small molecules such as glycerol and glucose, and their protein structure modeling also implicated the closed conformation of the E2 and a unique His residue (H94 equivalent for the J E2) in the catalytic mechanisms. I believe the current work is an unusually thorough and clean story and one that diverges sufficiently in its mechanistic analysis from that of Abdul Rehman et al., so as implied above, I am personally in favor of publication.

Minor comments:

Line 233: Looks to my eye like 4KS is still (weakly) modified by Ubc7/Cue1 alone in Fig. 4D, unlike what is stated here.

As pointed out in the Discussion, the evolutionary advantage of the expanded chemoselectivity of the J subfamily of E2s (and other E2s such as UBE2Q1/Q2) is unknown. An interesting speculation suggested was that since some J E2s function in ERAD, the expanded selectivity might allow a broader array of substrates to be ubiquitinated. I wondered if there were organismal lineages that lacked any hydroxyl-selective E2s. I know that at least the ERAD-linked E3 Doa10/MARCHF6 is missing from some lineages but am curious to see how this matches up with the J E2s. Not essential, but it might be interesting to look at.

Dear Editor, dear Referees,

We would like to thank the editor and the referees for time and effort spent on evaluating our manuscript and their encouraging and thoughtful comments. We have performed additional experiments and revised the manuscript to address concerns and suggestions raised during the review. Below, we provide a point-by-point response to the referees' comments (our response in blue).

Referee #1:

Swarnkar et al. manuscript is focused on the characterization of Ubc6, a yeast ubiquitin conjugating enzyme homologue of the human UBE2J2 ubiquitin conjugating enzyme. The authors produced a crystal structure of Ubc6 loaded with ubiquitin by mutating the catalytic cysteine (C87) to lysine, therefore allowing isopeptide bond formation between the ubiquitin C terminus and the lysine 87 side chain. The structure presents the Ubc6-Ub folded in the so called "closed" conformation in which the E2 interacts with the ubiquitin I44 hydrophobic patch. The authors identify S89 as important residue for the stability of the closed conformation state and also a new feature, the Thr flap, which overall interact and stabilizes the Ub-C terminus. Similarly, the authors identified His94 as an important residues for Ubc6 activity. The authors suggest that the Thr-flap plays a role similar to that of the conserved HPN motif in other E2s. The authors proceed to validate the role of these residues by reconstitute the Ubc6 activity in vitro either by discharge assay, ubiquitination of the Sbh2 substrate or by autoubiquitination assay. Furthermore, the authors study the impact of the E3 cognate Doa1 on Ubc6 activity and demonstrate that Doa1 sharpens the reactivity of Ubc6 toward serine but not lysine.

One of the interesting findings from the paper is that, even in the absence of an E3 ligase the Ubiquitin loaded Ubc6 is in the "closed" conformation which is thought to be activated for ubiquitin transfer. While ubiquitin loaded Ubc6 is indeed active in E3 independent ubiquitin transfer it is very substantially activated by the presence of the Doa10 RING E3 ligase (Fig. 6A-C). This then begs the question of how the E3 is further activating the thioester bond between ubiquitin and E2. Aside from suggesting that S89 might play a role in stabilising the ubiquitin C terminal tail the discussion of this issue is limited, although it is an important point. This point is also relevant to the section indicating that "RING E3 binding sharpens up the substrate profile towards OH-groups". While the data clearly shows that the RING enhances serine ubiquitination there is no real discussion on how structural alterations induced by the E3 might contribute to this effect. While obtaining a high-resolution structure for the ubiquitin loaded Ubc6 bound to the Doa 10 RING is clearly outwith the scope of this paper, it would be useful to have a more detailed analysis of the RING induced activity, particularly as it relates to substrate ubiquitination. As indicated in Suppl. Fig. 7K the RING enhances modification of Sbh2 serine residues to a greater extent than Sbh2 lysine residues. If these reactions could be done under single turnover conditions it would be possible to determine if this was an effect on K_m or on k_{cat} . This might allow conclusions to be drawn on whether the RING was enhancing binding (K_m) or altering catalysis (k_{cat}).

How RING E3s precisely enhance ubiquitin transfer is a longstanding question in this field. As the reviewer criticizes, we provide limited additional insight to this question. Indeed, we have tried to obtain a structure of yeast Ubc6 with the Doa10 RING domain and of Ube2J2 with the March6 RING domain. Unfortunately, these attempts were unsuccessful. In line with recent single molecule data by the Hay lab, our data on Ubc6 confirms that closed E2-Ub conformations are not sufficient (Branigan et al., 2020,

Nat. Commun.). Based on our mutagenesis we propose that RING binding also confines other flexible regions to more reactive conformations, namely the ubiquitin tail. Furthermore, we speculate that RING binding would also restrict conformational flexibility of the active site loop, that is indicated by the observations that (1) this region is unresolved in the Ube2J2 structure, (2) adopts an alternative conformation in the second copy of Ubc6-Ub in our crystal structure (Fig EV1B), and (3) our MD simulations (new Fig EV2G). We discuss this question now in more detail in the revised version in the paragraph starting with line 494.

With respect to the suggested experiment to determine K_m and k_{cat} for reactions with serine or lysine: Single turnover assays are technically difficult in liposome-based assays. Such an assay requires purification of ubiquitin-loaded Ubc6, followed by co-reconstitution into liposomes with Doa10 and Sbh2. However, during the reconstitution procedure, Ubc6-Ub either hydrolyses or undergoes autoubiquitination, which precludes this approach. Alternatively, we attempted to use a soluble version of purified Ubc6-Ub (as also used in Fig 6AB). However, adding soluble Ubc6-Ub to Doa10/Sbh2-liposomes results in much slower turnover of Sbh2 even at high concentrations of soluble Ubc6-Ub, presumably because a soluble version shows a lower affinity to Doa10. However, the bigger problem is that determination of k_{cat} and K_m require experimental conditions, under which the enzyme is saturated or nearly saturated with substrate. Unfortunately, this is technically impossible since high protein/lipid ratios in liposome reconstitutions lead to aggregation during the detergent removal process.

More generally, assays with a substrate like Sbh2, in which several residues can be ubiquitinated (serines, threonines, lysines), are in our opinion not suitable to answer this question. As we explain in the paragraph starting with line 290, several contributing factors need to be accounted for, such as the relative frequency of the individual amino acids and how well they are presented to the active site by the E3 ligase, i.e. the position in the primary sequence. Precisely for this reason, we resorted to discharge assays with free amino acids. With these assays, we were able to show that the RING-bound state clearly prefers Ub-transfer to free serine over transfer to free lysine. However, also in this case it is difficult to discern, whether the stability of the Michaelis complex (E2-Ub : free amino acid) or the ensuing chemical reactions are different for different nucleophiles. While several hundred mM of free amino acid (serine or lysine) is technically achievable, we suspect that such high concentration interfere with the interaction of the RING domain with Ubc6-Ub (see Appendix Figure S6C). This certainly seems to be the case for the Ube2J2/MARCHF6 pair, since we observe slower discharge rates in presence of free lysine than in its absence (see new Appendix Fig S6I). For these reasons, we are unfortunately unable to provide a definitive answer to this question.

Overall, the paper is well-written, it presents robust dataset and introduces incremental novelty with the first Ubc6-Ubiquitin loaded structure, as well as highlighting the role of the Thr flap. The paper discusses other significant aspects, which closely reflect previous literature on UBE2J2 (see Abdul Rehman SA et al., Sci Adv. 2024, doi: 10.1126/sciadv.adh0123), such as the central role of His 94 (which should corresponding to H101 in UBE2J2) and the preference for serine over threonine as a non-canonical substrate. The paper is of good quality and should be suitable for publication once the following points are addressed:

1. Title: The title is slightly misleading as it refers to the human J-family of ubiquitin-conjugating enzymes. While there is homology between Ubc6 and Ube2J1/J2, the authors do not present any data

on the J family itself. Thus, there is no proof that all of the interesting aspects highlighted for Ubc6 translate to UBE2J1/J2. Therefore, I recommend changing the title to something more reflective of the content of the paper, such as "Determinants of Chemoselectivity in Ubc6."

We thank the reviewer for pointing out this important qualification. We have now included more data on the human ortholog of Ubc6, Ube2J2. We think that this new data justifies the statement, that the proposed mechanism of non-canonical ubiquitination is conserved in the J2 family.

- In the new figures Fig 4I,J we now show that a chimeric version of Ube2J2/Ubc6, in which we replaced the UBC domain of Ubc6 with that of Ube2J2 shows an even stronger preference for hydroxy-group containing residues (Ser/Thr) in Sbh2 ubiquitination. Furthermore, a Ube2J2 mutant equivalent to H94Q (H101Q) also shows a similar reduction in Sbh2 ubiquitination (new Fig EV3F). We chose this approach, because it ensures that ubiquitination is directly comparable (i.e. not a different substrate, or different presentation by a different ligase such as MarchF6). Furthermore, the chimeric nature of the construct ensures that interactions of Ubc6 with Doa10 outside the UBC domain, as reported in the recent structure of Doa10 (Wu et al., Nat Commun. 2024, PMID: 38467638), and as previously indicated in our own work (Schmidt et al., elife 2020, PMID: 32588820) are likely maintained.
- We have added data on mutants of Ube2J2 in signature residues (new Fig 5N-P, Fig EV4M, Fig 6H, Appendix Figure S6I). These results show that Ube2J2 shows a similar dependence on signature motifs as yeast Ubc6.

However, it is correct that these conclusions cannot be extended to the J1 family. Our alignments of homologs of Ube2J1 and Ube2J2 and a phylogenetic tree (new: Appendix Figure S2) show that the split into these two families must have occurred early in the evolution of eukaryotes. Almost all eukaryotic lineages encode both, a J1 and a J2 homologue, with budding and fission yeast but also some animals like fruit fly being exceptions that lost a J1 homologue. The J1 family contains the conserved histidine, but the Thr-flap is not conserved (new: Appendix Figure S3). As the reviewer points out, it is thus possible that there are significant mechanistic differences between the J1 and J2 families. The title and abstract now clearly state that this paper is concerned with the J2 family of E2s. We discuss the question of conservation in the J1 family issue now in a paragraph starting with line 512.

We address the following points together:

2. Line 63: What is the CPY* substrate? Please provide some background information if it is mentioned.
3. Line 66-68: The sentence starting with "Furthermore, yeast Ubc..." is quite convoluted and needs to be rephrased for clarity.
4. Line 77: Similar to CPY*, "so called CES/D site" short background information is needed here as well. *This paragraph has been restructured for improved clarity (lines 68-85). "CPY*" and the "CES/D" site are now explained in the text (lines 72 and 90, respectively)*
5. Figure 3. It seems there is a mismatch between the Western Blot (Figure 3A) and the relative graph (Figure 3B). For instance, H94Q appears more stable than H94A; however, on the quantification graph, they look quite alike. Also, H94A is less stable than S94A, which is not reflected in the graph. There may be a labelling error and that the blue line in Figure 3A corresponds to H94A, while the green line corresponds to S89A.

We have re-checked the data and the analysis. We are certain, that there is no miss-assignment. However, we found a mistake in the average calculations, which we have fixed. In general, the data is noisy, and looking at a single blot might be misleading. The three mutants S89A, H94A and H94Q are similarly stabilized, whereas the phenotype of the T121A mutant weaker (in agreement with other data in the manuscript). As one would probably expect, the C87A mutant shows the strongest stabilization.

6. Line 275-277 and 292-294: It is worth mentioning that the preference of Ubc6 for serine over threonine mirrors that of UBE2J2 (see Abdul Rehman SA et al., Sci Adv. 2024, doi: 10.1126/sciadv.adh0123).

Such a statement has been added (line 371).

Referee #2:

This is an exceptionally thorough and beautifully written study that determines the biochemical mechanism of hydroxy group ubiquitination by the Ubc6 enzyme. There is a lot of interest and excitement in the discovery of non-amine and non-protein ubiquitination, and this study is one of the first to delve into how, at the chemical level, these processes are achieved.

I do not have any concerns about the data as presented (though there is one statement referring to determining the rate constant as data not shown -line 268- is there any practical reason why those data cannot be included in supplementary information?). The results and rationale for each experiment are clearly described, this has been carefully written, with care taken to indicate species, construct, conditions used in each experimental setup.

All determined rate constants are now available as Source Data to Fig 5.

The only real curiosity I have is what does that second copy of the Ubc6-isopeptide conjugate look like in the relevant region, and is it worth commenting on beyond attributing it to crystal packing?

We have expanded our discussion of this question. The structure of the second copy, in comparison to the first copy is shown (and already was shown in the original submission) in Fig EV1B, highlighting the relevant parts. To address the question how dynamic this region is, we provide an additional analysis of the MD simulations that is easier to grasp than the already provided Fig 2E. In the new Fig EV2G, it becomes apparent that the region from Cys87 to Trp98 is quite dynamic, thus providing a rationale for why the corresponding region in the Ube2J2 structure is unresolved and two conformations are captured in the Ubc6-Ub structure. We speculate that RING binding may reduce conformational flexibility in this region to stabilize a reactive conformation (paragraph starting with line 494).

A few tiny typos –

line 114 should include a citation to the published structure

The citation has been added for this statement (line 135). However, it was already cited on the first mention of the Ube2J2 structure in (line 128).

line 152 refers to supporting figures, should be consistent with supplementary as used throughout According to EMBO Journal formatting, we split the supplementary information into EV Figures and Appendix Figures. Referrals in the manuscript text now adhere to this nomenclature.

line 209 needs a hyphen between ubc6 and containing, otherwise it looks like the protein contains the liposomes.

We now use the terminology “Ubc6-liposomes” (line 233).

I really enjoyed reading it, and think it is a great mechanistic study, focusing on the actual molecular details of an important reaction, with careful, well-designed and well-executed experiments.

Referee #3:

The J subfamily of E2 ubiquitin-conjugating enzymes, such as yeast Ubc6, are known to ubiquitinate protein serine side chains, unlike most ubiquitination enzymes, which largely ubiquitinate lysine side chains and sometimes N-terminal primary amines. Swarner et al. perform a detailed analysis of how yeast Ubc6 is capable of this unusual 'chemoselectivity'. They employ an impressive array of biophysical and biochemical methods, including high-resolution X-ray crystallography (ranging from 1.2 Å resolution for the apo-E2 to 2.6 Å for the ubiquitin-charged E2 (linked by an isopeptide bond); NMR, MD simulations, mass spectrometry, complex biochemical reconstitutions, and even some in cellulose degradation assays.

The J E2s were known to lack several key features of canonical E2s such as the HPN motif. The authors here show how the functionality of these motifs is replaced by several others in Ubc6 (the 'Thr flap' and 'GRF loop'). These replacements rewire the active site so that the carbonyl carbon of the ubiquitin thioester with Ubc6 is more electrophilic, allowing attack by weaker nucleophiles such as hydroxyls. Additionally, a conserved histidine in this E2 subfamily, His94 in Ubc6, functions as a general base in promoting attack by hydroxyl nucleophiles.

These are well-executed experiments with very strong data. The paper is well written and careful in its interpretations. As someone very interested in such noncanonical ubiquitination mechanisms, I found this to be an outstanding study. Despite my highly positive view, two issues will need to be considered by the editor regarding the paper's suitability for the journal: 1) is such a detailed in vitro analysis to characterize a subfamily of E2s of broad enough interest to the EMBO Journal readership and 2) does the recent work from Ref. 30 (Abdul Rehman et al., Sci Adv 2024) detract from its novelty. The later paper analyzed both a J subfamily E2 (UBE2J2) and a novel E2 subfamily (UBE2Q1/Q2) that also ubiquitinates hydroxyl groups in proteins and small molecules such as glycerol and glucose, and their protein structure modeling also implicated the closed conformation of the E2 and a unique His residue (H94 equivalent for the J E2) in the catalytic mechanisms. I believe the current work is an unusually thorough and clean story and one that diverges sufficiently in its mechanistic analysis from that of Abdul Rehman et al., so as implied above, I am personally in favor of publication.

Minor comments:

Line 233: Looks to my eye like 4KS is still (weakly) modified by Ubc7/Cue1 alone in Fig. 4D, unlike what is

stated here.

Upon co-reconstitution of fluorescently labeled Sbh2 with Doa10, a weak labeled band appears, roughly the size of Doa10. We think that this band is an artifact from the formation of some SDS-resistant co-migration of the two proteins. While boiling of samples reduces the intensity of this band, it also leads to some aggregation of Sbh2, complicating the densitometric analysis. To account for the contribution this labeled band in densitometric analysis, we subtracted the intensity measured in samples without ATP from the signal determined at each time point, leading to values reported in Fig 4F. The same procedure was used for WT Sbh2 and Sbh2 4KS. We have updated the figure legend and the method section to draw attention to this complication.

As pointed out in the Discussion, the evolutionary advantage of the expanded chemoselectivity of the J subfamily of E2s (and other E2s such as UBE2Q1/Q2) is unknown. An interesting speculation suggested was that since some J E2s function in ERAD, the expanded selectivity might allow a broader array of substrates to be ubiquitinated. I wondered if there were organismal lineages that lacked any hydroxyl-selective E2s. I know that at least the ERAD-linked E3 Doa10/MARCHF6 is missing from some lineages but am curious to see how this matches up with the J E2s. Not essential, but it might be interesting to look at.

*This is indeed a fascinating question, but one that we have not systematically pursued and that is also beyond our technical expertise. In our analysis, we did not find a eukaryotic lineage that lacked both J isoforms, but to make such a statement on the individual organism level is more challenging. As pointed out before by Lester et al. (2000, PMID: 10708578), some lineages have lost the J1 isoform, e.g. the yeast *S. cerevisiae* and *S. pombe*, but also *Drosophila melanogaster* and other insects (our own analysis). We did not find an organism that lacked the J2 isoform, but a definitive statement would require a more thorough investigation.*

*Both J families and both main E3 ligases involved in ERAD, Hrd1 and Doa10/MARCHF6, emerged early in eukaryotic evolution and were maybe already present in the last eukaryotic common ancestor. An analysis of the literature suggests that Hrd1 usually co-opts the Ube2J1 isoform (several report on the human proteins, e.g. (Mueller et al., 2008, PMID: 18711132); UBC32 in *Arabidopsis*, Chen et al., 2016, PMID: 27322605), whereas Doa10/MARCH6 cooperates with the Ube2J2 isoform. It is thus indeed tempting to speculate that indeed J family E2s evolved together with E3 ligases involved in ERAD.*

*However, as the reviewer points out, Doa10/MARCH6 has been lost in some eukaryotic lineages, for example in ciliates like *Paramecium tetraurelia*. Yet *Paramecium* contains both J homologs (UNIPROT entries J2: A0BPF8, J1: A0C0Q6), indicating that these E2s can also act with other E3. This is clearly also the case in other organism with an expanded E3 repertoire. In mammals, Ube2J2 has been found to co-operate with MARCHF5 in mitochondria (e.g., Lin et al., 2024, PMID 38366087), with RNF139 (van der Weijer, 2017, PMID: 28743740) and others. Ube2J1, which is associated with the Hrd1 complex in both mammals and plants (UBC32 in *Arabidopsis*, Chen et al., 2016, PMID: 27322605) also acts together with other E3 in mammals, e.g. with RNF26 (Cremer, 2021, PMID: 33472082). On the other hand, Sommer and colleagues showed that in yeast, Hrd1 can also co-operate with the J2 isoform, Ubc6 (e.g. Lips et al., 2020, PMID: 33015833), and work by the Knop lab indicated that other E3s also interact with Ubc6 (Asi3, Nam7, Pex10; Khmelinskii et al., 2014, PMID: 25519137). Together, these observations illustrate that the ubiquitin system is quite malleable so that it will be difficult to make a statement about co-evolution of ERAD E3s and J E2.*

To figure out the advantage a changed chemoselectivity provides, we think it would be interesting to investigate regulation of E2 activity, as ON/OFF switches for E2s with a different chemoselectivity would allow E3 ligases to expand their substrate spectrum on demand. In the case of Ube2J1, phosphorylation at Ser184 affects its activity and downstream degradation events (Oh et al., 2006, PMID: 16720581; Elangovan et al., 2017, PMID: 28321712). To our knowledge, Ube2J2 has not been investigated in that respect. We added such speculative remarks at the end of the discussion (line 557).

Alex Stein

ADDENDUM

In response to comments we received from the editor, we made two more corrections:

In Fig 6C, not all data that we deposited in the Source data to this panel was plotted for the condition “10 μ M RING-CTE”. This has been corrected.

In Figure EV4H,I we detected a mistake in the calculation of the error bars for the nucleophiles Lys and His. Since these ratios are based on single experiments for the Ubc6 mutants (but multiple experiments for WT Ubc6), no error estimate can be assigned. By mistake, in the previous version of the figure, the error estimate for the mono-exponential fit for the mutant dataset was combined with the standard deviation of the WT dataset to calculate the error associated with the ratio. Source data for this figure is provided.

Dr. Alexander Stein
Max Planck Institute for Multidisciplinary Sciences
Am Fassberg 11
Göttingen 37083
Germany

22nd Oct 2024

Re: EMBOJ-2024-117844R
Determinants of chemoselectivity in ubiquitination by the J2 family of ubiquitin conjugating enzymes

Dear Alex,

Thank you for submitting your revised manuscript to The EMBO Journal. Two of the original referees have now assessed it once again (see comments below), and both of them are overall satisfied with your responses and revisions. We shall therefore be happy to accept the study for publication, pending addressing of a few outstanding editorial issues:

- Please reduce the number of keywords on the title page to 5 (currently, there are 6), preferring broader terms over particular protein names.
- In Appendix Figure S4, the "Sbh2 DL800" blot appears to be identical in the + and - Ubc6 conditions (panels C and D). Please clarify and, if warranted, adjust, and also send us/upload the respective unmodified/full raw blots for the experiments in panels C and D, for our inspection. Finally, please also check the corresponding figure legend, which refers to panels D and E, instead of C and D?
- Please provide suggestions for a short 'blurb' text prefacing and summing up the study in two sentences (max. 250 characters), followed by 3-5 one-sentence 'bullet points' with brief factual statements of key results of the paper; they will form the basis of an editor-written 'Synopsis' accompanying the online version of the article. Please also upload a synopsis image, which can be used as a "visual title" for the synopsis section of your paper. The image should be in PNG or JPG format with the modest dimensions of EXACTLY 550 pixels wide and 300-600 pixels high.

I am therefore returning the study to you once more, to allow you to incorporate these final changes and upload all requested files. Once we have received them, we shall hopefully be able to swiftly proceed with acceptance and production of the manuscript.

With kind regards,

Hartmut

- 1) Every manuscript requires a Data Availability section (even if only stating that no deposited datasets are included). Primary datasets or computer code produced in the current study have to be deposited in appropriate public repositories prior to resubmission, and reviewer access details provided in case that public access is not yet allowed. Further information: embopress.org/page/journal/14602075/authorguide#dataavailability
- 2) Each figure legend must specify
 - size of the scale bars that are mandatory for all micrograph panels
 - the statistical test used to generate error bars and P-values
 - the type error bars (e.g., S.E.M., S.D.)
 - the number (n) and nature (biological or technical replicate) of independent experiments underlying each data point
 - Figures may not include error bars for experiments with $n < 3$; scatter plots showing individual data points should be used instead.

- 3) Revised manuscript text (including main tables, and figure legends for main and EV figures) has to be submitted as editable text file (e.g., .docx format). We encourage highlighting of changes (e.g., via text color) for the referees' reference.
- 4) Each main and each Expanded View (EV) figure should be uploaded as individual production-quality files (preferably in .eps, .tif, .jpg formats). For suggestions on figure preparation/layout, please refer to our Figure Preparation Guidelines: <http://bit.ly/EMBOPressFigurePreparationGuideline>
- 5) Point-by-point response letters should include the original referee comments in full together with your detailed responses to them (and to specific editor requests if applicable), and also be uploaded as editable (e.g., .docx) text files.
- 6) Please complete our Author Checklist, and make sure that information entered into the checklist is also reflected in the manuscript; the checklist will be available to readers as part of the Review Process File. A download link is found at the top of our Guide to Authors: embopress.org/page/journal/14602075/authorguide
- 7) All authors listed as (co-)corresponding need to deposit, in their respective author profiles in our submission system, a unique ORCID identifier linked to their name. Please see our Guide to Authors for detailed instructions.
- 8) Please note that supplementary information at EMBO Press has been superseded by the 'Expanded View' for inclusion of additional figures, tables, movies or datasets; with up to five EV Figures being typeset and directly accessible in the HTML version of the article. For details and guidance, please refer to: embopress.org/page/journal/14602075/authorguide#expandedview
- 9) To facilitate reproducibility and cross-laboratory adoption of methodologies, please structure the Materials & Methods section as outlined in our guide to authors, including a completed Reagents and Tools Table that can be downloaded from our author guidelines as well (<https://www.embopress.org/page/journal/14602075/authorguide#structuredmethods>).
- 10) Digital image enhancement is acceptable practice, as long as it accurately represents the original data and conforms to community standards. If a figure has been subjected to significant electronic manipulation, this must be clearly noted in the figure legend and/or the 'Materials and Methods' section. The editors reserve the right to request original versions of figures and the original images that were used to assemble the figure. Finally, we generally encourage uploading of numerical as well as gel/blot image source data; for details see: embopress.org/page/journal/14602075/authorguide#sourcedata

At EMBO Press, we ask authors to provide source data for the main manuscript figures. Our source data coordinator will contact you to discuss which figure panels we would need source data for and will also provide you with helpful tips on how to upload and organize the files.

In the interest of ensuring the conceptual advance provided by the work, we recommend submitting a revision within 3 months (20th Jan 2025). Please discuss the revision progress ahead of this time with the editor if you require more time to complete the revisions. Use the link below to submit your revision:

Link Not Available

Referee #1:

The authors have addressed our initial comments by carrying out additional experiments and modifying the text accordingly. We now feel that the manuscript is suitable for publication in EMBO Journal.

Referee #3:

My initial review of this paper was strongly positive. The response of the authors to my minor comments is fine. There are some significant questions still unanswered, such as the mechanism of Doa10 RING enhancement of Ubc6 activity (as originally pointed out by Reviewer 1), but getting the answers are evidently going to be difficult, and I would demand this for the current paper.

Dear Hartmut,

thank you for your email from last week and please apologize that it took us so long to respond. Below, please find our responses to the points raised.

- Please reduce the number of keywords on the title page to 5 (currently, there are 6), preferring broader terms over particular protein names.

We have uploaded the manuscript text with one keyword removed.

- In Appendix Figure S4, the "Sbh2 DL800" blot appears to be identical in the + and - Ubc6 conditions (panels C and D). Please clarify and, if warranted, adjust, and also send us/upload the respective unmodified/full raw blots for the experiments in panels C and D, for our inspection. Finally, please also check the corresponding figure legend, which refers to panels D and E, instead of C and D?

Thank you for finding this embarrassing mistake. As you point out correctly, the images in S4C and S4D are identical, with the one shown in S4D being the wrong one. We have replaced the image in S4D with the correct one, which is derived from the same original gel. For your inspection, we have uploaded the original raw images as Figure Source Data "Figure Source Data Appendix S4". Furthermore, we have corrected the corresponding figure legend. The corrected version of the Appendix has been uploaded.

- Please provide suggestions for a short 'blurb' text prefacing and summing up the study in two sentences (max. 250 characters), followed by 3-5 one-sentence 'bullet points' with brief factual statements of key results of the paper; they will form the basis of an editor-written 'Synopsis' accompanying the online version of the article. Please also upload a synopsis image, which can be used as a "visual title" for the synopsis section of your paper. The image should be in PNG or JPG format with the modest dimensions of EXACTLY 550 pixels wide and 300-600 pixels high.

We have uploaded a synopsis image and the short text. Please let me know if the image is according to the journals expectations or if something more sophisticated is required.

With kind regards

Alex

Dr. Alexander Stein
Max Planck Institute for Multidisciplinary Sciences
Am Fassberg 11
Göttingen 37083
Germany

31st Oct 2024

Re: EMBOJ-2024-117844R1
Determinants of chemoselectivity in ubiquitination by the J2 family of ubiquitin-conjugating enzymes

Dear Alex,

Thank you for submitting your final revised manuscript for our consideration. I am pleased to inform you that we have now accepted it for publication in The EMBO Journal.

With kind regards,

Hartmut
